# Asymmetric requirement of Dpp/BMP morphogen dispersal in the *Drosophila* wing disc

Shinya Matsuda [1✉], Jonas V. Schaefer [2], Yusuke Mii [3,4], Yutaro Hori [5], Dimitri Bieli[1], Masanori Taira[6,7], Andreas Plückthun [2] & Markus Affolter [1✉]

How morphogen gradients control patterning and growth in developing tissues remains largely unknown due to lack of tools manipulating morphogen gradients. Here, we generate two membrane-tethered protein binders that manipulate different aspects of Decapentaplegic (Dpp), a morphogen required for overall patterning and growth of the *Drosophila* wing. One is "HA trap" based on a single-chain variable fragment (scFv) against the HA tag that traps HA-Dpp to mainly block its dispersal, the other is "Dpp trap" based on a Designed Ankyrin Repeat Protein (DARPin) against Dpp that traps Dpp to block both its dispersal and signaling. Using these tools, we found that, while posterior patterning and growth require Dpp dispersal, anterior patterning and growth largely proceed without Dpp dispersal. We show that *dpp* transcriptional refinement from an initially uniform to a localized expression and persistent signaling in transient *dpp* source cells render the anterior compartment robust against the absence of Dpp dispersal. Furthermore, despite a critical requirement of *dpp* for the overall wing growth, neither Dpp dispersal nor direct signaling is critical for lateral wing growth after wing pouch specification. These results challenge the long-standing dogma that Dpp dispersal is strictly required to control and coordinate overall wing patterning and growth.

[1] Biozentrum, University of Basel, Basel, Switzerland. [2] Department of Biochemistry, University of Zurich, Zurich, Switzerland. [3] National Institute for Basic Biology and Exploratory Research Center on Life and Living Systems (ExCELLS), National Institutes of Natural Sciences, Okazaki, Aichi, Japan. [4] JST PRESTO, Kawaguchi, Saitama, Japan. [5] Institute for Quantitative Biosciences, The University of Tokyo, Tokyo, Japan. [6] Department of Biological Sciences, Graduate School of Science, The University of Tokyo, Tokyo, Japan. [7] Present address: Department of Biological Sciences, Faculty of Science and Engineering, Chuo University, Tokyo, Japan. ✉email: shinya.matsuda@unibas.ch; markus.affolter@unibas.ch

A fundamental question in developmental biology is how proteins work together to orchestrate developmental processes. Forward and reverse genetic approaches based on mutants and RNAi, together with biochemical analyses, provide insights into how proteins function. However, interpretational gaps often remain between the mutant phenotypes and the underlying mechanisms.

Recently, small, high-affinity protein binders, such as nanobodies, single-chain variable fragments (scFvs), Designed Ankyrin Repeat Proteins (DARPins), and others, have emerged as versatile tools to fill this gap. By fusing these protein binders to well-characterized protein domains and expressing the fusion proteins in vivo, protein function can be directly manipulated in a predicted manner[1–5]. For example, when a protein functions with multiple parameters, protein binder tools targeting each or a subset of these parameters could help to dissect the requirement of each parameter. However, it remains challenging to design and customize such protein binder tools.

A class of molecules that exert its function with multiple parameters are morphogens, secreted molecules that disperse from a localized source and regulate target gene expression in a concentration-dependent manner[6–9]. A morphogen gradient is characterized by its parameters such as rates of secretion, diffusion, and degradation[10]. Temporal dynamics of a morphogen gradient also impact cell fates decisions[11]. Despite a variety of parameters involved, morphogen dispersal is generally thought to be critical for morphogen function based on severe morphogen mutant phenotypes and long-range action of morphogens to reorganize patterning and growth.

However, a recent study challenged this basic assumption for the case of the Wingless (Wg) morphogen, the main Wnt in *Drosophila*, by showing that a membrane-tethered non-diffusible form of Wg can replace the endogenous Wg without strongly affecting appendage development[12]. Although the precise contribution of Wg dispersal requires further investigations[13–16], the study raises the question of how important morphogen dispersal is for tissue patterning and growth in general.

In contrast to Wg, *Decapentaplegic* (*dpp*), the vertebrate BMP2/4 homolog, is thought to act as a bona fide morphogen in the *Drosophila* prospective wing. Dpp disperses from a narrow anterior stripe of cells along the anterior−posterior (A−P) compartment boundary to establish a characteristic morphogen gradient in both compartments (Fig. 1a)[17,18]. How the Dpp dispersal-mediated morphogen gradient achieves and coordinates overall wing patterning and growth has served as a paradigm to study morphogens[19]. However, despite intensive studies, it remains controversial how Dpp/BMP disperses[17,20–23], controls growth[24–33], and coordinates patterning and growth (i.e. scaling)[32,34–36]. Regardless of the actual mechanisms, all the studies are based on the assumption that Dpp dispersal from the anterior stripe of cells controls overall wing patterning and growth, in line with the severe *dpp* mutant phenotypes.

To directly manipulate dispersal of Dpp, we recently generated morphotrap, a membrane-tethered anti-GFP nanobody, to trap GFP-tagged Dpp and thereby manipulate its dispersal[37]. Using morphotrap, the authors showed that a substantial amount of GFP-Dpp secreted from the anterior stripe of cells can reach the peripheral wing disc and that blocking GFP-Dpp dispersal from the source cells causes severe adult wing patterning and growth defects[37]. These results support the critical role of Dpp dispersal for overall wing patterning and growth[37]. However, the application of morphotrap was limited to rescue conditions by overexpression of GFP-Dpp, due to the lack of an endogenous *GFP-dpp* allele.

In this study, we first generated a platform to manipulate the *dpp* locus and inserted a tag into *dpp* in order to investigate the precise requirement of the endogenous Dpp morphogen gradient for wing patterning and growth. We found that while a *HA-dpp* allele was functional, a *GFP-dpp* allele was not, thus limiting morphotrap application. To manipulate the endogenous Dpp morphogen gradient, we then generated two protein binder tools analogous to morphotrap. One is HA trap based on anti-HA scFv that traps HA-Dpp through the HA tag to mainly block Dpp dispersal, the other is Dpp trap based on anti-Dpp DARPin that directly binds to Dpp to block Dpp dispersal and signaling in the source cells. Thus, these tools allowed us to distinguish the requirements of Dpp dispersal and cell-autonomous signaling in the source cells for wing pouch growth and patterning.

Here, we show, using these tools, that while posterior patterning and growth require Dpp dispersal, anterior patterning and growth largely proceed without Dpp dispersal but require cell-autonomous Dpp signaling in the source cells. We show that *dpp* transcriptional refinement from an initially uniform to a localized expression and persistent signaling in transient *dpp* source cells allow relatively normal anterior patterning and growth despite the absence of Dpp dispersal. Furthermore, despite a critical requirement of *dpp* for overall wing growth, we also find that neither Dpp dispersal nor direct signaling is critical for the lateral wing pouch to grow once the wing pouch is defined. These results challenge the long-standing dogma that Dpp dispersal controls overall wing patterning and growth and call for a revision of how Dpp controls and coordinates wing patterning and growth.

## Results

**Generation of a functional *HA-dpp* allele**. To manipulate the endogenous Dpp morphogen gradient, we utilized a MiMIC transposon inserted in the *dpp* locus (*dpp^{MI03752}*), which allows to replace the sequence between the two *attP* sites in the transposon with any sequence inserted between two inverted *attB* sites upon integrase expression[38]. A genomic fragment containing sequences encoding a tagged version of *dpp* followed by an FRT and a marker was first inserted into the locus (Fig. 1b), then the endogenous *dpp* exon was removed upon FLP/FRT recombination to keep only the tagged *dpp* exon (Fig. 1b). Using this strategy, we inserted different tags into the *dpp* locus and found that while a *GFP-dpp* allele was homozygous lethal during early embryogenesis, a *HA-dpp* allele was functional without obvious phenotypes[39] (Fig. 1c, see below "Methods"). Immunostainings for the HA-tag including permeabilization steps showed HA-Dpp expression in an anterior stripe of cells along the A−P compartment boundary in the late third instar wing disc (Fig. 1d). In contrast, immunostainings for the HA-tag without permeabilization, which allows antibodies to access only the extracellular antigens, revealed that a shallow extracellular HA-Dpp gradient overlapped with the gradient of phosphorylated Mad (pMad), a downstream transcription factor of Dpp signaling (Fig. 1e). Similar HA tag knock-in *dpp* alleles have recently been generated by a CRISPR approach[25].

**Generation and characterization of HA trap**. Since we could not apply morphotrap due to the lethality of the *GFP-dpp* allele, we generated an HA trap, analogous to morphotrap. HA trap consists of an anti-HA scFv[40] fused to the transmembrane domain of CD8 and mCherry (Fig. 2a). HA trap expression in the anterior stripe of cells of wild-type wing discs using *ptc-Gal4* did not interfere with Dpp signaling in the wing disc or patterning and growth of the adult wing (Supplementary Fig. 1). Thus, HA trap is inert in the absence of an HA-tagged protein. While we attempted to visualize extracellular HA-Dpp distribution upon HA trap expression, we noticed that the HA tag can no longer be

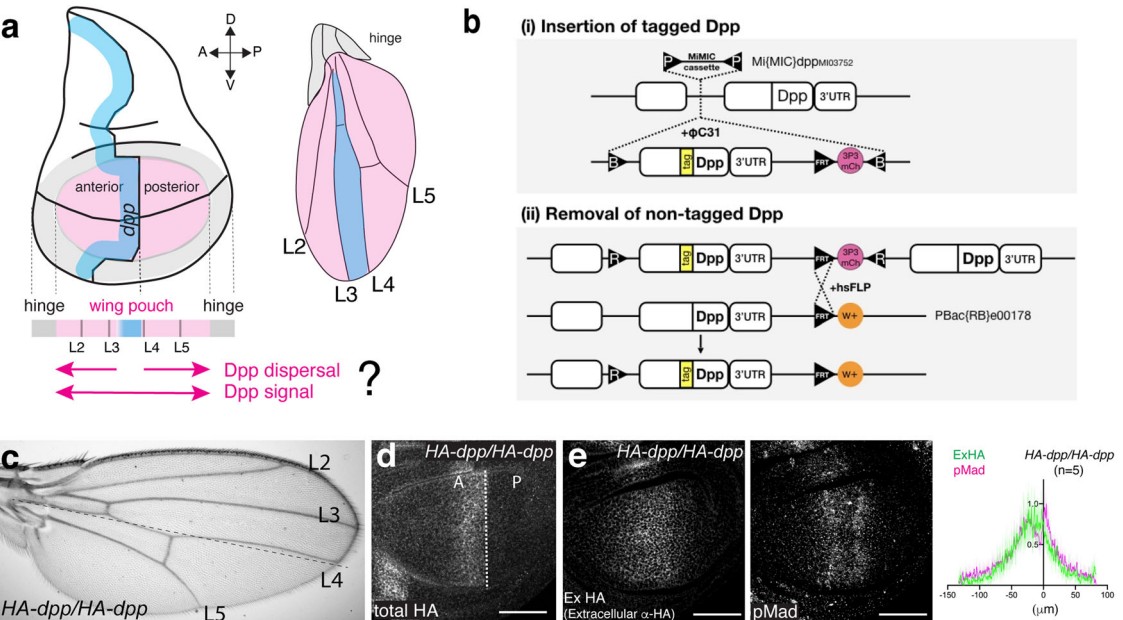

**Fig. 1 Generation of a functional *HA-dpp* allele. a** A schematic view of the wing disc and the adult wing. **b** A schematic view of a platform manipulating endogenous *dpp* locus. **c** Adult wing of a homozygous *HA-dpp* fly. **d** Conventional α-HA staining of *HA-dpp* homozygous wing disc. **e** Extracellular α-HA staining (Ex HA) and α-pMad staining of *HA-dpp* homozygous wing disc. Average fluorescence intensity profile of extracellular HA and pMad staining of *HA-dpp/HA-dpp* wing disc (n = 5). Data are presented as mean ± SD. Scale bar 50 μm.

used for immunostaining when bound to HA trap. We therefore additionally inserted an Ollas tag to generate a functional *Ollas-HA-dpp* allele in order to visualize the extracellular Dpp distribution using the antibody against the Ollas tag. The extracellular Ollas-HA-Dpp gradient was similar to the extracellular HA-Dpp gradient (Fig. 2b, c).

To test if HA trap can efficiently trap Ollas-HA-Dpp in the Dpp-producing cells, HA trap was expressed in the anterior stripe of cells using *ptc-Gal4* in Ollas-HA-Dpp heterozygous wing discs, since *ptc-Gal4* expression largely overlaps with *dpp*-producing cells[41]. Under this condition, extracellular immunostainings for the Ollas-tag revealed that Ollas-HA-Dpp accumulated on the anterior stripe of cells, and that the extracellular gradient was abolished (Fig. 2d, e). To test if HA trap can trap Ollas-HA-Dpp outside the anterior stripe of cells, clones of cells expressing *Gal4* were randomly induced by heat-shock inducible FLP to express HA trap under UAS control. We found that Ollas-HA-Dpp accumulated in clones of cells expressing HA trap induced outside the main *dpp* source cells in both compartments (Fig. 2f–i, arrow). If HA trap can efficiently trap Ollas-HA-Dpp in the source cells, the clonal Ollas-HA-Dpp accumulation should be blocked upon HA trap expression in the source cells. Indeed, we found that clonal Ollas-HA-Dpp accumulation in both compartments was drastically reduced upon HA trap expression using *ptc-Gal4* (Fig. 2j–k, arrow), indicating that the HA trap can block HA-Dpp dispersal efficiently.

It has been shown that overexpression of GFP-Dpp from the anterior stripe cells leads to accumulation of GFP-Dpp in clones of cells expressing morphotrap in the peripheral regions[37]. In contrast, we found that Ollas-HA-Dpp accumulated in clones of cells expressing HA trap near the source cells but not in the peripheral regions (Fig. 2l, arrowhead). This raises a question whether Dpp can act in the peripheral regions at physiological levels.

**Asymmetric patterning and growth defects by HA trap.** After we validated that HA trap can efficiently block Dpp dispersal, we then expressed HA trap using different Gal4 driver lines in *HA-*

*dpp* homozygous wing discs to address the requirement of Dpp dispersal. Normally, Dpp binds to the Dpp receptors Thickveins (Tkv) and Punt, inducing a pMad gradient and an inverse gradient of Brk, a transcription repressor repressed by Dpp signaling. The two opposite gradients regulate growth and patterning (nested target gene expression, such as *sal*, and *omb*) to define adult wing vein positions (such as L2 and L5) (Fig. 3a)[10,19,42–44].

Upon HA trap expression in the anterior stripe of cells using *ptc-Gal4*, pMad, Sal, and Omb expression were undetectable in the P compartment and Brk was also upregulated in the P compartment (Fig. 3b−f), indicating that HA trap efficiently blocked HA-Dpp dispersal from source cells and interfered with patterning. The posterior wing pouch growth was also affected as revealed by the expression of an intervein marker DSRF and a wing pouch marker *5xQE.DsRed*[45] (Fig. 3b arrow, 3g). Interestingly, although *5xQE.DsRed* contains five copies of the 806 bp Quadrant Enhancer (QE) of the wing master gene *vg* containing a Mad binding site and is therefore thought to be directly regulated by Dpp signaling[46,47], *5xQE.DsRed* remained expressed in the P compartment without detectable Dpp signaling (Fig. 3b, arrow). In the A compartment, pMad was slightly reduced in the anterior medial region (Fig. 3b, c), probably because HA trap partially blocked Dpp signaling upon binding to HA-Dpp (Fig. 2l, arrow). Nevertheless, the anterior Brk gradient was not strongly affected (Fig. 3b, d). Although maximum intensity of Sal or Omb was reduced, nested expression of Sal and Omb was maintained in the A compartment and the anterior growth defects was milder than the posterior growth defects (Fig. 3b, e, f, g). Consistent with these phenotypes in the wing discs, while posterior patterning and growth were severely affected, anterior patterning and growth were relatively normal in the resulting adult wings (Fig. 3h−j).

Similar asymmetric defects in patterning and growth were observed upon HA trap expression in the region covering the entire wing pouch using *nub-Gal4* (Fig. 3k−t) or in the entire anterior compartment using *ci-Gal4* (Supplementary Fig. 2a−j). Furthermore, even when HA trap was expressed using both *nub-Gal4* and *ptc-Gal4*, the resulting phenotypes were not enhanced (Supplementary Fig. 3). To test whether the posterior growth

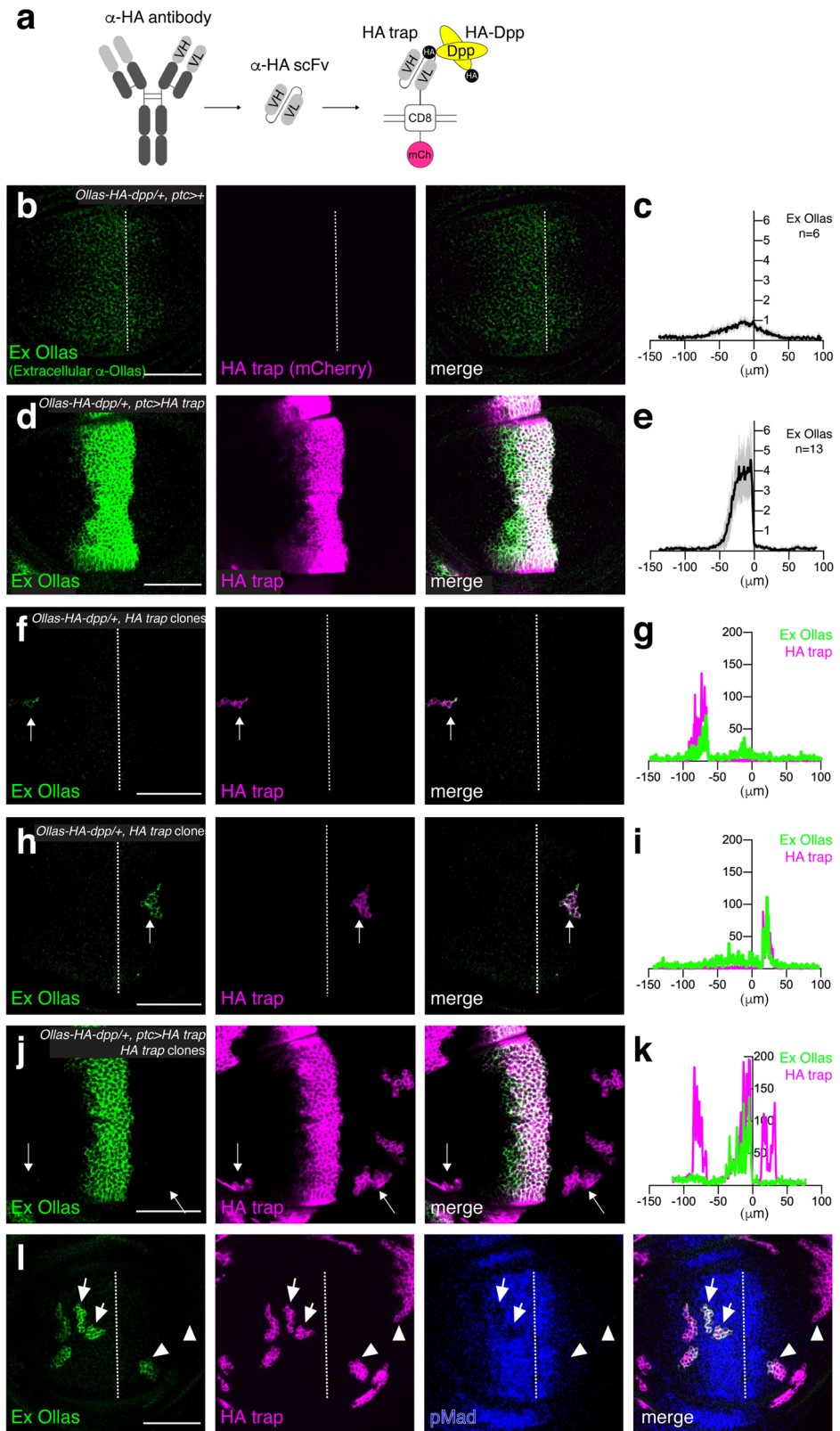

defects upon HA trap expression is caused by cell death, Caspase-3 was analyzed. We found that Caspase-3 was not upregulated upon HA trap expression, and blocking apoptosis by apoptosis inhibitor p35 did not rescue these growth defects upon HA trap expression (Supplementary Fig. 4). Thus, the posterior growth defects upon HA trap expression is not caused by cell death. Taken together, these results suggest that, while critical for

posterior patterning and growth, Dpp dispersal is largely dispensable for anterior patterning and growth.

**Lateral wing pouch growth without Dpp signaling**. A critical role of Dpp dispersal for posterior patterning and growth is consistent with a role of Dpp as a morphogen. However, the overall phenotypes caused by HA trap was surprisingly mild

**Fig. 2 Generation and characterization of HA trap. a** A schematic view of HA trap (VH variable heavy chain, VL variable light chain, mCh mCherry). **b−e** Extracellular α-Ollas staining (Ex Ollas), HA trap (mCherry), and merge of control *Ollas-HA-dpp/+* disc (**b**), and of *Ollas-HA-dpp/+ ptc > HA trap* disc (**d**). **c, e** Average fluorescence intensity profile of extracellular α-Ollas staining of (**b**) and (**d**) respectively. *Ollas-HA-dpp, ptc > +* disc (control) (n = 6) (**c**), and *Ollas-HA-dpp, ptc > HA trap* disc (n = 13) (**e**). Data are presented as mean ± SD. **f−k** Extracellular α-Ollas staining, HA trap (mCherry), and merge of *Ollas-HA-dpp/+* disc with an anterior clone of cells expressing HA trap (**f**), of *Ollas-HA-dpp/+* disc with a posterior clone of cells expressing HA trap (**h**), and of *Ollas-HA-dpp/+* disc with HA trap expression using *ptc-Gal4* and clones of cells expressing HA trap in both compartments (**j**). **g, i, k** Quantification of extracellular α-Ollas staining and HA trap (mCherry) of (**f**), (**h**), (**j**), respectively. Arrows indicate clones of cells expressing HA trap where quantification was performed. **l** Extracellular α-Ollas staining, HA trap (mCherry), pMad, and merge of *Ollas-HA-dpp/+* wing disc with clones of cells expressing HA trap. Arrows indicate clones of cells expressing HA trap where pMad signal is reduced upon trapping Ollas-HA-Dpp. Arrow heads indicate a clone of cells expressing HA trap that accumulates Ollas-HA-Dpp near the source cells and a clone of cells expressing HA trap that does not accumulate Ollas-HA-Dpp far from the source cells. Dashed white lines mark the A−P compartment border. Scale bar 50 μm.

when compared to the phenotypes seen in *dpp* mutants (see below). Given the requirement of Dpp signaling for cell proliferation and survival in the entire wing pouch[48], it was surprising that about 40% of the posterior wing pouch was able to grow and differentiate into adult wing tissue without detectable Dpp signaling (Fig. 3).

We therefore tested whether the posterior growth and *5xQE.DsRed* expression seen upon HA trap expression is caused by low levels of HA-Dpp leaking from the HA trap expressed in the source. In this case, the posterior growth and *5xQE.DsRed* expression seen upon HA trap expression should be dependent on *tkv*, an essential receptor for Dpp signaling. To test this, mutant clones of *tkv^{a12}* (characterized as a null allele[49,50]) were induced in wing discs expressing HA trap with *ptc-Gal4* between mid-second and beginning of third instar stages and analyzed in the late third instar stage. We found that *tkv^{a12}* clones often survived and expressed the *5xQE.DsRed* reporter in the anterior lateral regions as well as in the entire posterior region. We also noticed that *tkv^{a12}* clones survived and expressed the *5xQE.DsRed* reporter even next to the source cells in the P compartment (Fig. 4a). These results indicate that the lateral growth and *5xQE.DsRed* expression seen upon HA trap expression is independent of Dpp signaling, and not caused by a leakage of HA-Dpp from the HA trap, even if such leakage would occur.

To test whether Dpp signaling-independent growth occurs also during normal development, *tkv^{a12}* clones were induced in the wild-type wing disc during mid-second and early third instar stage. We found that *tkv^{a12}* clones were eliminated from the medial regions but often survived and expressed the *5xQE.DsRed* reporter in the lateral wing pouch (Fig. 4b). Since *tkv^{a12}* may not be a complete null allele, we then inserted an FRT cassette in the *tkv* locus and generated a *tkv* flip-out allele (*tkvHA^{FO}*) to induce FLP/FRT-mediated excision of *tkv*. By generating *tkv* null clones upon heat-shock inducible FLP expression, we confirmed that *tkv* null clones often survived and expressed the *5xQE.DsRed* reporter in the lateral wing pouch (Fig. 4c, d, arrow). We also found that, while most often eliminated, medial *tkv* null clones survived and expressed *5xQE.DsRed* in rare cases (Fig. 4d), indicating that Dpp signaling is dispensable for *5xQE.DsRed* expression also in the medial region, but medial cells lacking Dpp signaling are normally eliminated[48].

How can Dpp signaling-independent wing pouch growth and *5xQE.DsRed* expression be reconciled with a critical role of Dpp signaling for the entire wing pouch growth?[48] First, *tkv* clones generated in the developing wing pouch have been shown to be eliminated by apoptosis or extrusion and do not survive in the adult wing[48,51]. However, *tkv* clones survive better in the P compartment where Dpp signaling is blocked by HA trap (Fig. 4a) and in the lateral region of wild-type wing disc where Dpp signaling is generally low (Fig. 4b−d). This raises a possibility that *tkv* clones are eliminated when surrounded by wild-type cells, even if *tkv* clones could grow and survive to a

certain extent. Second, wing pouch and *5xQE.DsRed* expression were completely lost in *dpp* mutants (see below). It has been shown that initial wing pouch specification is mediated by Dpp derived from the peripodial membrane, which covers the developing wing pouch, and this early *dpp* expression in the peripodial membrane is lost in *dpp* disc alleles[52]. Thus, wing pouch and *5xQE.DsRed* expression could be lost in *dpp* disc alleles due to failure of initial specification of the wing disc and subsequent elimination of cells.

To minimize these potential problems, we applied Gal80ts to conditionally remove *dpp* from the entire A compartment using *ci-Gal4*. At the permissive temperature of 18 °C, Gal80ts actively represses Gal4 activity. At restrictive temperature of 29 °C, Gal80ts can no longer block Gal4 activity; thus, Gal4 can be temporally activated using temperature shifts. Upon FLP expression, *dpp* was removed by FLP/FRT-mediated excision via *dpp^{FO}* allele[24], in which an FRT cassette was inserted into the *dpp* locus. To remove *dpp* from the beginning of second instar stage when the wing pouch is specified, the larvae were raised at 18 °C for 4 days and then shifted to 29 °C. By removing *dpp* from the entire A compartment using *ci-Gal4* under this condition, we found that *5xQE.DsRed* remained expressed despite severe growth defects in the late third instar stage (Fig. 4e−h). Similarly, genetic removal of *tkv* via *tkvHA^{FO}* from the A compartment using *ci-Gal4* or from the P compartment using *hh-Gal4* from the second instar stage revealed that, despite severe growth defects, *5xQE.DsRed* remained expressed in each compartment lacking *tkv* (Supplementary Fig. 5). Surprisingly, similar results were obtained even when *tkv* was removed from the entire P compartment using *hh-Gal4* from the embryonic stages without Gal80ts (Supplementary Fig. 6). These results further support the presence of Dpp signaling-independent *5xQE.DsRed* expression and wing pouch growth.

How is *5xQE.DsRed* expression regulated if QE is not directly regulated by Dpp signaling? While *5xQE.DsRed* expression is completely lost in *dpp* mutants, we found that *5xQE.DsRed* reporter expression was rescued in *dpp*, *brk* double mutant wing discs (Supplementary Fig. 7), indicating that *5xQE.DsRed* expression is largely induced by repressing *brk*, similar to the regulation of other *dpp* target genes. Indeed, QE has been shown to be activated in *brk* mutant clones in the wing disc[53]. However, this notion appears inconsistent with the fact that *5xQE.DsRed* expression was not repressed in the region where Brk is high in various conditions, in which Dpp signaling is compromised (Figs. 3b, 4h and Supplementary Fig. 5d′, h′). We noticed that the observed high Brk levels upon Dpp trapping were comparable to the Brk level in the lateral region of the control wing disc (Fig. 3d, n), and Brk and *5xQE.DsRed* were co-expressed in the lateral region of the control wing disc (Fig. 4f and Supplementary Fig. 5b′ f′). Thus, we speculate that Brk is not sufficient to repress *5xQE.DsRed* expression at physiological levels in lateral regions and that there are additional inductive inputs such as Wg[45,54].

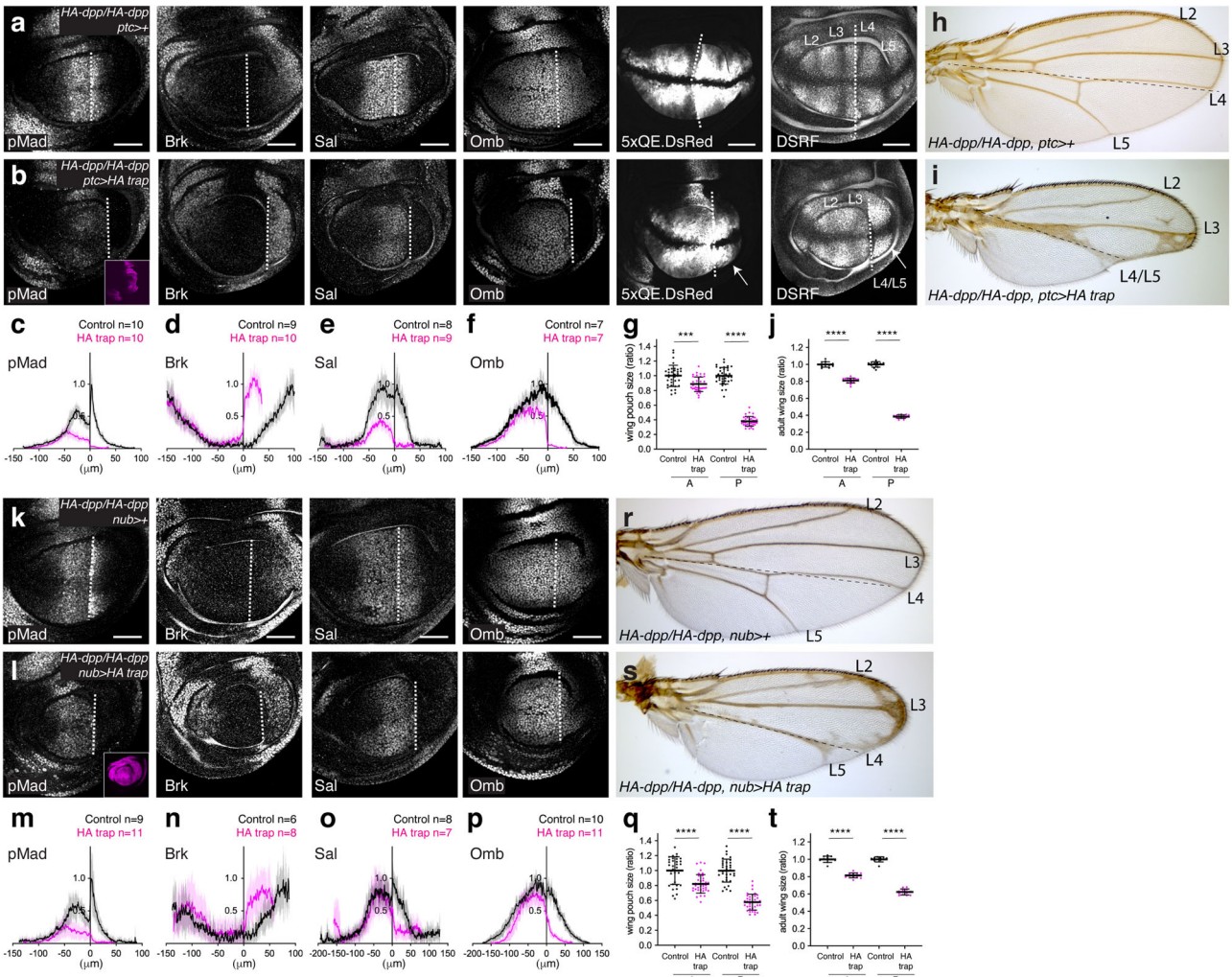

**Fig. 3 Asymmetric and minor patterning and growth defects by HA trap. a, b** α-pMad, α-Brk, α-Sal, α-Omb, 5xQE.DsRed, DSRF, and HA trap (mCherry) (inset) of *HA-dpp/HA-dpp, ptc > +* control wing disc (**a**) and *HA-dpp/HA-dpp, ptc > HA trap* wing disc (**b**). **c−f** Average fluorescence intensity profile of α-pMad (**c**), α-Brk (**d**), α-Sal (**e**), α-Omb (**f**) staining in (**a**, **b**). Data are presented as mean ± SD. **g** Comparison of compartment size of *HA-dpp/HA-dpp, ptc > +* control wing pouch (*n* = 35) and *HA-dpp/HA-dpp, ptc > HA trap* wing pouch (*n* = 37). Data are presented as mean ± SD. Two-sided unpaired Student's *t* test with unequal variance was used for the comparison of the A compartment (*p* = 0.0002) and for comparison of the P compartment (*p* < 0.0001). ****p* < 0.001, *****p* < 0.0001. **h, i** Adult wing of *HA-dpp/HA-dpp, ptc > +* (control) (**h**) and *HA-dpp/HA-dpp, ptc > HA trap* (**i**). **j** Comparison of compartment size of (**h**) and (**i**). *HA-dpp/HA-dpp, ptc > +* control adult wing (*n* = 12) and *HA-dpp/HA-dpp, ptc > HA trap* adult wing (*n* = 16). Data are presented as mean ± SD. Two-sided unpaired Student's *t* test with unequal variance was used for comparison of the A compartment (*p* < 0.0001) and for comparison of the P compartment (*p* < 0.0001). *****p* < 0.0001. **k, l** α-pMad, α-Brk, α-Sal, α-Omb, and HA trap (mCherry) (inset) of *HA-dpp/HA-dpp, nub > +* control wing disc (**k**) and *HA-dpp/HA-dpp, nub > HA trap* wing disc (**l**). **m−p** Average fluorescence intensity profile of α-pMad (**m**), α-Brk (**n**), α-Sal (**o**), α-Omb (**p**) staining in (**k**, **l**). Data are presented as mean ± SD. **q** Comparison of compartment size of *HA-dpp/HA-dpp, nub > +* control wing pouch (*n* = 33) and *HA-dpp/HA-dpp, nub > HA trap* wing pouch (*n* = 38). Data are presented as mean ± SD. Two-sided unpaired Student's *t* test with unequal variance was used for comparison of the A compartment (*p* < 0.0001) and for comparison of the P compartment (*p* < 0.0001). *****p* < 0.0001. **r, s** Adult wing of *HA-dpp/HA-dpp, nub > +* (control) (**r**) and *HA-dpp/HA-dpp, nub > HA trap* (**s**). **t** Comparison of compartment size of (**r**) and (**s**). *HA-dpp/HA-dpp, nub > +* control adult wing (*n* = 11) and *HA-dpp/HA-dpp, nub > HA trap* adult wing (*n* = 12). Data are presented as mean ± SD. Two-sided unpaired Student's *t* test with unequal variance was used for comparison of the A compartment (*p* < 0.0001). Two-sided Mann−Whitney test was used for comparison of the P compartment (*p* < 0.0001). *****p* < 0.0001. Dashed white lines mark the A−P compartment border. Scale bar 50 μm.

**Severe patterning and growth defects by Dpp trap.** Even if the lateral wing pouch region can grow independent of Dpp signaling after wing pouch specification (Fig. 4e−h), this growth cannot account for the overall minor growth phenotypes caused by HA trap (Fig. 3 and Supplementary Fig. 2a−j). How can relatively normal patterning and growth be achieved without Dpp dispersal? Since pMad was completely lost in *dpp* mutants (Fig. 4g) but remained active in the source cells upon HA trap expression (Fig. 3 and Supplementary Fig. 2a−j), we asked whether Dpp signaling in the source cells could account for the minor phenotypes caused by HA trap.

To test this, we selected DARPins, protein binders based on ankyrin repeats[55–57], that bind to the mature Dpp ligand and block Dpp signaling. For each of the 36 candidates obtained from the in vitro screening, we generated a Dpp trap by fusing the anti-Dpp DARPin to the transmembrane domain of CD8 and mCherry (Fig. 5a). By expressing each trap in the wing disc, we identified one Dpp trap (containing DARPin 1242_F1), which efficiently blocked Dpp dispersal (Fig. 5b) and signaling (Fig. 5c, d). We found that the expression of the Dpp trap using *ptc-Gal4* (Fig. 5c−i), *nub-Gal4* (Fig. 5k−t), and *ci-Gal4* (Supplementary Fig. 2k−t) caused severe signaling defects as well as patterning

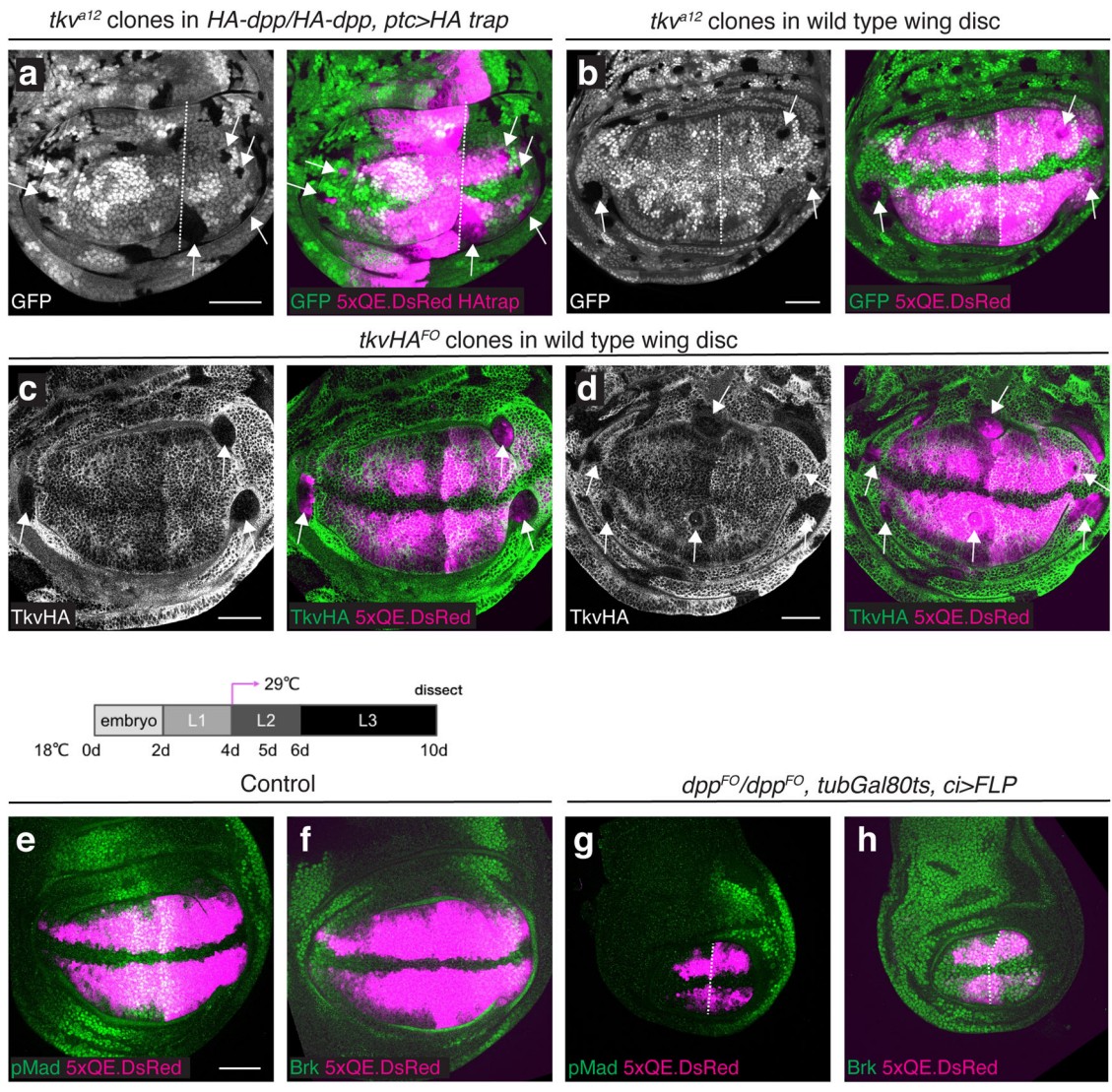

**Fig. 4 Lateral wing pouch growth without Dpp signaling. a, b** $tkv^{a12}$ clones (indicated by the absence of GFP signal) induced in *HA-dpp/HA-dpp, ptc > HA* trap wing discs (**a**) and in wild-type wing discs (**b**). **c, d** $tkvHA^{FO}$ clones (indicated by the absence of α-HA staining) in wild-type wing discs. Clones were induced at 60−72 h AEL (after egg laying) during mid-second to early third instar stages. **e**−**h** α-pMad and 5xQE.DsRed (**e, g**) and α-Brk and 5xQE.DsRed (**f, h**) of control wing disc (**e, f**) and *5xQE.DsRed, dpp^{FO}/dpp^{FO}, tubGal80ts, ci > UAS-FLP* wing disc (**g, h**). Crosses were shifted from 18 °C to 29 °C at 4-day AEL (early second instar). Scale bar 50 μm.

and growth defects, similar to *dpp* mutants (Fig. 4g, h). Adult wings expressing Dpp trap using *nub-Gal4* were recovered and also showed severe patterning and growth defects comparable to *dpp* mutants (Fig. 5s). Although Caspase-3 was upregulated upon Dpp trap expression (Supplementary Fig. 4a, c, d), the growth defects were not rescued by p35 (Supplementary Fig. 4h−j), indicating that apoptosis was not the main cause of growth defects caused by Dpp trap. Furthermore, these severe phenotypes were not due to a common scaffold effect of DARPins, since one of the traps (containing DARPin 1240_C9) that failed to trap Dpp did not interfere with pMad accumulation in the wing disc or patterning and growth of the adult wing when expressed using *ptc-Gal4* (Supplementary Fig. 8).

We note that upon Dpp trap expression, Sal expression was lost from the medial region but appeared to be upregulated in the lateral region (Fig. 5d, l and Supplementary Fig. 2o, arrow). It has previously been shown that Sal is expressed not only in the medial region but also in the lateral region[58]. The same study also showed that the medial Sal expression is Dpp signaling-dependent but lateral Sal expression is Brk-dependent. Thus,

upregulation of Brk upon Dpp trap expression could cause the lateral Sal upregulation. However, when we focused on the peripheral region of the control wing disc (basal confocal section), we noticed that the lateral Sal expression of the control wing disc was actually comparable to that of the wing disc expressing Dpp trap (Supplementary Fig. 9a, b). Thus, when we focused on the medial Sal expression (apical confocal section), the lateral Sal expression of the control wing disc was simply missed due to the tissue architecture. Consistently, when *dpp* was removed from the entire A compartment using *ci-Gal4* from mid-second instar stage, Sal expression was lost from the medial region but not significantly upregulated in the lateral region (Supplementary Fig. 9c−e).

By comparing the phenotypes caused by Dpp trap and HA trap, we noticed that the phenotypes caused by Dpp trap were much stronger than those caused by HA trap (Figs. 3, 5 and Supplementary Fig. 2). Indeed, comparison of each compartment size when expressing HA trap and Dpp trap using different Gal4 driver lines also showed that each compartment size was smaller upon Dpp trap expression than upon HA trap expression (Fig. 5j,

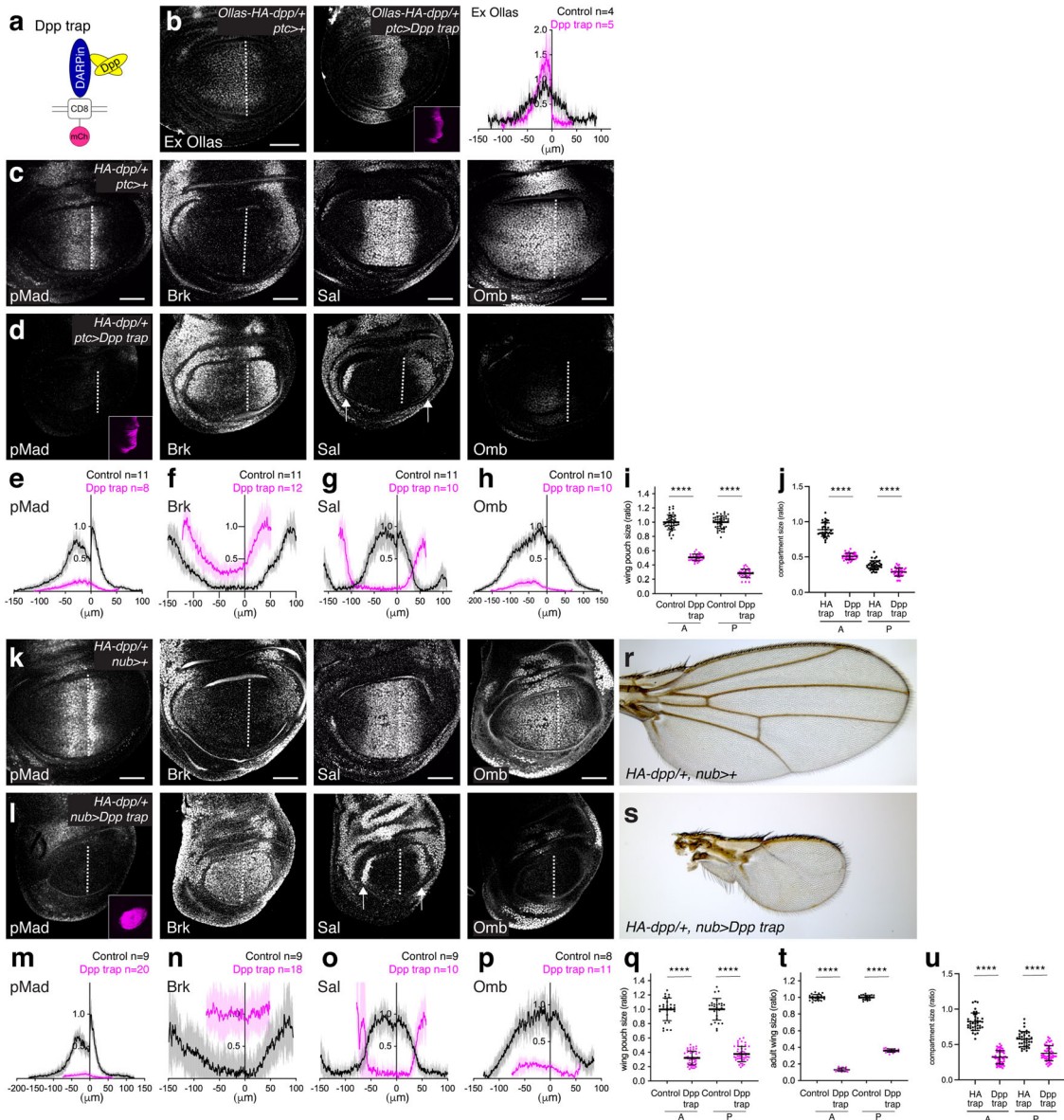

**Fig. 5 Severe patterning and growth defects by Dpp trap. a** A schematic view of Dpp trap based on DARPins against Dpp (mCh mCherry). **b** Extracellular α-Ollas staining and Dpp trap expression (mCherry) (inset) of *Ollas-HA-dpp/+*, *ptc >* + control wing disc (left) and *Ollas-HA-dpp/+*, *ptc > Dpp trap* (right). Average fluorescence intensity profile of extracellular α-Ollas staining of *Ollas-HA-dpp/+*, *ptc >* + wing disc (control) (*n* = 4) and *Ollas-HA-dpp/+*, *ptc > Dpp trap* wing disc (*n* = 5). Data are presented as mean ± SD. **c**, **d** α-pMad, α-Brk, α-Sal, α-Omb staining, and Dpp trap (mCherry) expression (inset) of *HA-dpp/+*, *ptc >* + control wing disc (**c**) and *HA-dpp/+*, *ptc > Dpp trap* wing disc (**d**). **e−h** Average fluorescence intensity profile of α-pMad (**e**), α-Brk (**f**), α-Sal (**g**), α-Omb (**h**) staining in (**c**, **d**). Data are presented as mean ± SD. **i** Comparison of compartment size of *HA-dpp/+*, *ptc >* + control wing pouch (*n* = 44) and *HA-dpp/+*, *ptc > Dpp trap* wing pouch (*n* = 39). Data are presented as mean ± SD. Two-sided unpaired Student's *t* test with unequal variance was used for comparison of the A compartment (*p* < 0.0001) and for comparison of the P compartment (*p* < 0.0001). ****$p$ < 0.0001. **j** Comparison of normalized compartment size of wing pouch upon HA trap (*n* = 37) and Dpp trap (*n* = 39) expression using *ptc-Gal4* (the same data set from Figs. 3g and 5i). Data are presented as mean ± SD. Two-sided unpaired Student's *t* test with unequal variance was used for comparison of the A compartment (*p* < 0.0001) and for comparison of the P compartment (*p* < 0.0001). ****$p$ < 0.0001. **k**, **l** α-pMad, α-Brk, α-Sal, α-Omb staining, and Dpp trap (mCherry) expression (inset) of *HA-dpp/+*, *nub >* + control wing disc (**k**) and *HA-dpp/+*, *nub > Dpp trap* wing disc (**l**). **m−p** Average fluorescence intensity profile of α-pMad (**m**), α-Brk (**n**), α-Sal (**o**), α-Omb (**p**) staining in (**k**, **l**). Data are presented as mean ± SD. **q** Comparison of compartment size of *HA-dpp/+*, *nub >* + control wing pouch (*n* = 28) and *HA-dpp/+*, *nub > Dpp trap* wing pouch (*n* = 47). Data are presented as mean ± SD. Two-sided Mann−Whitney test was used for comparison of the A compartment (*p* < 0.0001). Two-sided unpaired Student's *t* test with unequal variance was used for comparison of the P compartment (*p* < 0.0001). ****$p$ < 0.0001. **r**, **s** Adult wing of *HA-dpp/+*, *nub >* + control wing disc (**r**) and *HA-dpp/+*, *nub > Dpp trap* wing disc (**s**). **t** Comparison of compartment size of (**r**, **s**). *HA-dpp/+*, *nub >* + control adult wing (*n* = 20) and *HA-dpp/+*, *nub > Dpp trap* adult wing (*n* = 20). Data are presented as mean ± SD. Two-sided unpaired Student's *t* test with unequal variance was used for comparison of the A compartment (*p* < 0.0001) and for comparison of the P compartment (*p* < 0.0001). ****$p$ < 0.0001. **u** Comparison of normalized compartment size of wing pouch upon HA trap (*n* = 38) and Dpp trap (*n* = 47) expression using *nub-Gal4* (the same data set from Figs. 3q and 5q). Data are presented as mean ± SD. Two-sided Mann−Whitney test was used for comparison of the A compartment (*p* < 0.0001). Two-sided unpaired Student's *t* test with unequal variance was used for comparison of the P compartment (*p* < 0.0001). ****$p$ < 0.0001. Scale bar 50 μm.

u and Supplementary Fig. 2u). To test if the difference could be due to more efficient blocking of Dpp dispersal by Dpp trap than by HA trap, each trap was expressed in the anterior stripe of cells using *ptc-Gal4* and posterior pMad signal was analyzed, since the extracellular staining was not sensitive enough to detect significant differences in leakage (Figs. 2e and 5b), and the posterior pMad activation would reflect the amount of leaked Dpp since the two traps were specifically expressed in the anterior stripe of cells. We found that HA trap blocked posterior pMad signal more efficiently than Dpp trap (Supplementary Fig. 10), indicating that Dpp trap actually blocks Dpp dispersal less efficiently than HA trap. Thus, the severe phenotypes caused by Dpp trap are likely because Dpp trap blocks Dpp signaling more efficiently than HA trap.

Interestingly, despite the slight leakage of Dpp from the Dpp trap (Supplementary Fig. 10), anterior Dpp trap expression caused more severe posterior growth defects than HA trap (Fig. 5j and Supplementary Fig. 2u), indicating that anterior Dpp signaling is non-autonomously required for the posterior growth. We note that, even though the anterior Dpp signaling was eliminated, genetic removal of *tkv* from the A compartment using *ci-Gal4* did not interfere with posterior growth as severe as Dpp trap (Supplementary Fig. 5c, d), probably because Dpp secreted from the A compartment can disperse to control posterior growth. Taken together, these results suggest that Dpp signaling in the source cells is required for a majority of patterning and growth seen upon HA trap expression.

**Rescue of *dpp* mutants by cell-autonomous Dpp signaling**. Our results so far suggest that, while the requirement for Dpp dispersal is relatively minor and asymmetric, Dpp signaling in the source cells is critically required for the majority of patterning and growth seen upon blocking Dpp dispersal. This raises the question of how cell-autonomous Dpp signaling in the source cells can control patterning and growth outside the anterior stripe of cells, the main *dpp* source cells. There are a couple of possible scenarios; if *dpp* expression is successively restricted to the anterior stripe of cells during development, the anterior stripe of cells may retain and deliver the earlier Dpp signaling to the peripheral region after they leave from the anterior stripe of cells via proliferation[59], or downstream factor(s) of Dpp signaling in the anterior stripe of cells may act non-autonomously to control patterning and growth outside the stripe of cells. Alternatively, *dpp* expression may not be restricted to the anterior stripe of cells in the early stages, similar to what has been shown in the case of *wg*[12].

Before we address these possibilities, we first asked how important cell-autonomous Dpp signaling in the source cells is for wing pouch patterning and growth. If the relatively mild phenotypes caused by HA trap are due to cell-autonomous Dpp signaling in the source cells, a constitutively active version of Tkv (TkvQD)[50] expressed in the anterior stripe of cells using *dpp-Gal4* should rescue severe patterning and growth defects in *dpp* mutants (Fig. 6a, c) to an extent mimicking the phenotypes caused by HA trap (Fig. 3). Indeed, under this condition, pMad activation in the anterior stripe of cells rescued nested Sal and Omb expression in the A compartment (Fig. 6b). Interestingly, growth, but not patterning, was also partially rescued in the P compartment as indicated by DSRF and *5xQE.DsRed* expression (Fig. 6b, arrow), thus indeed mimicking phenotypes caused by HA trap (Fig. 3). Unfortunately, the resulting adult flies were not recovered at 25 °C. However, when the temperature was shifted from 25 °C to 18 °C during mid- to late-third instar stages in order to reduce Gal4 activity during pupal stages, rare survivors were recovered from the pupal cases or managed to hatch

although they died shortly after hatching. In such survivors, although the anterior wing veins tends to be affected, probably due to continuous TkvQD expression during pupal stages even in the lower temperature, the anterior growth was rescued more than the posterior growth, similar to phenotypes caused by HA trap (Fig. 6d). These results suggest that the phenotypes caused by HA trap expression largely depend on cell-autonomous Dpp signaling in the source cells.

How can cell-autonomous Dpp signaling in the source cells control posterior growth if *dpp* expression is restricted to the A compartment? One trivial possibility is that the posterior growth was induced by non-specific *dpp-Gal4* expression in the P compartment. To test this, *dpp-Gal4* was converted into a ubiquitous *LexA* driver to express TkvQD permanently in lineage of *dpp-Gal4* (Fig. 6e). In this setup, pMad was constitutively activated in all the cells where *dpp-Gal4* has been expressed, including cells expressing *dpp-Gal4* non-specifically. Under this condition, pMad, Sal, and Omb were uniformly upregulated in the A compartment, and Brk was completely lost in the entire A compartment (Fig. 6f), indicating that *dpp-Gal4* has been expressed in the entire A compartment[60]. In contrast, pMad was not activated in the P compartment, but *5xQE.DsRed* was still induced in the P compartment (Fig. 6f, arrow), indicating that non-autonomous posterior growth control by anterior Dpp signaling is permissive rather than instructive (see "Discussion").

**Initial uniform *dpp* transcription in the anterior compartment**. Next, we asked how cell-autonomous Dpp signaling in the source cells can control anterior patterning and growth. The uniform lineage of Dpp-producing cells in the A compartment[60] (Fig. 6f) suggests two possibilities; either *dpp* expression is always restricted to the anterior stripe of cells but the lineage of these cells can cover the lateral region via proliferation[59], or earlier *dpp* expression covers the entire A compartment as in the case of *wg*[12]. Since the existing *dpp-Gal4* line is derived from a fragment of the *dpp* disc enhancer inserted outside the *dpp* locus, we first generated an endogenous *dpp-Gal4* line using our platform (Fig. 7a). We traced its lineage with G-TRACE analysis, in which RFP expression labels the real-time Gal4-expressing cells and GFP expression labels the entire lineage of the Gal4-expressing cells[60] (Fig. 7b). We found that the lineage of Dpp-producing cells indeed covers the entire A compartment (Fig. 7c).

To distinguish between the two possibilities mentioned above, we then generated a *dpp* transcription reporter line by inserting a destabilized GFP (half-life <2 h) into the *dpp* locus (Fig. 7d). Consistent with the latter possibility, we found that this *dpp* transcription reporter was uniformly expressed in the entire A compartment until the early third instar stage (Fig. 7e, e′, f, f′) and refined to an anterior stripe expression during much of the third instar stage (Fig. 7g, g′, h, h′). To directly follow *dpp* transcription, we also performed smFISH using RNAscope technology to visualize *dpp* transcripts in situ. Consistent with the dynamic expression of the *dpp* transcription reporter, we found uniform anterior *dpp* transcription until the early third instar stage (Fig. 7i, i′, j, j′) and an anterior stripe of *dpp* transcription in the later stages (Fig. 7k, k′, l, l′). Despite the initial broad anterior *dpp* expression, we found that pMad signal is low in the middle of the wing disc and graded toward the lateral regions, similar to the pMad gradient in the later stages (Supplementary Fig. 11).

**Transient *dpp* source outside Sal domain is required for anterior patterning and growth**. The earlier anterior *dpp* source outside the anterior stripe of cells could provide a local *dpp* source to control anterior patterning and growth when Dpp dispersal is blocked. However, since *ptc-Gal4* is also initially expressed in the

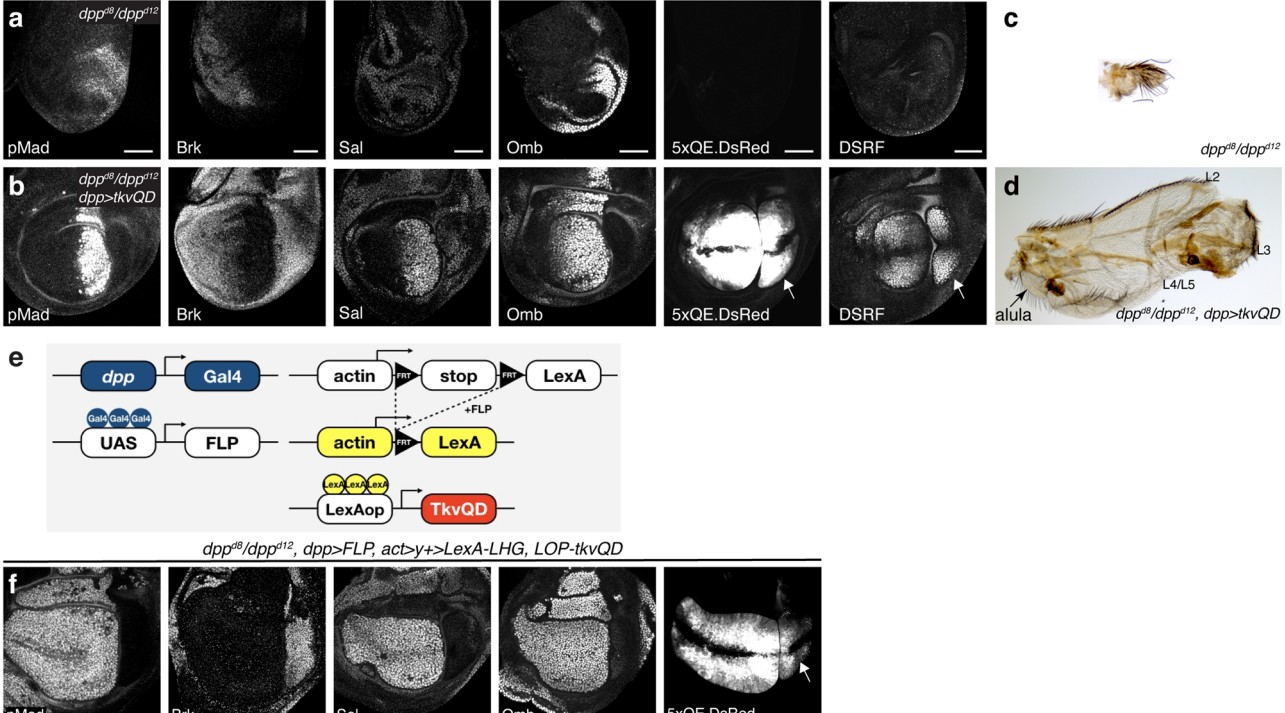

**Fig. 6 Rescue of *dpp* mutant by cell-autonomous Dpp signaling. a, b** α-pMad, α-Brk, α-Sal, α-Omb, 5xQE.DsRed, and DSRF staining of **a** *dpp^{d8}/dpp^{d12}* wing disc and **b** *dpp^{d8}/dpp^{d12}, dpp > tkvQD* wing disc. Arrows indicate rescued posterior wing pouch identified by 5xQE.DsRed and DSRF staining. **c** Adult wing of *dpp^{d8}/dpp^{d12}*. **d** Adult wing of *dpp^{d8}/dpp^{d12}, dpp > tkvQD*. **e** A schematic view of converting *dpp-Gal4* into a *LexA* driver, which is permanently expressed in lineage of *dpp-Gal4* expressing cells. In this experimental setup, the lineage of *dpp-Gal4* (including lineage of non-specific *dpp-Gal4* expression) will permanently activate TkvQD and thus pMad signaling. **f** α-pMad, α-Brk, α-Sal, α-Omb staining, and *5xQE.DsRed* signal of *dpp^{d8}/dpp^{d12}, dpp > FLP, act > y + >LexA-LHG, LOP-tkvQD* wing disc. Arrow indicates rescued posterior wing pouch identified by 5xQE.DsRed staining. Scale bar 50 μm.

entire A compartment[41], the relatively minor defects by HA trap could be due to the perdurance of Dpp signaling via artificially stabilized HA-Dpp by HA trap. To avoid HA trap expression in the entire A compartment, we applied *tubGal80ts* to express HA trap using *ptc-Gal4* at defined time points in the A compartment. To do so, the larvae were raised at 18 °C until a temperature shift to 29 °C to induce *Gal4* expression (Supplementary Fig. 12a). Upon HA trap expression from the mid-second instar stage, the lineage of *ptc-Gal4* covered at most the anterior Sal domain (see below), which corresponds to the region between L2 and L4 in the adult wing. We found that the later the temperature shift, the milder the posterior growth defects (Supplementary Fig. 12a, b). In contrast, the A compartment size (between L1 and L4) remained rather normal, independent of the timing of the temperature shift (Supplementary Fig. 12a, c). Interestingly, the size of peripheral regions (between L1 and L2) and the specification of L2 were not affected, independent of the timing of the temperature shift (Supplementary Fig. 12a, d). These results are consistent with a role of an anterior lateral *dpp* source for patterning and/or growth in the anterior lateral regions.

To directly test this, we applied Gal80ts to genetically remove *dpp* via *dpp^{FO}* allele upon FLP expression using *ptc-Gal4* from the mid-second instar stage. To do so, the larvae were raised at 18 °C for 5 days before a temperature shift to 29 °C and were then dissected 48 h later. In this setup, *dpp* was removed approximately from the anterior Sal region, where cells in which the FRT cassette was removed were marked by lacZ staining (Fig. 8a−f). Given that it takes about 20 h to eliminate the majority of Dpp protein under this condition[24,41], wing pouches are devoid of the majority of the Dpp protein derived from the anterior stripe of cells for 28 h at 29 °C until they reach the late third instar stage,

which corresponds to a lack of Dpp protein secreted from the main source from early third instar stages onward.

Under this setup, we found that pMad, Sal, and Omb were significantly reduced in the P compartment, consistent with the removal of the main *dpp* source (Fig. 8a−f). In contrast, in the A compartment, low levels of pMad persisted and Brk remained graded with lowest expression outside the lacZ positive region (Fig. 8d, arrow). As a consequence, while Sal was completely lost (Fig. 8e), weak Omb remained expressed in the A compartment (Fig. 8f). Consistent with a critical role of the *dpp* stripe for wing pouch growth[25,41,61], both anterior and posterior growth were affected (see below, Fig. 8k).

To test if the remaining anterior Dpp signaling activity is due to Dpp produced outside the anterior Sal domain, we then compared removal of *dpp* from the anterior Sal domain using *ptc-Gal4* (the same setup above) with removal of *dpp* from the entire A compartment using *ci-Gal4* from the mid-second instar stage. We found that the anterior weak Dpp signaling and Omb expression seen upon removal of *dpp* from the anterior Sal domain using *ptc-Gal4* (Fig. 8g, h) was completely lost by removing *dpp* from the entire A compartment using *ci-Gal4* (Fig. 8i, j). Furthermore, anterior growth defects upon removal of *dpp* from the anterior Sal domain was further enhanced upon removal of *dpp* from the entire A compartment (Fig. 8k), indicating that the anterior *dpp* source outside the anterior Sal domain is locally required for anterior Dpp signaling and growth.

How can the transient *dpp* transcription sustain Dpp target gene expression? One possibility is that persistent low pMad levels are continuously required to repress Brk. Alternatively, Brk repression by early Dpp signaling persists in the later stages without continuous Dpp signaling, for example, by epigenetic

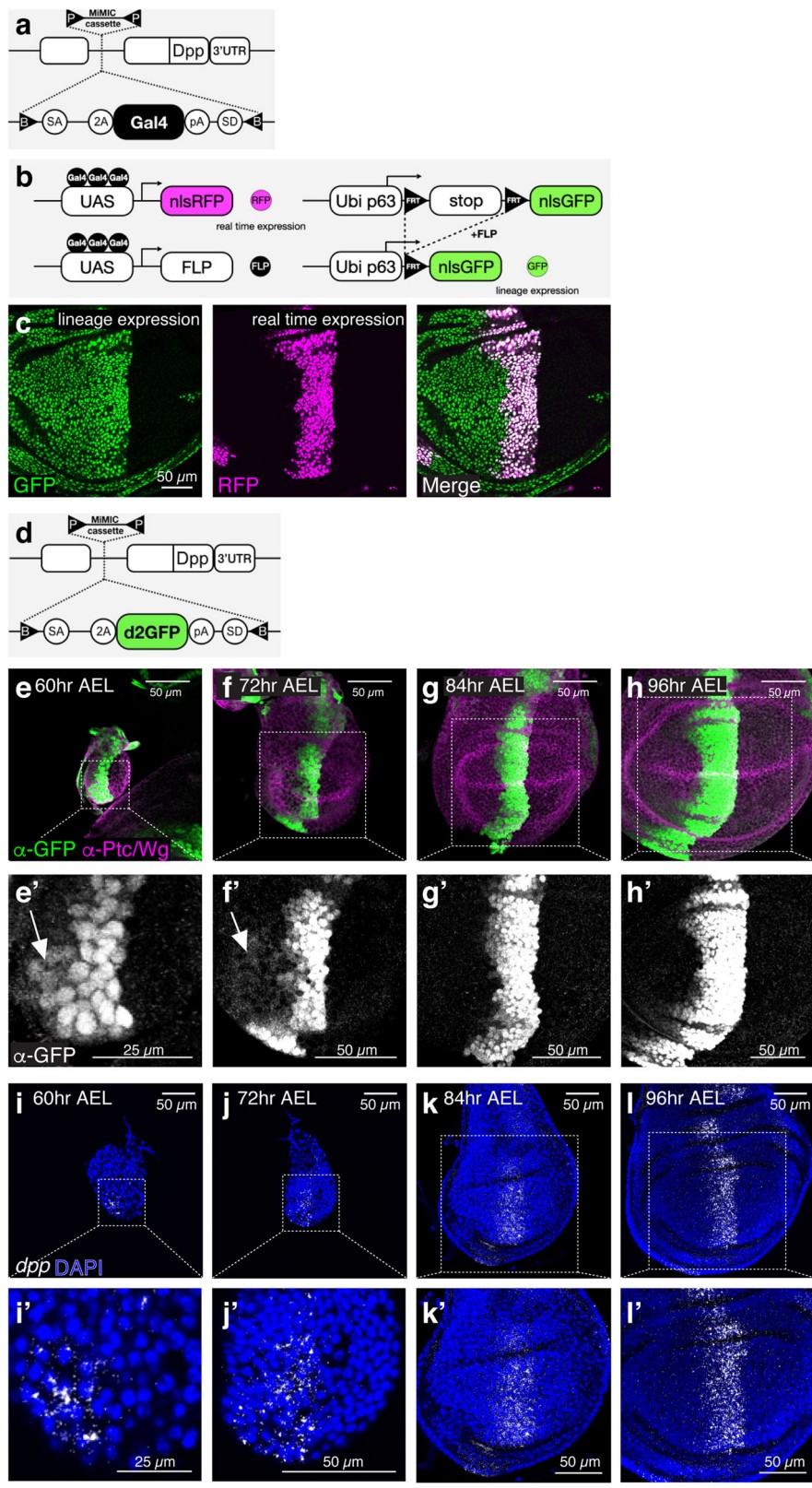

**Fig. 7 Initial uniform *dpp* transcription in the anterior compartment. a** A schematic view of a *Gal4* insertion into the *dpp* locus. **b** A schematic view of G-TRACE analysis. While RFP expression labels the real-time *Gal4*-expressing cells, GFP expression labels the lineage of the *Gal4*-expressing cells. **c** G-TRACE analysis of the endogenous *dpp-Gal4*. Scale bar 50 μm. **d** A schematic view of *d2GFP* insertion into the *dpp* locus. **e**–**h** α-GFP and α-Ptc/Wg staining of wing disc expressing the *d2GFP* reporter at mid-second instar stage (60 h AEL) (**e**), at early third instar stage (72 h AEL) (**f**), at mid-third instar stage (84 h AEL) (**g**), and at mid- to late- third instar stage (96 h AEL) (**h**). **e'**–**h'** Magnified wing discs from (**e**–**h**). Arrows indicate *dpp* transcription outside the stripe of cells. Scale bars as indicated. **i**–**l** smFISH against *dpp* using RNA scope technology. *yw* wing disc at 60 h AEL (**i**), 72 h AEL (**j**), 84 h AEL (**k**), 96 h AEL (**l**). **i'**–**l'** Magnified wing discs from (**i**–**l**). Scale bars as indicated.

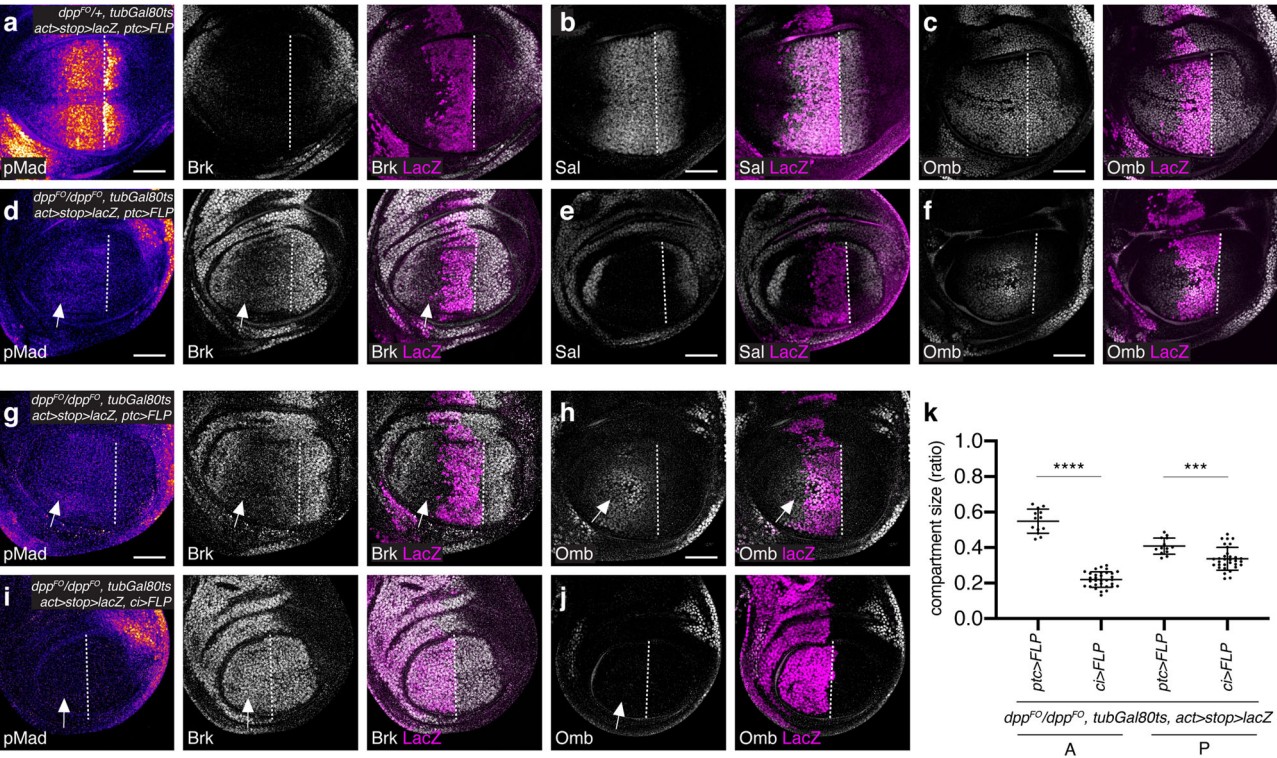

**Fig. 8 Transient anterior *dpp* source outside Sal domain is required for anterior patterning and growth. a–f** α-pMad, α-Brk, and α-LacZ (**a**, **d**), α-Sal and α-LacZ (**b**, **e**), α-Omb and α-LacZ (**c**, **f**) staining of *dpp^FO^/+, tubGal80ts, act > stop > lacZ, ptc > FLP* control wing disc (**a–c**) and *dpp^FO^/dpp^FO^, tubGal80ts, act > stop > lacZ, ptc > FLP* wing disc (**d–f**). Crosses were shifted from 18 °C to 29 °C at 5-day AEL (mid-second instar). α-LacZ staining marks the region where *dpp* is removed upon FLP expression. Dashed white lines mark the A–P compartment border. Scale bar 50 μm. **g–j** α-pMad, α-Brk, and α-LacZ (**g**, **i**), α-Omb and α-LacZ (**h**, **j**) staining of *dpp^FO^/dpp^FO^, tubGal80ts, act > stop > lacZ, ptc > FLP* wing disc (**g**, **h**) and *dpp^FO^/dpp^FO^, tubGal80ts, act > stop > lacZ, ci > FLP* (**i**, **j**). The genotypes of the wing discs in (**d–f**) and in (**g**, **h**) are identical. Crosses were shifted from 18 °C to 29 °C at 5-day AEL (mid-second instar). α-LacZ staining marks the region where *dpp* is removed upon FLP expression. Dashed white lines mark the A–P compartment border. Scale bar 50 μm. **k** Comparison of each compartment size of wing discs (**g–j**). *dpp^FO^/dpp^FO^, tubGal80ts, act > stop > lacZ, ptc > FLP* wing disc (n = 13) and *dpp^FO^/dpp^FO^, tubGal80ts, act > stop > lacZ, ci > FLP* wing disc (n = 32). Data are presented as mean ± SD. Two-sided unpaired Student's *t* test with unequal variance was used for comparison of the A compartment size (p < 0.0001) and for comparison of the P compartment size (p = 0.0002). ***p < 0.001, ****p < 0.0001.

regulation or via autoregulation. To distinguish between these possibilities, we applied *tubGal80ts* to genetically remove *tkv* via *tkvHA^FO^* allele upon FLP expression from the entire A compartment using *ci-Gal4* at different time points. To do so, the larvae were raised at 18 °C until a temperature shift to 29 °C to induce Gal4 expression. Consistent with a role of *tkv* in wing pouch growth, the earlier *tkv* was removed, the smaller the A compartment was (Supplementary Fig. 13). We found that Brk is largely derepressed as early as 16 h after *tkv* was removed (Supplementary Fig. 13). By considering perdurance activity of Gal80ts for 6 h after temperature shift[62], we can estimate that Brk is derepressed within 10 h at 29 °C after Dpp signaling was lost. Given that the transient *dpp* transcription, which terminated in the early third instar stage, can sustain the anterior pMad signaling and Brk repression at least 28 h at 29 °C after Dpp protein from the main source is eliminated, these results suggest that persistent weak pMad signaling is continuously required to repress Brk.

Taken together, Dpp dispersal-independent anterior patterning and growth can therefore be achieved by a combination of a persistent weak signaling by transient *dpp* transcription outside the stripe and a stronger signaling by continuous *dpp* transcription in the anterior stripe of cells.

## Discussion

It has long been thought that Dpp dispersal from the anterior stripe of cells generates the morphogen gradient in both compartments to control overall wing patterning and growth mainly based on the complete loss of wing tissue in *dpp* disc mutant alleles. Here, we generated two protein binder tools, namely HA trap and Dpp trap, to manipulate distinct parameters of the Dpp morphogen to determine the requirement of Dpp dispersal and cell-autonomous signaling in the source cells. We show that, although endogenous Dpp indeed generates a gradient in both compartments, requirement of Dpp dispersal for wing patterning and growth is much less than previously thought (Fig. 9).

**New protein binder tools manipulating distinct aspects of Dpp.** Although nanobodies against GFP have been used most intensively in the field[63,64], fusion to GFP could affect protein functions, as is the case for GFP-Dpp. To bypass this, we generated HA trap, which is analogous to morphotrap and provides an alternative way to trap secreted proteins. Although HA trap can trap HA-Dpp as efficient as morphotrap, we found several differences between the two. First, while trapping GFP-Dpp by morphotrap in the source cells activates Dpp signaling in at least one cell row in the P compartment[37], trapping HA-Dpp by HA trap did not (Fig. 3b). We think that the difference is not because GFP-Dpp trapped by morphotrap activates Dpp signaling in trans, since clonal accumulation of GFP-Dpp by morphotrap in the P compartment failed to do so[37]. Second, while morphotrap could trap GFP-Dpp even in the peripheral regions[37], HA trap did not (Fig. 2l). Third, while trapping GFP-Dpp by morphotrap in the source cells induced excessive Dpp signaling in the source cells and caused severe defects in the adult wing[37], trapping HA-

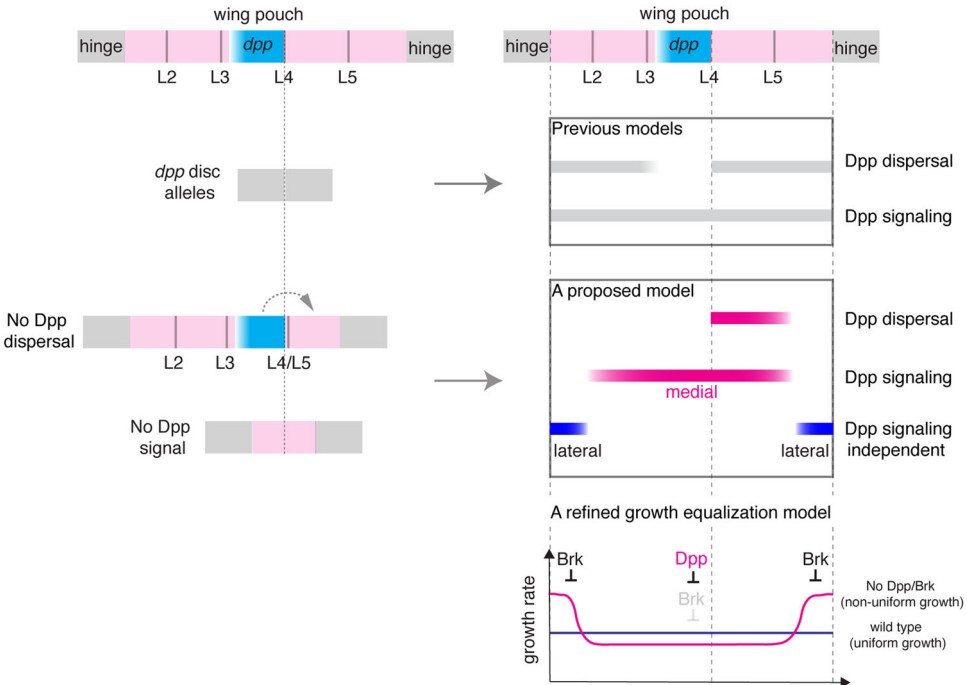

**Fig. 9 Asymmetric requirement of Dpp dispersal for wing patterning and growth, and a refined growth equalization model.** Based on complete lack of wing pouch in *dpp* disc alleles, it has long been thought that Dpp dispersal from the anterior stripe of cells controls the overall patterning and growth of the Drosophila wing pouch. In contrast, the present study shows that while critical for the posterior patterning and growth, Dpp dispersal is largely dispensable for the anterior patterning and growth. The asymmetric requirement of Dpp dispersal is in part due to *dpp* transcriptional refinement from an initially uniform to a localized expression and persistent signaling in transient *dpp* source cells. Furthermore, despite a critical requirement of Dpp signaling for the medial wing pouch growth, Dpp signaling is dispensable for the lateral wing pouch growth after wing pouch specification. We propose a refined growth equalization model, in which Dpp signaling removes a growth repressor Brk to allow medial regions to grow, while Dpp signaling sets Brk expression to the lateral region to repress the lateral growth with higher growth rate to equalize growth rates. In this refined model, both the Dpp signaling-dependent medial region and -independent lateral region exist within the wing pouch.

Dpp by HA trap in the source cells slightly reduced Dpp signaling in the source cells and caused relatively minor defects in the adult wing (Fig. 3 and Supplementary Fig. 2a−j). We speculate that these differences are in part due to overexpression of GFP-Dpp. Due to excess amount of GFP-Dpp, some GFP-Dpp may leak from the morphotrap to activate Dpp signaling in the P compartment, or may reach to the peripheral regions. Excessive Dpp signaling may cause cell death during pupal stages to cause severe adult wing defects as previously shown[65]. These differences highlight the importance of investigating endogenous protein functions.

In addition to creating, generating, HA trap, we isolated DARPins against Dpp, and generated a Dpp trap analogous to HA trap. Interestingly, while HA trap blocks mainly Dpp dispersal (Fig. 3), we found that Dpp trap blocks Dpp dispersal and cell-autonomous signaling in the source cells (Fig. 5). We speculate that HA trap binds to the HA tag and thereby allows Dpp to bind to its receptors, while Dpp trap directly binds to Dpp to block its interaction with the receptors. Regardless of the actual mechanisms underlying this difference, these tools allowed us to dissect the requirements of dispersal and cell-autonomous signaling in the source cells for wing pouch growth and patterning. Relatively mild phenotypes by HA trap and severe phenotypes by Dpp trap indicate a minor and asymmetric role of Dpp dispersal and a critical role of cell-autonomous Dpp signaling in the source cells, respectively. Furthermore, these results also suggest that the severe *dpp* mutant phenotypes do not reflect the role of Dpp dispersal alone, but reflect the role of both Dpp dispersal and cell-autonomous Dpp signaling in the source cells, with more contribution from the latter parameter.

**Asymmetric requirement of Dpp dispersal**. The relatively minor requirement of Dpp dispersal for the anterior compartment is reminiscent of the minor requirement of dispersal of Wg[12]. Wg dispersal is largely dispensable for the wing growth, likely due to the early uniform *wg* transcription in the entire wing pouch and a memory of the earlier signaling[12]. In contrast, the requirement of Dpp dispersal is asymmetric along the A−P axis due to the early uniform *dpp* transcription in the A compartment. In both cases, transcriptional refinement of each morphogen and persistent signaling by transient morphogen expression appear to be a key for robustness against the absence of dispersal of each morphogen.

Although not identified, a *dpp* source outside the anterior stripe of cells has previously been implicated to control the entire wing pouch growth based on the minor growth defects by removal of *dpp* from the anterior stripe of cells using *dpp-Gal4* and severe growth defects by removal of *dpp* from the A compartment using *ci-Gal4*[24]. However, we and others showed that removal of *dpp* using *dpp-Gal4* was imprecise and inefficient, and that *dpp* derived from the anterior stripe of cells is indeed critical for wing growth[25,41,61]. Thus, the presence of a *dpp* source outside the anterior stripe of cells has been questioned. Our results suggest that such a *dpp* source indeed exists and contributes to the anterior patterning and growth, but not to the growth of the entire wing pouch (Figs. 7, 8).

It remains unknown how transient *dpp* expression can maintain Dpp signaling in the anterior lateral region (Fig. 8). One possibility is that feedback factors control the duration of Dpp signaling. Various feedback factors have been shown to regulate Dpp signaling. For example, the *Drosophila* tumor

necrosis factor α homolog Eiger and a secreted BMP-binding protein Crossveinless-2 (Cv-2) are positively regulated by Dpp signaling to either positively and negatively influence Dpp signaling in the early embryo, respectively[66]. While JNK is not activated by Eiger in the wild-type wing pouch[67], Cv-2 acts as a positive feedback factor to specify the posterior crossvein during pupal stages[68]. Whether these two factors act during wing pouch patterning and growth remains to be addressed. Pentagon (Pent) is a secreted feedback factor repressed by Dpp signaling to positively regulate Dpp signaling in the wing disc[69]. Pent is produced in the lateral region of the wing pouch and regulates Dpp signaling and proliferation there. Thus, in addition to its role for scaling, Pent may control the duration of Dpp signaling in the lateral region. Another possibility is that, since it takes quite some time (20−24 h at 29 °C) to eliminate Dpp protein upon excision of *dpp*[24,41], relatively stable *dpp* mRNA and/or protein after the termination of transient *dpp* transcription may contribute to Brk repression (~28 h at 29 °C).

**Growth without Dpp dispersal and signaling**. Despite a critical requirement of *dpp* for the entire wing pouch growth, our results uncover Dpp dispersal- and signaling-independent lateral wing pouch growth (Figs. 3, 4, 6 and Supplementary Figs. 5, 6) The presence of Dpp signaling-independent growth appears inconsistent with the fact that the wing pouch is completely lost in *dpp* mutants (Fig. 6a, c and Supplementary Fig. 7b). This could be in part due to a failure of the initial specification of the wing pouch in *dpp* disc alleles[52]. Indeed, despite severe growth defects, part of the wing pouch could still grow upon removal of *dpp* after wing pouch specification (Fig. 4e−h) and upon removal of *tkv* after wing pouch specification (Supplementary Fig. 5).

Consistent with the presence of Dpp signaling-independent lateral growth, it has been shown that lateral wing fates are less sensitive than medial wing fates in various *dpp* mutant alleles[70]. However, this appears counter-intuitive since the lateral region, where morphogen level are low, is expected to be more sensitive than the medial region to a reduction of morphogen levels. It has been proposed that another BMP-type ligand, Glass bottom boat (Gbb), which is expressed ubiquitously in the wing pouch, contributes to lateral cell fates[70]. However, since Gbb signaling is also mediated by Tkv[71] but the lateral growth is independent of Tkv (Figs. 3, 4, 6 and Supplementary Figs. 5, 6), we think that the lateral region develops independent of direct Dpp and Gbb signaling. Thus, the lateral region is less sensitive than the medial region in various *dpp* mutant alleles probably because the lateral region can develop independent of Dpp signaling. What regulates the Dpp signaling-independent lateral wing pouch growth? Given that *5xQE.DsRed* is also dependent on Wg[45,54], Wg may regulate *5xQE.DsRed* expression and growth in the absence of Dpp signaling.

In addition to the Dpp signaling-independent lateral growth, requirement of anterior Dpp signaling for posterior growth (Fig. 5j and Supplementary Fig. 2u) and rescue of posterior growth in *dpp* mutants by anterior Dpp signaling (Fig. 6) indicates that anterior Dpp signaling is non-autonomously involved in posterior lateral growth. We note that similar rescue of posterior growth in *dpp* mutant by anterior Dpp signaling has previously been recognized, but the rescued posterior growth was interpreted as growth of the hinge region, without immunostainings for relevant markers[59]. It remains unknown how anterior Dpp signaling contributes to posterior growth. One possibility is that factors from the A compartment may act non-autonomously to promote posterior growth. Such factors include, but are not limited to, direct downstream factors of Dpp signaling. Given that *5xQE.DsRed* is dependent on Wg[45,54], Wg derived from the

rescued A compartment may regulate *5xQE.DsRed* expression and growth in the P compartment. Alternatively, mechanical forces may be involved in the non-autonomous growth. It has been proposed that growth factors such as Dpp induce medial growth and subsequently stretch the peripheral regions to induce lateral growth. As the wing disc grows, the peripheral regions in turn compresses the medial region of the wing disc to inhibit the growth of the medial region[72–76]. Thus, the growth of the A compartment may stretch the P compartment cells to stimulate their proliferation. It has also been shown that juxtaposition of cells with different Dpp signaling level can induce proliferation non-autonomously but the growth is transient[26,27]. Therefore, we think it is unlikely that the difference of Dpp signaling levels between two compartments can induce sustained growth.

**A refined growth model**. The presence of Dpp signaling-independent lateral wing pouch growth is at odds with all the growth models assuming that Dpp dispersal directly controls overall wing patterning and growth based on the complete loss of wing tissue in *dpp* mutants[24–33] (Fig. 9). For example, no wing pouch growth is expected without Dpp signaling due to a lack of either a temporal increase of Dpp signaling (temporal model)[32], a detectable Dpp signal (threshold model)[25,61,77], or a slope of Dpp signaling activity (gradient model)[26,27].

It has recently been proposed that Wg and Dpp control wing pouch size by two distinct mechanisms[33,45,54,78]. One is an intracellular mechanism, in which Vg controls its own expression through QE in response to Wg and Dpp; the other one is an intercellular mechanism, in which Vg-positive wing pouch cells send a feed-forward signal to induce QE-dependent *vg* expression in the neighboring pre-wing cells to recruit them into the wing pouch in response to Dpp and Wg. The dispersal of each morphogen is critical for the two mechanisms in the model[33,45,54,78]. In the genetic setup used, in which morphogens and *vg* expression are eliminated, the two mechanisms appear to recapitulate the dynamics of *vg* expression seen during normal wing disc development, in which *vg* expression is gradually expanded in the wing pouch area. However, although sufficient, these mechanisms do not necessarily account for *vg* expression under physiological conditions. Indeed, we found Dpp dispersal and/or signaling-independent *vg* expression in various conditions.

Among the models, the presence of Dpp signaling-independent lateral wing pouch growth appears most consistent with the growth equalization model, in which Dpp signaling removes Brk to allow medial regions to grow, while Dpp signaling limits Brk expression to lateral regions to suppress Dpp signaling-independent lateral growth with higher proliferation nature to equalize the non-uniform growth[22,28] (Fig. 9). However, the identity of medial and lateral regions remained undefined in this model. Given the complete lack of wing pouch in *dpp* mutants, it is tempting to speculate that the Dpp signaling-dependent medial region corresponds to the entire wing pouch region, and the Dpp signaling-independent lateral region corresponds to the hinge region located next to the wing pouch region. Indeed, using morphotrap, such a Dpp signaling-insensitive posterior growth has previously been observed and interpreted as the growth of the hinge region due to severe adult wing defects[37]. However, in contradiction to this interpretation, which would predict overgrowth of the hinge region in *brk* mutant, a massive overgrowth of wing pouch region was observed in *brk* mutant (Supplementary Fig. 7). Based on our results that the lateral wing pouch regions can grow independent of direct Dpp signaling (Figs. 3, 4, 6 and Supplementary Figs. 5, 6), we therefore suggest to refine the growth equalization model and propose that both Dpp signaling-dependent medial and Dpp signaling-independent lateral regions

are located within the wing pouch (Fig. 9). The permissive role of Dpp signaling in modulating a non-uniform growth potential within the wing pouch raises questions about what kind of instructive signals control proliferation and growth, and how the non-uniform growth potential emerges within the wing pouch independent of the Dpp/Brk system[44].

In summary, our approach applying customized protein binder tools to manipulate distinct parameters of Dpp challenges the long-standing dogma that Dpp dispersal controls overall wing patterning and growth. Given that the tools developed in this study are easily applicable in other tissues, it would be interesting to investigate the precise requirement of Dpp dispersal and signaling in other tissues or between different organs.

## Methods

**Data reporting**. No statistical methods were used to predetermine sample size. The experiments were not randomized, and investigators were not blinded to allocation during experiments and outcome assessment.

**Fly stocks**. Flies were kept in standard fly vials (containing polenta and yeast) in a 25 °C incubator. The following fly lines were used: $dpp^{FO}$, $dpp$-Gal4, UAS-FLP (Matthew Gibson), $ptc$-Gal4 (BL2017), P{act5C(FRT.polyA)lacZ.nls1}3, ry506 (BL6355), w[*]; P{w[+mC]=UAS-RedStinger}6, P{w[+mC]=UAS-FLP.Exel}3, P{w[+mC]=Ubi-p63E(FRT.STOP)Stinger}15F2 (G-TRACE)(BL28281), $brk^{XA}$ (BL58792), $dpp^{MI03752}$ (BL36399), PBac{RB}e00178, Dp(2;2)DTD48 (Bloomington Stock Center). $omb$-LacZ (Kyoto101157). $act > Stop$, $y + >LexA^{LHG}$, $tkv^{a12}$, UAS-TkvQD, pLexAop-TkvQD (Konrad Basler), 5xQE.DsRed (Gary Struhl), UAS/Lex-Aop-HA trap (this study), UAS/LexAop-Dpp trap (F1) (this study), UAS/LexAop-Dpp trap (C9) (this study), $dpp^{d8}$, $dpp^{d12}$, $nub$-Gal4 (II), $ci$-Gal4 (II), $hh$-Gal4 (III), UAS-p35(III), tub-Gal80ts (III) are described from Flybase. $tub > CD2$, Stop > Gal4, UAS-nlacZ (Francesca Pignoni). TkvHA (Giorgos Pyrowolakis).

**Genotypes by figures**. Each genotype by figures was provided as a Supplementary Table 1.

**Immunostainings and antibodies**. Staged larvae were dissected and transferred directly to cold fixative (4% PFA in PBS) and fixed for 20 min at room temperature. After fixation, discs were extensively washed with PBT (PBS plus 0.3% Triton-X) and blocked in PBT plus 5% normal goat serum (Sigma-Aldrich) for >30 min at 4 °C, followed by incubation with primary antibody overnight at 4 °C. The next day discs were washed in PBT and then incubated in secondary antibody in PBT plus 5% normal goat serum for 2 h at room temperature on a rotor without light. After another round of washes with PBT, samples were mounted in Vectashield (H-1000, Vector Laboratories)[37]. For extracellular staining, dissected larvae were incubated with primary antibodies in M3 medium for 1 h on ice before fixation to allow antibodies to access only the extracellular antigens. Each fly cross was set up together with a proper control and genotypes were processed in parallel. If the genotype could be distinguished, experimental and control samples were processed in the same tube. To minimize variations, embryos were staged by collecting eggs for 2–4 h. An average intensity image from three sequential images from a representative wing disc is shown for all the experiments. Images of wing discs were obtained using a Leica SP5-II-MATRIX confocal microscope (section thickness 1 μm) and Leica LAS AF (ver. 2.6.0.7266). Images were analyzed using ImageJ (v.2.0.0-rc69/1.52p). Figures were prepared using Omero (ver5.9.1) and Illustrator (24.1.3).

The following primary antibodies were used: anti-HA (3F10, Roche, 11867423001; 1:300 for conventional staining, 1:20 for extracellular staining), anti-Ollas (L2, Novus Biologicals, NBP1-06713; 1:300 for conventional signaling, 1:20 for extracellular staining), anti-phospho-Smad1/5 (41D10, Cell Signaling, #9516; 1:200), anti-Brk (obtained from Gines Morata; 1:1000), anti-Sal (obtained from Rosa Barrio; 1:500), anti-Omb (obtained from Gert Pflugfelder; 1:500), anti-Wg (4D4, DSHB, University of Iowa; 1:120), anti-Ptc (DSHB, University of Iowa; 1:40), anti-β-Galactosidase (Z3781, Promega; 1:1000), anti-β-Galactosidase (ab9361, abcam; 1:1000), anti-Cleaved Caspase-3 (#9661, Cell Signaling; 1:500). The following secondary antibodies were used at 1:500 dilutions in this study. Goat anti-chicken IgY (H + L) DyLight 680(#SA5-10074, Invitrogen), Alexa Fluor 488 AffiniPure goat anti-mouse IgG, Fcγ fragment specific (115-545-071, Jackson ImmunoResearch), goat anti-mouse IgG (H + L) Alexa Fluor Plus 488 (A32723, Thermo Fisher), goat anti-mouse IgG (H + L) Alexa Fluor 568 (#A-11004, Thermo Fisher), Alexa Fluor 680 AffiniPure goat anti-mouse IgG, Fcγ fragment specific (115-625-071, Jackson ImmunoResearch), goat anti-rabbit IgG (H + L) Alexa Fluor 488 (#A-11008, Thermo Fisher), F(ab')2-goat anti-rabbit IgG (H + L) Alexa Fluor 568 (#A-21069, Thermo Fisher), goat anti-rabbit IgG (H + L) Alexa Fluor 680 (# A-21109, Thermo Fisher), goat anti-guinea pig IgG (H + L) Alexa Fluor 488 (#A-11073, Thermo Fisher), goat anti-rat IgG Fc (FITC) (ab97089, Abcam), goat anti-rat IgG (H + L), Alexa Fluor 680 (#A-21096, Thermo Fisher).

**RNAscope**. smFISH using RNAscope technology with the probe against $dpp$ (Cat No. 896761) has previously been successful to visualize $dpp$ mRNA in the germline stem cell niche[79]. The accession number for the probes against $dpp$ target 682−1673 is NM_057963.5. RNAscope was performed in an Eppendorf tube. Larvae were dissected in PBS and fixed in 4% paraformaldehyde in PBS for 30 min. Fixed larvae were washed with PBS and then with PBT (PBS containing 0.03% Triton X-100). Fixed larvae were dehydrated at RT (room temperature) in a series of 25, 50, 75 and 100% methanol in PBT, and stored overnight in 100% methanol at −20 °C. Larvae were rehydrated at RT in a series of 75, 50, 25, 0% methanol in PBT. Larvae were treated with protease using Pretreat 3 (RNAscope H2O2 & Protease Plus Reagents; ACD, 322330) at RT for 5 min. After washing with PBT, hybridization using probes against $dpp$ was performed overnight at 40 °C. The following day, larvae were washed with RNAscope wash buffer and re-fixed with 4% paraformaldehyde in PBS for 10 min. After washing with PBS, fluorescent signal was developed using RNAscope Fluorescent Multiplex Reagent Kit according to the manufacturer's instructions. Wing discs were mounted in Vectashield (H-1000, Vector Laboratories) and dissected for imaging. Images of wing discs were obtained using a Leica SP5-II-MATRIX confocal microscope (section thickness 1 μm) and Leica LAS AF (ver. 2.6.0.7266). Images were analyzed using ImageJ (v.2.0.0-rc69/1.52p). Figures were prepared using Omero (ver5.9.1) and Illustrator (24.1.3).

## Quantification

*Quantification of pMad, Brk, Sal, and Omb.* From each z-stack image, signal intensity profile along A/P axis was extracted from average projection of three sequential images using ImageJ (v.2.0.0-rc69/1.52p). Each signal intensity profile collected in Excel (Ver. 16.51) was aligned along A/P compartment boundary (based on anti-Ptc staining) and average signal intensity profile from different samples was generated and plotted by the script (wing_disc-alignment.py). The average intensity profile from control and experimental samples was then compared by the script (wingdisc_comparison.py). Both scripts can be obtained from https://etiennees.github.io/Wing_disc-alignment/. The resulting signal intensity profiles (mean with SD) were generated by Prism (v.8.4.3(471)). Figures were prepared using Omero (ver5.9.1) and Illustrator (24.1.3).

*Quantification of wing pouch size and adult wing size.* The A and P compartment of the wing pouches were approximated by Ptc/Wg staining and positions of folds, and the A/P compartment boundary of the adult wings were approximated by L4 position. The size of each compartment was measured using ImageJ (v.2.0.0-rc69/ 1.52p) and collected in Excel (Ver. 16.51). Scatter dot plots (mean with SD) were generated by Prism (v.8.4.3(471)). Figures were prepared using Omero (ver5.9.1) and Illustrator (24.1.3).

## Generation of *HA-dpp* and *GFP-dpp* knock-in allele

*Cloning of plasmids for injection.* A fragment containing multi-cloning sites (MCS) between two inverted attB sites was synthesized and inserted in the pBS (BamHI) vector (from Mario Metzler). A genomic fragment of $dpp$ between $dpp^{MI03752}$ and PBac{RB}e00178 (about 4.4 kb), as well as an FRT and 3xP3mCherry were inserted in this MCS by standard cloning procedures. A fragment encoding HA tag or GFP was inserted between the XhoI and NheI sites inserted after the last Furin processing site[18].

*Inserting* dpp *genomic fragments in the* dpp *locus (Fig. 1b (i)).* The resulting plasmids were injected in yw M{vas-int.Dm}zh-2A; $dpp^{MI03752}$/Cyo, P23. P23 is a transgene containing a $dpp$ genomic fragment to rescue $dpp$ haploinsufficiency. After the hatched flies were backcrossed, flies that lost $y$ inserted between inverted attP sites in the mimic transposon lines were individually backcrossed to establish each stock (yw M{vas-int.Dm}zh-2A; HA-dpp(w-)/Cyo, P23 and yw M{vas-int.Dm} zh-2A; GFP-dpp(w-)/Cyo, P23). The orientation of inserted fragments was determined by PCR using primers provided as Supplementary Table 2.

*Removal of the endogenous* dpp *exon by FLP/FRT recombination (Fig. 1b (ii)).* Males from the above stock were crossed with females of genotype hsFLP; al, Pbac{RB}e00178 /SM6, al, sp and subjected to heat-shock at 37 °C for 1 h/day until flies hatch. Pbac{RB}e00178 contains FRT sequence and w+. Upon recombination, the $dpp$ genomic fragment with a tag is followed by FRT and w+. In the case of HA-dpp, males of hsFLP; HA-dpp/al, PBac{RB}e00178 were recovered and crossed with yw; al, b, c, sp/ SM6, al, sp. From this cross, HA-dpp allele with a successful recombination (males of yw; HA-dpp(w+)/SM6, al, sp or yw; HA-dpp(w+)/al, b, c, sp) were screened for w+ and against al. Each male was then individually crossed with virgins of yw; al, b, c, sp/ SM6, al, sp to establish the yw; HA-dpp/SM6, al, sp stock. However, in the case of GFP-dpp, hsFLP; GFP-dpp/al, PBac{RB}e00178 flies were not recovered, indicating that GFP-dpp allele is haploinsufficient. Indeed, insertion of GFP-dpp in the $dpp$ locus and maintenance of the resulting stock (GFP-dpp(w-)/Cyo, P23) required P23, a transgene containing a $dpp$ genomic fragment to rescue $dpp$ haploinsufficiency. We thus concluded that GFP-dpp allele is haploinsufficient. Consistent with this, the original stock (GFP-dpp(w−)/Cyo, P23) never became homozygous.

**Construction of α-HA scFv**. The accession numbers for VH and VL of anti-HA antibody are VH: LC522514 and VL: LC522515. cDNA of HA scFv was constructed by combining coding sequences of variable regions of the heavy chain ($V_H$: 1−423 of LC522514) and of the light chain ($V_L$: 67−420 of LC522515) cloned from anti-HA hybridoma (clone 12CA5)[40] with a linker sequence (5′-accggtGGC GGAGGCTCTGGCGGAGGAGGTTCCGGCGGAGGTGGAAGCgatatc-3′) in the order of $V_H$-linker-$V_L$. The coding sequence of HA scFv was cloned into pCS2+mcs-2FT-T for FLAG-tagging. Requests for HA scFv should be addressed to Y.M. (mii@nibb.ac.jp). To generate HA trap, the region encoding morphotrap (VHH-GFP4) was replaced with *Kpn*I and *Sph*I sites in pLOTattB-VHH-GFP4:CD8-mChery[37]. A fragment encoding HA scFv was amplified by PCR and then inserted via *Kpn*I and *Sph*I sites by standard cloning procedures.

**Selection of Dpp-binding DARPins and generation of Dpp trap**. Streptavidin-binding peptide (SBP)-tagged mature C-terminal domain of Dpp was cloned into pRSFDuet vector by a standard cloning. Dpp was overexpressed in *E. coli*, extracted from inclusion bodies, refolded, and purified by heparin affinity chromatography followed by reverse phase HPLC[80]. To isolate suitable DARPins, SBP-tagged Dpp was immobilized on streptavidin magnetic beads and used as a target for DARPin selections by employing multiple rounds of Ribosome Display[81,82]. Due to the aggregation and precipitation propensity of the purified SBP-Dpp, the refolded dimers previously stored in 6 M urea buffer (6 M urea, 50 mM Tris-HCl, 2 mM EDTA pH8.0, 0.25 M NaCl) were diluted to a concentration of 100−120 μg/ml in the same buffer and subsequently dialyzed against 4 mM HCl at 4 °C overnight. To ensure binding of correctly folded Dpp to the beads, this solution was diluted five times in the used selection buffer just prior to bead loading and the start of the ribosome display selection. In each panning round, the target concentration presented on magnetic beads was reduced, while the washing stringency was simultaneously increased to enrich for binders with high affinities[81]. In addition, from the second round onward, a pre-panning against Streptavidin beads was performed prior to the real selection to reduce the amounts of matrix binders. After four rounds of selection, the enriched pool was cloned into an *E. coli* expression vector, enabling the production of both N-terminally $His_8$- and C-terminally FLAG-tagged DARPins. Nearly 400 colonies of transformed *E. coli* were picked and the encoded binders expressed in small scale. Bacterial crude extracts were subsequently used in enzyme-linked immunosorbent assay (ELISA) screenings, detecting the binding of candidate DARPins to streptavidin-immobilized Dpp, or just streptavidin (indicating background binding) by using a FLAG-tag based detection system. Of those 127 candidate DARPins interacting with streptavidin-immobilized Dpp, 73 (or 57%) specifically bound to Dpp (i.e., having at least threefold higher signal for streptavidin-immobilized Dpp than to streptavidin alone). Thirty-six of these (50%) revealed unique and full-length sequences. To generate Dpp trap, the region encoding morphotrap (VHH-GFP4) was replaced with *Kpn*I and *Sph*I sites in pLOTattB-VHH-GFP4:CD8-mChery[37]. Each fragment encoding a DARPin was amplified by PCR and then inserted via *Kpn*I and *Sph*I sites by standard cloning procedures.

**Generation of *tkvHA*$^{FO}$ (Flip-out) allele**. The *tkvHA* allele was previously described[83]. An FRT cassette was inserted in the re-insertion vector for *tkvHA* (Genewiz) and re-inserted into the attP site in the *tkv* locus.

**Generation of endogenous *dpp-Gal4***. pBS-KS-attB2-SA(1)-T2A-Gal4-Hsp70 (addgene 62897) was injected in the *yw M{vas-int.Dm}zh-2A; dpp*$^{MI03752}$/Cyo, P23 stock. Since the Gal4 insertion causes haploinsufficiency, the *dpp-Gal4* was recombined with *Dp(2;2)DTD48* (duplication of *dpp*) for G-TRACE analysis.

**Generation of an endogenous *dpp* reporter**. A DNA fragment containing T2A-d2GFP-NLS was synthesized and used to replace the region containing *T2A-Gal4* in pBS-KS-attB2-SA(1)-T2A-Gal4-Hsp70 via BamHI to generate pBS-KS-attB2-SA(1)-T2A-d2GFP-NLS-Hsp70 (Genewiz). The resulting plasmid was injected in the *yw M{vas-int.Dm}zh-2A; dpp*$^{MI03752}$/ Cyo, P23 stock.

**Statistics and reproducibility**. All images were obtained from multiple animals ($n > 3$). The experiments were repeated at least two times independently with similar results. Statistical significance was assessed by Prism (v.8.4.3(471)) based on the normality tests. The observed phenotypes were highly reproducible as indicated by the significance of *p* values obtained by statistical tests.

**Reporting summary**. Further information on research design is available in the Nature Research Reporting Summary linked to this article.

## Data availability
The data that support all experimental findings of this study are available within the paper and its Supplementary Information files or from the corresponding author S.M. upon request. Raw data necessary to reproduce all statistical analyses and results in the paper are provided in the source data file provided with this paper. The accession numbers for VH and VL of anti-HA antibody are VH: LC522514 and VL: LC522515.

The accession number for the probes against *dpp* target 682-1673 is NM_057963.5. Source data are provided with this paper.

## Code availability
Both wing_disc-alignment.py and wingdisc_comparison.py are available from (https://etiennees.github.io/Wing_disc-alignment/). Briefly, wing_disc-alignment.py was used to align each signal intensity profile along the A/P compartment boundary and generate average signal intensity profile from different samples. wingdisc_comparison.py was then used to compare the average intensity profile from control and experimental samples.

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

## Acknowledgements

We thank Konrad Basler, Giorgos Pyrowolakis, Gustavo Aguilar, and Sheida Hadji Rasouliha for comments on the manuscript. Stocks obtained from the Bloomington *Drosophila* Stock Center (NIH P40OD018537) were used in this study. We thank Konrad Basler, Gary Struhl, Matthew Gibson, Giorgos Pyrowolakis, and Kyoto Stock Center for flies. We thank Gines Morata, Rosa Barrio, Gert Pflugfelder, and the Developmental Studies Hybridoma Bank at The University of Iowa for antibodies. We thank the Biozentrum Imaging Core Facility for the maintenance of microscopes and support. We thank Etienne Schmelzer for scripts for quantification and Oguz Kanca for the idea to manipulate the *dpp* locus. We thank Dietmar Schreiner and Caroline Bornmann for introducing RNAscope and sharing reagents, and Minkyoung Lee for the advice for smFISH and sharing reagents. We would like to thank Bernadette Bruno, Gina Evora, and Karin Mauro for constant and reliable supply with world's best fly food. We further acknowledge all current and former members of the High-Throughput Binder Selection facility at the Department of Biochemistry of the University of Zurich for their contribution to the establishment of the semi-automated ribosome display that resulted in the generation of the used anti-Dpp DARPin binders, particularly Thomas Reinberg. S.M. has been supported by a JSPS Postdoctoral Fellowship for Research Abroad and the Research Fund Junior Researchers University of Basel, and is currently supported by an SNSF Ambizione grant (PZ00P3_180019). Y.M. has been supported by JST PRESTO (JPMJPR194B) and JSPS KAKENHI (18K14720). The work in the Affolter laboratory was supported by grants from Kantons Basel-Stadt and Basel-Land and from the SNSF grant 310030_192659 (M.A.).

## Author contributions

This project was conceived by S.M. and M.A. S.M. designed, performed, and analyzed all the experiments except isolation of α-HA scFv and DARPins. J.V.S. and A.P. isolated DARPins against Dpp. Y.M., Y.H., and M.T. cloned α-HA scFv. D.B. helped with Dpp purification. The main text was written by S.M. and M.A. with comments from all the authors.

## Competing interests

The authors declare no competing interests.
