## [Peer Review File · Nature Communications]

Asymmetric requirement of Dpp/BMP morphogen dispersal in the *Drosophila* wing discREVIEWER COMMENTS

Reviewer #1 (Remarks to the Author):

In this work, the authors look at assess the requirement of Dpp dispersal and signalling on the patterning and growth of the *Drosophila* wing disc, challenging the dogma that morphogen dispersal is critical to action. To do this, they make use of two ligand traps building on the concept of their previously published morphotrap. The distinct action of these traps allows them to distinguish the roles of Dpp dispersal (via their HA-Trap) from Dpp dispersal and signalling (via their Dpp-Trap) on patterning and growth. This leads the authors to make three major claims: Firstly, that anterior patterning and growth are not dependent on Dpp dispersal from anterior stripe source cells, whereas posterior patterning and growth is; Secondly, that this is because of a previously unidentified transient uniform Dpp expression in the anterior wing pouch compartment, with signalling maintained once the Dpp expression domain refines to the anterior stripe; Thirdly, that lateral wing pouch growth can proceed independently of direct Dpp signalling. These conclusions lead the authors to build on and substantiate the previously proposed growth equalisation model. They present *in vivo* evidence for Dpp signalling dependent medial but Dpp signalling independent lateral growth, suggesting both medial and lateral regions are in the wing pouch.

In this manuscript the authors make two main assumptions:

1) That the HA-Trap captured Dpp is still able to bind Tkv and functionally signal, whereas Dpp-Trap captured ligand is not. This assumption is based on the observation that pMad staining, though somewhat diminished, overlaps with the HA-Trap captured ligand domain (Figs. 2b, k, and Extended Data Fig. 2a). This contrasts Dpp-Trap which shows a near complete absence of pMad staining upon expression of Dpp-Trap (Figs. 4d, l and Extended Data Fig. 2k). The authors attribute this difference in activity to the direct binding of Dpp by Dpp-Trap, thereby obstructing ligand-receptor interactions, whereas by binding the HA tag of HA-Dpp, the ligand dimer itself is free to bind, one would assume, proximal receptor. While this explanation is reasonable for the purposes of this manuscript, further characterisation of these traps would be useful for future use of these systems. This could include assessment of receptor binding by HA-Dpp-Trap complexes via SPR or BLI. For completeness, HA trap could be expressed in non-HA-tagged Dpp flies and a control Dpp trap, using a DARPIn that exhibited weak or no binding to Dpp, could be used to show no aberrant effects of the traps on signalling input. It may be simpler to conduct such controls in cell culture and assess read-out via a simple pMad assay.

2) That the transient anterior Dpp expression domain maintains signalling once Dpp is transcriptionally refined to the anterior stripe. This is an underpinning concept to the explanations provided for the Dpp dispersal independent anterior patterning and growth. To address this the authors remove Tkv and observe a de-repression of Brk (Extended Data Fig. 6), leading them to conclude maintenance of signal is at the level of the receptor or higher as opposed to an epigenetic change conferring "memory" in the transiently expressing cells. I am unclear what the authors are proposing for how this persistent signalling might be maintained. This also raises the question of how uniform signal induction and maintenance could lead to correct patterning of the anterior wing pouch.

In the embryo, Dpp signalling is thought to induce a positive feedback effect through the TNF homologue *eiger* (Gavin-Smyth, J., Wang, Y. C., Butler, I. & Ferguson, E. L. A genetic network conferring canalization to a bistable patterning system in *Drosophila*. *Curr. Biol.* 23, 2296–2302 (2013)). Have the authors considered investigating a role for *eiger* in propagation of signalling in the transient source cells?

In addition to these main points, there are several other points requiring further clarification or correction:

- In Fig. 1i the Ollas staining and subsequent quantification shows much more intense but spatially restricted signal. Is the implication that the HA-Trap is concentrating Dpp in this region? If so, could the authors demonstrate equal Dpp levels by integrating the area under the traces in Figs. 1h and 1j –

or by another means.

- In Extended Data Fig. 1b, a *ptc*>HAtrap line is used to assess Caspase3 induction whereas assessment of the Dpp Trap Caspase3 induction uses the broader *nub*>Dpptrap. In addition, the rescue experiment in Extended Data Fig. 1e-g uses the *nub*>HAtrap. Is there a reason for this discrepancy in lines used?
- On line 133 the authors state that “despite undetectable Dpp signaling, the posterior wing pouch grew substantially as judged by the expression of an intervein marker DSRF and a wing pouch marker 5xQE.DsRed (a reporter of the Quadrant Enhancer (QE) of the wing master gene *vg*) (Fig. 2b arrow, 2i)”. However, both by eye in Fig. 2b and by quantification in Fig. 2i the posterior wing pouch is diminished in size. By “grew substantially” do the authors mean that there was more posterior growth than was expected rather than relative to the control? Whilst only minor, perhaps this could be rephrased for clarity.
- In the figure legend for Extended Data Figure 4, line 38, the images are mislabelled. I believe the bottom image is “wing disc expressing HA trap using *ptc*-Gal4” not “wing disc expressing Dpp trap using *ptc*-Gal4”.
- The legend in Figure 6 appears to be mislabelled as the numbers in the figure do not match the annotations used in the text.
- In Figs. 4d, g, l and q and Extended Data Figs. 2 o and p the use of Dpp trap appears to lead to the induction of *Sal* expression in peripheral regions not observed either in control or HA trap discs. In Fig. 7a-f, the authors show that by removing Dpp from the anterior stripe, *Sal* also adopts this peripheral pattern, though say that “Under this condition, while *Sal* expression was completely lost,...” . The authors do not show the *Sal* pattern when Dpp is removed from the entire anterior wing pouch in Fig. 7i and j. This would be good to include given the unexpected pattern of *Sal* compared to *Omb* in the Dpp-signalling compromised wing discs. Is there any relevancy to this pattern, as *Omb* notably does not follow the same pattern, or is it a side effect of the restricted growth phenotype observed upon Dpp trap expression?

Overall, this manuscript is generally well written, clear, and succinct. The authors make an astute summary of their findings, explaining how their approach, the use of ligand traps targeting different actions of the Dpp, meets their aim to assess the requirement of Dpp dispersal and signalling on the patterning and growth of the *Drosophila* wing disc. They place these findings in the context of the current understanding in the field; Considering various models of growth regulation they explain how their findings support one such model, before identifying the major outstanding questions in light of their findings (e.g. how non-uniform growth potential is conferred in the medial and lateral cells). I believe this work will be of interest to others in the field, not just for its findings regarding Dpp dispersal, but also the experimental system used to regionally restrict Dpp dispersal and signalling which will be of interest to others working on Dpp signalling and morphogen action. Their findings challenge the dogma that morphogens depend on dispersal to pattern tissues and by providing evidence to support one model of wing pouch patterning and growth, this paper moves the understanding of the field forward.

Reviewer #2 (Remarks to the Author):

This manuscript by Matsuda et al uses synthetic protein binding tools (a single chain antibody or DARPIn) to manipulate Dpp dispersal and growth in the *Drosophila* wing disc, in order to evaluate the contribution of each to growth and patterning. Based on their findings, the authors suggest that, while Dpp dispersal is required for posterior growth and patterning, there is less of a requirement for Dpp dispersal in the anterior. Instead, they provide evidence for a transient source of Dpp in the anterior, with persistent signalling from this source allowing anterior growth and patterning in the absence of Dpp dispersal.

Overall, I found the data to be of high quality and I think the findings will be of general interest. However, I think that a non-specialist reader would benefit from greater clarification in the writing and

further explanation of the rationale behind some of the experiments, as described below.

It is interesting that the authors argue for transient dpp transcription outside the stripe - the authors acknowledge in the discussion that the idea of an anterior source of Dpp is controversial, with recent reports arguing against it. Based on this, I would like to see more direct evidence for dpp transcription outwith the stripe during disc development, as the destabilized GFP reporter is limited in its ability to report dynamic information about transcriptional activity. Now that smFISH method has been applied to wing discs allowing single mRNAs to be detected (Bakker et al, 2020, elife), it would strengthen the authors argument if they could use smFISH to directly visualize the changing dpp expression domain over time, with single molecule sensitivity.

I am confused by the description of Fig 3. In the text, it says 'we tested whether there was a substantial leakage of Dpp from the HA trap', but it is not explained how the subsequent experiments do this. Is it that tkv clones are normally eliminated unless Brk levels are high? Therefore, the authors infer from the survival of medial tkv clones that Brk is high and therefore Dpp has been successfully sequestered by the HA trap? If so, then I do not see what this tells us over and above the Brk staining shown in Fig. 2b. Maybe I have misunderstood this, but I don't see any clear evidence that the HA trap is not leaky. In contrast, based on the stainings shown in Fig. 2b, pMad appears slightly higher at the 0um position then declines over the distance to 50um (albeit at a greatly decreased level relative to wt), and Brk staining shows a peak in this region. As these 2 staining patterns are not flat, does this not suggest that there is slight leakage from the HA trap?

The authors should clarify what 5xQE.DsRed is, as I don't think it is referenced, and the only explanation appears to be that it is the vg quadrant enhancer. I presume based on its behaviour this is a different enhancer from the vg quadrant enhancer that the Laughon lab suggested Mad directly binds to and activates? Also, is the data in Extended Fig 3 to show that 5xQE.DsRed does not require Dpp for activation, but instead needs Dpp to repress Brk to allow expression of the reporter? If so, why is it expressed in the posterior where there is high Brk in the HA trap discs?

The authors suggest a memory of earlier signalling to allow anterior patterning and growth in the absence of Dpp dispersal. Could this relate to the ideas about signal duration being as important as signal strength in some contexts? Also, they suggest that is not mediated by epigenetic regulation or autoregulation of target gene expression, as they report that Brk is quickly derepressed upon tkv removal. Am I correct in thinking that the time frame of analysis following genetic removal of tkv is at least 16 hrs? If so, I do not see how this rules out epigenetic regulation or autoregulation.

It would help the reader to briefly explain G-trace.

What is the explanation for the sal expression pattern in Fig. 4d,g? I am not sure why it increases laterally.

The authors suggest that the posterior growth control by anterior Dpp signaling is permissive - can they speculate about how this may occur mechanistically?

Reviewer #3 (Remarks to the Author):

This paper from the Affolter lab addresses the long standing problem of understanding the importance of morphogen dispersal in the formation and interpretation of morphogen gradients. This is an issue that the Affolter lab have addressed for many years.

Here the authors develop two different traps. One is the HA trap that presents dispersal of HA-tagged Dpp, but has little effect on signaling by source cells. The other is a Dpp trap made from a DARPIn that

both inhibits Dpp dispersal and signaling.

The authors find that posterior patterning and growth require Dpp dispersal, while anterior patterning and growth is largely independent of Dpp dispersal. This conclusion contradicts the conclusions that the same lab reached using morphotrap traps that trap GFP-tagged Dpp. They go on to show that in the anterior compartment, DPP expression is refined from an initial uniform expression domain, and they also present evidence for another transient source of Dpp expression that is important for signaling in the anterior compartment. Finally they show that lateral wing growth depends neither on Dpp signaling nor dispersal.

In general I think this is a solid piece of work that goes some way to answering the debate in the field about the role of Dpp dispersal. However, there are some key controls missing. I also feel that the paper is written in a cryptic style that makes it difficult to understand for a non-expert as explained below.

1. In Figure 2, the authors haven't formally proved that Dpp dispersal is inhibited in the anterior. This needs to be demonstrated in these particular experiments.
2. The authors conclude that HA-Dpp can still signal in the source cells, and explain this as a result of the HA antibody binding the HA tag and thus not preventing Dpp binding to its receptors. Why then doesn't HA-antibody bound HA-Dpp signal in the posterior compartment?
3. In Figure 5, later wings should also be shown.
4. In Fig 6, the authors show that Dpp expression is initially uniform. Is this also true of P^{Mad} signaling at these early stages?
5. Concerning the clarity of the paper. The authors assume that the readers know where Ptc and Nub are expressed in the wing disc to be able to interpret the experiments. This needs to be made clear. The authors should also explain the G-TRACE analysis.
6. The authors never mention the existence of Gbb, which is another BMP family member expressed much more broadly than Dpp and shown to be important for cell fate specification in the wing and low threshold responses at the edges of the P^{Mad} gradient. The authors need to address the role of Gbb in their experiments. If it is heterodimerized with Dpp, it too could be sequestered by these traps. In addition, Gbb homodimers, which would be unaffected, may contribute to some of the signaling that the authors ascribe to Dpp.
7. In Figure 6, what are the panels under d, e, f, and g?

Reviewer #4 (Remarks to the Author):

In this manuscript by Matsuda et al., the authors design synthetic tools to bind and "trap" Dpp, which would prima facie affect its signaling range. They also used a similar tool to trap HA, which affects signaling of HA-tagged Dpp in a different way than the Dpp trap. Using these two newly-developed tools, the authors found that, in the *Drosophila* wing disc, cells in the posterior compartment require the dispersal of Dpp (ie, the Dpp- or HA-trap inhibited patterning and growth), while cells in the anterior compartment do not. The fact that anterior compartment cells can still proliferate semi-normally when Dpp dispersal is blocked was attributed to the lingering effects of an earlier stage in which dpp was expressed throughout the entire A compartment, and thus it was sufficient for Dpp to signal only to the source cells. It remained unclear how the transient exposure to Dpp could be "remembered" by these cells, but it is a very interesting hypothesis that will hopefully

get tested in the future. Similarly, it was unclear how the P compartment could still grow a little bit without Dpp ever reaching those cells, but this phenotype required the A compartment to receive Dpp signaling at some level. How such a non-autonomous effect occurred remained a mystery, but again, it is an interesting hypothesis.

The broader impact of this work is the development of the tool, which can then be applied to other morphogen systems, or any other system in which hindering protein movement could be beneficial to the researcher.

That being said, there are several outstanding questions that the authors must address before the manuscript can be considered further. In general the major problem is the paper is written so that only a specialist can read it. The authors often skip explanatory details of what they did, perhaps to economize space. I do not fault the authors for this; it is easy to forget how much others don't know. However, this economy of space results in many experimental details being left out (they are not in the Figure Legends, the Methods, or Supplementary Materials either). Oftentimes, the experiments have complex genotypes (found in the Methods, which was well done) or manipulations, and as such, they are difficult to interpret without more detailed explanations. To go along with that point, the flow of logic is also left out in many cases. In at least one instance, the lack of detail makes it difficult to inferring the logical connection between the experimental set up, the observed data, and the authors' conclusion. The authors should keep in mind that Dpp signaling in the wing disc is a system with a host of complicated (and frequently conflicting) literature. I urge the authors to consider this and flesh out the details of the experimental set ups and the logical connections from question/hypothesis to conclusion for each experiment. It is difficult to convey a positive tone in a critical review, but I honestly hope the authors see these criticisms as encouragements to make their manuscript more clear so that readers other than wing disc specialists can understand it.

First, the growth phenotype of the posterior compartment is difficult to convey. Without knowledge of the literature, which the authors do not spell out concretely, someone reading this paper would be very confused as to whether the posterior growth was "inhibited" or "substantial". Thus, the authors need to pay very careful attention to how they describe the phenotype. For example, on lines 132 and 138, the authors use the word "substantial," but that is quite vague. It makes it sound like there is a "lot" of growth, when in fact there is less growth than normal. A more precise word needs to be used. As another example, between Lines 140 and 149, the authors first say the growth defect is severe, (and that Dpp dispersal is critical for growth) to saying the posterior compartment does grow, to saying that the growth contradicts previous reports (what reports? What did these reports conclude?).

Sentences such as (Line 117), "...clonal accumulation of Ollas-HA-Dpp in HA trap clones in the receiving cells was undetectable upon HA trap expression with *ptc*-Gal4 (Fig. 1k-n, arrow), indicating that the HA trap can block HA-Dpp dispersal efficiently," do not sufficiently explain what is happening in the experiment, nor what to expect from the experiment. What do the authors mean by "receiving cells"? What is the staining in the middle panel of Fig. 1k? The label just says "HA trap." Is it an anti-cherry staining?

More on the above sentence: the authors simply call them HA trap clones. I suspect the average reader would assume most of the disc is hetero (or hemi)-zygous for HA trap and the clones would be homozygous for the HA trap (with twin spots of cells lacking HA trap). There is very little in the manuscript at that point to help the reader ascertain what is meant by "HA trap clones." There is a lot happening in these experiments, and a single sentence is not enough to explain the experimental set up, the flow of thought, nor the logic of the conclusion.

Another example (Line 149), "When *tkva12* clones characterized as a null allele 42, 43 were induced in wing discs expressing HA trap with *ptc*-Gal4, *tkva12* clones survived and expressed the 5xQE.DsRed reporter in the anterior lateral regions as well as in the entire posterior region, even next to the source cells (Fig. 3a), indicating that leakage is negligible." Again, there is a lot going on in this experiment,

and a single sentence is not sufficient to explain it. Why should a clone homozygous for a null allele for tkv show that there is no leakage of HA-Dpp out of the HA trap? The best I could come up with that matches the authors' conclusion is that, when Dpp dispersal is blocked by the HA trap, then clones of tkv survive because they are not outcompeted by other cells in the P compartment which would have been growing had those other cells received the leaked Dpp. That's why in the wildtype disc, only lateral clones are found (no medial). This seems like a convoluted logic, but if that's correct, then it should be spelled out explicitly. If it's not correct, then the actual flow of logic should be spelled out.

Another example (Line 155), "In rare cases, even medial tkv null clones survived and expressed 5xQE.DsRed (Fig. 3d), indicating that the elimination of tkv mutant clones masked Dpp signaling-independent 5xQE.DsRed expression in previous studies." Again, it is unclear how the conclusion follows from the observation. There is also no explanation of what the previous studies found, how those conflict with the authors' data, or a logical flow of thought by which the authors resolve the conflict. Further, there is no citation of said previous studies for the readers to try to figure this out themselves.

Another example (the very following sentence), "Consistently, upon genetic removal of dpp from the entire A compartment as early as the beginning of second instar stage, when the wing pouch is defined, 5xQE.DsRed remained expressed despite severe growth defects (Fig. 3e-h)." An entire series of panels, including the experimental set up, what to expect, and why, is compressed into a sentence.

Another example (Line 221): "we expressed HA trap with ptc-Gal4 at defined time points using tubGal80ts. Upon HA trap expression from the mid-second instar stage..." Temperature shift experiments could be quite complicated, depending on the set up. The authors did not give any indication of what temperatures the larvae were raised at, or when temperature shifts happened. Were the larvae raised at the permissive temperature until a permanent shift to the restrictive, or was there only a short period of the restrictive before shifting back? When did the temperature shift happen? How did they synchronize the ages of the larvae?

Line 227: Same as above – the authors did not explain the temperature shift experiment. When did the flp-out occur?

Line 233: The authors abruptly state "By removing dpp from the entire A compartment" without any explanation that, to do this, they used a different Gal4 line. This should be stated explicitly ("ci" doesn't appear anywhere in the paragraph), because this is embedded in a paragraph that begins with a sentence saying the authors used ptc-Gal4. Nor is there any explicit connection made between ci and "entire A compartment." It is not enough to assume the readers will figure this out on their own from reading the Figure legend.

Line 238: "Upon genetic removal of tkv, Brk was quickly de-repressed (Extended Data Fig. 6), indicating that the "memory" of an earlier signal is not..." There is very little explanation of what the authors did in this experiment. The genotype is given in the Methods section, but that is not enough. The Extended Data Fig. 6 legend is sparse on detail, and is also confusing as stated. Are all discs the same age, and the only difference is how long tkv had been removed? Why does the 5th disc look smaller than the rest is it younger? Is it a growth phenotype? Is the scale bar the same in all images? Furthermore, what do the authors mean by "quickly?" This is a vague word. How long before Brk was de-repressed? How does that compare to the expected time and/or time scales of other events in this system?

Line 306: "ubiquitous blocking of Dpp dispersal by HA trap during development caused lethality (data not shown)." This is an interesting claim, but it needs further explanation. Are the authors referring to an HA trap that was expressed starting in the early embryonic stages? Is that what they mean by "ubiquitous" and "during development?" This statement is too vague, and genotypes and other

experimental details are not given. Perhaps that was the point? Perhaps, since the authors wanted put this claim into the Discussion section rather than the Results section, the thinking is it would not need to be fully explained? But that is not the case. In general, new claims about data should not be introduced in the Discussion section. If this is a phenotype they noticed upon early iterations of creation of the HA trap, then it could go in one of the first subsections of the Results section. [Also, I am not sure if "data not shown" is acceptable for a Nature Communications paper.] If the authors think it is not a solid enough result to go into the Results, then neither should it appear in the Discussion.

Minor:

- It is unclear from the abstract or intro why HA trap should affect Dpp.
- The expected function of OLLAS is not explained.
- Line 117, "Furthermore, clonal accumulation of Ollas-HA-Dpp in HA trap clones in the receiving cells was undetectable..." This does not seem to be the case. There is clear, detectable green fluorescence in the clone that the arrow is pointing to. Unless the authors mean something different from what I am inferring, in which case the flow of logic should be explained more clearly.
- Fig. 1 legend: arrows should be explained in the legend.
- The abbreviation "ExHA" and similar are not explained.
- The specific method that generates an anti-ExHA (and similar), and what the results would mean, are not explained.
- Fig 1o: it is not clear what is being merged in the fourth panel. Panel 1 is green, Panel 2 is magenta, and if those two were merged, it would produce white. However, Panel 3 itself is white, so it is unclear what white means in Panel 4. On the clones, white appears to be the merge of green and magenta, but elsewhere, white appears to be pMad. How would one distinguish?
- Sentence that begins on line 128 with "Upon HA trap expression..." is ambiguous. Does the whole sentence refer to the P compartment, or just the Brk upregulation? The fact that "in the P compartment" comes after the "and" makes it difficult to know. That prepositional phrase should come earlier in the sentence to relieve the ambiguity. Or, if it's the latter, then "in the whole pouch" should be applied to the statement about pMad, Sal, and Omb.
- Sentence that starts on line 149 with "When tkva12 clones characterized" is very long and awkward. Other sentences in this section also fall prey to the same difficulty, likely because the authors are trying to squeeze so much information into a single sentence (see my "major criticisms" above).
- Line 176: "Interestingly, anterior Dpp trap expression caused ... anterior growth defects than HA trap." The authors should edit for grammar. As written, both parts of the sentence should work within the sentence as a whole. That is, when you remove "more severe posterior growth defects as well as," the sentence should still make sense. Perhaps this could be solved just by putting "more" in front of anterior.
- Line 178: "This non-autonomous effect was hardly seen by simply removing tkv from the entire A compartment, probably because Dpp can still disperse and control patterning and growth under this condition (data not shown)." There are several issues with this sentence. First, it is unclear what the authors mean by the vague statement "...effect was hardly seen..." Second, it would be nice if the authors could comment as to how Dpp signaling could take place without tkv. Third, and I am not sure about this, but my guess is that Nature frowns upon "data not shown."
- Line 180 (next sentence): the authors should probably temper their claim a bit. They should include a qualifier, such as "likely".
- Line 184: "Taken together..." It would be nice if the authors could comment on how signaling in just the source cells, without Dpp dispersal, could affect growth in the rest of the pouch.
- Line 193: Should be "depend".
- Line 194: It is nice to see this question asked, but it should probably come earlier in the manuscript (see "Line 184" comment).
- Lines 195 – 201: It would be nice if the authors could comment on the anterior compartment's phenotype in which the dpp-Gal4 cells grow to engulf the entire A compartment. For example, Fig 5d shows that pMad, Sal, and Omb expand to the entire A compartment, and Brk is completely lacking

there.

- Line 205: the above point is slightly resolved here (incl. ref 47). However, this point should come in the previous section of the paper, when the reader would encounter the phenotype (referenced in Fig. 5d). The reader should not have to wait and wonder. Even so, this section with the destabilized GFP (Fig 6) is quite elegant and clearly proves the authors' point.
- Same line: "lineages...were uniform," perhaps? "Lineages" is a plural noun, correct?
- Fig. 6 Legend: I am pretty sure the legend does not correspond to the figure.
- Fig. 7 Legend: There needs to be a more clear explanation of the difference between Fig. 7d and 7g, as well as Fig. 7f vs 7h.
- Line 259: should be "activates"
- Line 282: "generates a" or "generates the"
- Line 288: "generates"
- Around or after Line 320: It would be nice if the authors explicitly stated their preferred explanation as to *how there could be* "non-autonomous posterior growth induction by anterior Dpp signal" rather than only stating what it is not due to.

Point-by-point response to the reviewers' comments

Reviewer comments are displayed in blue, our replies to these comments in black.

Reviewer #1 (Remarks to the Author):

In this work, the authors look at assess the requirement of Dpp dispersal and signalling on the patterning and growth of the *Drosophila* wing disc, challenging the dogma that morphogen dispersal is critical to action. To do this, they make use of two ligand traps building on the concept of their previously published morphotrap. The distinct action of these traps allows them to distinguish the roles of Dpp dispersal (via their HA-Trap) from Dpp dispersal and signalling (via their Dpp-Trap) on patterning and growth. This leads the authors to make three major claims: Firstly, that anterior patterning and growth are not dependent on Dpp dispersal from anterior stripe source cells, whereas posterior patterning and growth is; Secondly, that this is because of a previously unidentified transient uniform Dpp expression in the anterior wing pouch compartment, with signalling maintained once the Dpp expression domain refines to the anterior stripe; Thirdly, that lateral wing pouch growth can proceed independently of direct Dpp signalling. These conclusions lead the authors to build on and substantiate the previously proposed growth equalisation model. They present *in vivo* evidence for Dpp signalling dependent medial but Dpp signalling independent lateral growth, suggesting both medial and lateral regions are in the wing pouch.

In this manuscript the authors make two main assumptions:

1) That the HA-Trap captured Dpp is still able to bind Tkv and functionally signal, whereas Dpp-Trap captured ligand is not. This assumption is based on the observation that pMad staining, though somewhat diminished, overlaps with the HA-Trap captured ligand domain (Figs. 2b, k, and Extended Data Fig. 2a). This contrasts Dpp-Trap which shows a near complete absence of pMad staining upon expression of Dpp-Trap (Figs. 4d, l and Extended Data Fig. 2k). The authors attribute this difference in activity to the direct binding of Dpp by Dpp-Trap, thereby obstructing ligand-receptor interactions, whereas by binding the HA tag of HA-Dpp, the ligand dimer itself is free to bind, one would assume, proximal receptor. While this explanation is reasonable for the purposes of this manuscript, further characterisation of these traps would be useful for future use of these systems. This could include assessment of receptor binding by HA-Dpp-Trap complexes via SPR or BLI. For completeness, HA trap could be expressed in non-HA-tagged Dpp flies and a control Dpp trap, using a DARPIn that exhibited weak or no binding to Dpp, could be used to show no aberrant effects of the traps on signalling input. It may be simpler to conduct such controls in cell culture and assess read-out via a simple pMad assay.

We thank the reviewer for these suggestions. Although the fact that refolded Dpp tends to precipitate in the normal buffer prevented us from performing SPR or BLI, we expressed HA trap using *ptc*-Gal4 in the absence of a HA-tagged protein as a control (Extended Data Fig. 1) and confirmed that HA trap is inert in the absence of a HA-tagged protein. These results were described as follows (line120-122):

“HA trap expression in the anterior stripe of cells of wild type wing discs using *ptc-Gal4* did not interfere with Dpp signaling in the wing disc or patterning and growth of the adult wing (Extended Data Fig. 1)”

For DARPin, we expressed a control Dpp trap derived from a DARPin with no measurable affinity to Dpp using *ptc-Gal4* (Extended Data Fig. 8) and described the result as follows (line277-280):

“These severe phenotypes were not due to a common scaffold effects of DARPins, since one of the traps (containing DARPin 1240_C9) that failed to trap Dpp did not interfere with pMad accumulation in the wing disc or patterning and growth of the adult wing when expressed using *ptc-Gal4* (Extended Data Fig. 8)”

2) That the transient anterior Dpp expression domain maintains signalling once Dpp is transcriptionally refined to the anterior stripe. This is an underpinning concept to the explanations provided for the Dpp dispersal independent anterior patterning and growth. To address this the authors remove Tkv and observe a de-repression of Brk (Extended Data Fig. 6), leading them to conclude maintenance of signal is at the level of the receptor or higher as opposed to an epigenetic change conferring “memory” in the transiently expressing cells. I am unclear what the authors are proposing for how this persistent signalling might be maintained. This also raises the question of how uniform signal induction and maintenance could lead to correct patterning of the anterior wing pouch. In the embryo, Dpp signalling is thought to induce a positive feedback effect through the TNF homologue *eiger* (Gavin-Smyth, J., Wang, Y. C., Butler, I. & Ferguson, E. L. A genetic network conferring canalization to a bistable patterning system in *Drosophila*. *Curr. Biol.* 23, 2296–2302 (2013)). Have the authors considered investigating a role for *eiger* in propagation of signalling in the transient source cells?

We thank the reviewer for pointing this out. It remains currently unknown how the transient Dpp source maintains weak but persistent Dpp signaling after *dpp* expression is refined to the anterior stripe of cells. Therefore, we discussed potential mechanisms underlying the “memory” effect in the discussion, including (1) feedback mechanisms to sustain Dpp signaling, and (2) stability of *dpp* mRNA and/or Dpp protein as follows (line504-517);

“It remains unknown how transient *dpp* expression can maintain Dpp signaling in the anterior lateral region (Fig. 8). One possibility is that feedback factors control the duration of Dpp signaling. Various feedback factors have been shown to regulate Dpp signaling. For example, the *Drosophila* tumor necrosis factor α homolog *Eiger* and a secreted BMP-binding protein *Crossveinless-2* (*Cv-2*) are positively regulated by Dpp signaling to either positively and negatively influence Dpp signaling in the early embryo, respectively⁶⁶. While JNK is not activated by *Eiger* in the wild type wing pouch⁶⁷, *Cv-2* acts as a positive feedback factor to specify the posterior crossvein during pupal stages⁶⁸. Whether these two factors act during wing pouch patterning and growth remains to be addressed. *Pentagon* is a secreted feedback factor repressed by Dpp signaling to positively regulate Dpp signaling in the wing disc⁶⁹. *Pent* is produced in the lateral region of the wing pouch and regulates Dpp signaling and proliferation there. Thus, in addition to its role for scaling, *Pent* may control the

duration of Dpp signaling in the lateral region. Another possibility is that, since it takes quite some time (20-24 hr at 29 °C) to eliminate Dpp protein upon excision of *dpp*^{24,41}, relatively stable *dpp* mRNA and/or protein after the termination of transient *dpp* transcription may contribute to Brk repression (~28 hr at 29 °C). ”

Eiger is one of the candidates that may confer “memory” of signaling as a feedback regulator. Although *puc-LacZ*, a readout of the Eiger/JNK signaling, is not activated in the wing pouch (Adachi-Yamada et al., 1999), a direct experimental setup to test whether Eiger is involved in the memory would be to remove *dpp* only from an anterior stripe of cells in an *eiger* mutant background to ask whether pMad signaling and Omb expression are lost. However, since the *eiger* locus is close to the locus where *ptc-Gal4* is inserted, we could so far not test this hypothesis directly.

In addition to these main points, there are several other points requiring further clarification or correction:

- In Fig. 1i the Ollas staining and subsequent quantification shows much more intense but spatially restricted signal. Is the implication that the HA-Trap is concentrating Dpp in this region? If so, could the authors demonstrate equal Dpp levels by integrating the area under the traces in Figs. 1h and 1j – or by another means.

As the reviewer pointed out, in Fig.2d in the revised manuscript (=Fig. 1i in the previous version), the Ollas staining and subsequent quantification indeed showed a much more intense but spatially restricted signal. This implies that Ollas-HA-Dpp was trapped in the main source cells by HA trap and accumulated there. However, it is clear that the amount of total extracellular Dpp protein is not equal; the amount of Ollas-HA-Dpp upon HA trap expression is much higher than that of Ollas-HA-Dpp in the control, likely because Dpp is stabilized upon binding to HA trap.

- In Extended Data Fig. 1b, a *ptc>HAtrap* line is used to assess Caspase3 induction whereas assessment of the Dpp Trap Caspase3 induction uses the broader *nub>Dpptrap*. In addition, the rescue experiment in Extended Data Fig. 1e-g uses the *nub>HAtrap*. Is there a reason for this discrepancy in lines used?

To test for cell death, we selected the two conditions based on the strength of the phenotypes. Among the Gal4 lines used in this study, HA trap expression showed the strongest phenotypes with *ptc-Gal4*, and Dpp trap showed the strongest phenotypes with *nub-Gal4*. We chose the most severe conditions to properly evaluate Caspase3 induction, in order to exclude the possibility that Caspase is not affected because of the minor growth defects observed. To avoid any confusion, we explained our approach in the figure legend (in Extended Data Fig. 4 in the revised manuscript).

For the rescue experiments, we used *nub-Gal4* mainly due to a technical reason. We wanted to block cell death in “the entire wing pouch” to test if cell death is the reason for the smaller wing disc size upon HA trap expression or Dpp trap expression. To do so, *nub-Gal4* was convenient since *nub-Gal4* expression covers the entire wing pouch region, and HA trap expression or Dpp trap expression using *nub-Gal4* showed growth defects. By co-expressing

HA trap or Dpp trap with p35, we thus could easily ask whether cell death is responsible for the growth defects by HA trap or Dpp trap.

- On line 133 the authors state that “despite undetectable Dpp signaling, the posterior wing pouch grew substantially as judged by the expression of an intervein marker DSRF and a wing pouch marker 5xQE.DsRed (a reporter of the Quadrant Enhancer (QE) of the wing master gene *vg*) (Fig. 2b arrow, 2i)”. However, both by eye in Fig. 2b and by quantification in Fig. 2i the posterior wing pouch is diminished in size. By “grew substantially” do the authors mean that there was more posterior growth than was expected rather than relative to the control? Whilst only minor, perhaps this could be rephrased for clarity.

We agree that we should pay careful attention to how we describe the phenotype in Fig.3 in the revised manuscript (=Fig.2 in the previous version). We now described the following two points separately; (1) Upon HA trap expression, the posterior growth was “inhibited” compared with the control, and (2) Despite the growth defects, the posterior compartment did still grow “substantially” compared with the more severe *dpp* mutant phenotypes.

We first described the “inhibition” of growth as follows (line158-164):

“The posterior wing pouch growth was also affected as revealed by the expression of an intervein marker DSRF and a wing pouch marker *5xQE.DsRed*⁴⁵ (Fig. 3b arrow, 3i). Interestingly, although *5xQE.DsRed* contains five copies of the 806 bp Quadrant Enhancer (QE) of the wing master gene *vg* containing a Mad binding site and is therefore thought to be directly regulated by Dpp signaling^{46,47}, *5xQE.DsRed* remained expressed in the P compartment without detectable Dpp signaling (Fig. 3b arrow).”

We then described the “substantial” growth in the following paragraph. We avoided to use “substantial” in the text but precisely described the proportion of the posterior compartment (40%) that still grew upon HA trap expression. Furthermore, we briefly explained why the posterior growth without detectable Dpp signaling upon HA trap expression contradicts results presented in a previous report (Burke 1996) (line182-187);

“A critical role of Dpp dispersal for posterior patterning and growth is consistent with a role of Dpp as a morphogen. However, the overall phenotypes caused by HA trap was surprisingly mild when compared to the phenotypes seen in *dpp* mutants (Fig. 6a, c). Given the requirement of Dpp signaling for cell proliferation and survival in the entire wing pouch⁴⁸, it was surprising that about 40% of the posterior wing pouch was able to grow and differentiate into adult wing tissue without detectable Dpp signaling (Fig. 3d, i, j).”

- In the figure legend for Extended Data Figure 4, line 38, the images are mislabelled. I believe the bottom image is “wing disc expressing HA trap using *ptc*-Gal4” not “wing disc expressing Dpp trap using *ptc*-Gal4”.

We corrected to “wing disc expressing HA trap using *ptc*-Gal4” in line 105 in Extended Data Figure 10 in the revised manuscript (=Extended Data Figure 4 in the previous version).

- The legend in Figure 6 appears to be mislabelled as the numbers in the figure do not match the annotations used in the text.

We corrected the mislabeling in Figure 7 in the revised manuscript (=Figure 6 in the previous version).

- In Figs. 4d, g, l and q and Extended Data Figs. 2 o and p the use of Dpp trap appears to lead to the induction of Sal expression in peripheral regions not observed either in control or HA trap discs. In Fig. 7a-f, the authors show that by removing Dpp from the anterior stripe, Sal also adopts this peripheral pattern, though say that “Under this condition, while Sal expression was completely lost,...” . The authors do not show the Sal pattern when Dpp is removed from the entire anterior wing pouch in Fig. 7i and j. This would be good to include given the unexpected pattern of Sal compared to Omb in the Dpp-signalling compromised wing discs. Is there any relevancy to this pattern, as Omb notably does not follow the same pattern, or is it a side effect of the restricted growth phenotype observed upon Dpp trap expression?

We agree with the reviewer that the use of Dpp trap appears to lead to the induction of Sal expression in peripheral regions. It has previously been shown that Sal is expressed not only in the medial region but also in the lateral region (Journal of Cell Science 2012 125: 5811-5818; doi: 10.1242/jcs.110569). The same study also showed that the medial Sal expression is Dpp signaling-dependent but that lateral Sal expression is Brk-dependent. Thus, one possibility is that lateral Sal is induced by upregulation of Brk upon Dpp trap expression. However, when we re-analyzed the peripheral region of the control wing disc (in a basal confocal section), we found the lateral Sal expression of the control wing disc is actually comparable to that of the wing disc expressing Dpp trap (Extended Data Fig. 9a, b). Thus, it appears that the lateral Sal expression in the control wing disc (found in a basal confocal section) was simply missed when we focused on the medial Sal expression (found in a more apical confocal section) due to the curved tissue architecture of the relatively large wing disc.

To further follow up on this, as requested by the reviewer, we also removed *dpp* from the entire anterior wing pouch and analyzed Sal expression (Extended Data Fig. 9c-e). Consistent with the above results, we found that medial Sal expression was lost but lateral Sal expression was not significantly changed upon removal of *dpp*. These results were described as follows (line282-293);

“We note that upon Dpp trap expression, Sal expression was lost from the medial region but appeared to be upregulated in the lateral region (Fig. 5d, l, Extended Data Fig. 2o, arrow). It has previously been shown that Sal is expressed not only in the medial region but also in the lateral region⁵⁸. The same study also showed that the medial Sal expression is Dpp signaling-dependent but lateral Sal expression is Brk-dependent. Thus, upregulation of Brk upon Dpp trap expression could cause the lateral Sal upregulation. However, when we focused on the peripheral region of the control wing disc (basal confocal section), we noticed that the lateral Sal expression of the control wing disc was actually comparable to that of the wing disc expressing Dpp trap (Extended Data Fig. 9a, b). Thus, when we focused on the medial Sal expression (apical confocal section), the lateral Sal expression of the control wing disc

was simply missed due to the tissue architecture. Consistently, when *dpp* was removed from the entire A compartment using *ci-Gal4* from mid-second instar stage, *Sal* expression was lost from the medial region but not significantly upregulated in the lateral region (Extended Data Fig. 9c-e).”

Overall, this manuscript is generally well written, clear, and succinct. The authors make an astute summary of their findings, explaining how their approach, the use of ligand traps targeting different actions of the Dpp, meets their aim to assess the requirement of Dpp dispersal and signalling on the patterning and growth of the *Drosophila* wing disc. They place these findings in the context of the current understanding in the field; Considering various models of growth regulation they explain how their findings support one such model, before identifying the major outstanding questions in light of their findings (e.g. how non-uniform growth potential is conferred in the medial and lateral cells). I believe this work will be of interest to others in the field, not just for its findings regarding Dpp dispersal, but also the experimental system used to regionally restrict Dpp dispersal and signalling which will be of interest to others working on Dpp signalling and morphogen action. Their findings challenge the dogma that morphogens depend on dispersal to pattern tissues and by providing evidence to support one model of wing pouch patterning and growth, this paper moves the understanding of the field forward.

We would like to thank the reviewer for positive comments on our work.

Reviewer #2 (Remarks to the Author):

This manuscript by Matsuda et al uses synthetic protein binding tools (a single chain antibody or DARPIn) to manipulate Dpp dispersal and growth in the Drosophila wing disc, in order to evaluate the contribution of each to growth and patterning. Based on their findings, the authors suggest that, while Dpp dispersal is required for posterior growth and patterning, there is less of a requirement for Dpp dispersal in the anterior. Instead, they provide evidence for a transient source of Dpp in the anterior, with persistent signalling from this source allowing anterior growth and patterning in the absence of Dpp dispersal.

Overall, I found the data to be of high quality and I think the findings will be of general interest. However, I think that a non-specialist reader would benefit from greater clarification in the writing and further explanation of the rationale behind some of the experiments, as described below.

We would like to thank the reviewer for the positive comments on our work. We fully agree with the reviewer that the manuscript was written in a way that only specialists working on the topic could understand. In the revised manuscript, we tried our best to explain the logical connections from question/hypothesis to conclusion, experimental setup, results, interpretations, etc. as detailed as possible, so that non-specialists can also follow the text and understand.

It is interesting that the authors argue for transient *dpp* transcription outside the stripe - the authors acknowledge in the discussion that the idea of an anterior source of Dpp is controversial, with recent reports arguing against it. Based on this, I would like to see more direct evidence for *dpp* transcription outwith the stripe during disc development, as the destabilized GFP reporter is limited in its ability to report dynamic information about transcriptional activity. Now that smFISH method has been applied to wing discs allowing single mRNAs to be detected (Bakker et al, 2020, elife), it would strengthen the authors argument if they could use smFISH to directly visualize the changing *dpp* expression domain over time, with single molecule sensitivity.

We thank the reviewer for this suggestion. We searched for literature that applied smFISH method to visualize *dpp* transcription and found such a report that successfully visualize *dpp* transcription in the Drosophila germline stem cell niche using RNAscope technology (Development (2018) 145, dev158527. doi:10.1242/dev.158527). We purchased the probe used in the study and tried smFISH using RNAscope technology in the wing disc. Consistent with the *dpp* transcription reporter expression, we found uniform anterior *dpp* transcription until the early third instar stage (Fig. 7i, i', j, j') and an anterior stripe of *dpp* transcription in the later stages (Fig. 7k, k', l, l'). Thus, we now provided direct evidence for the dynamic *dpp* transcription during wing disc development in the revised version of the manuscript. We described these results (line372-376) and methods (line 718-733) as follows:

(line372-376)

“To directly follow *dpp* transcription, we also performed smFISH using RNAscope technology to visualize *dpp* transcripts *in situ*. Consistent with the dynamic expression of the *dpp* transcription reporter, we found uniform anterior *dpp* transcription until the early third

instar stage (Fig. 7i, i', j, j') and an anterior stripe of *dpp* transcription in the later stages (Fig. 7k, k', l, l')."

(line 718-733)

"RNAscope

smFISH using RNAscope technology with the probe against *dpp* (Cat No. 896761) has previously been successful to visualize *dpp* mRNA in the germline stem cell niche⁷⁹. The probes against *dpp* target 682-1673 of MM_057963.5 (accession number from NCBI). RNAscope was performed in an Eppendorf tube. Larvae were dissected in PBS and fixed in 4% paraformaldehyde in PBS for 30 min. Fixed larvae were washed with PBS and then with PBT (PBS containing 0.03% Triton X-100). Fixed larvae were dehydrated at RT (room temperature) in a series of 25%, 50%, 75% and 100% methanol in PBT, and stored overnight in 100% methanol at -20 °C. Larvae were rehydrated at RT in a series of 75%, 50%, 25%, 0% methanol in PBT. Larvae were treated with protease using Pretreat 3 (RNAscope H2O2 & Protease Plus Reagents; ACD, 322330) at RT for 5 min. After washing with PBT, hybridization using probes against *dpp* was performed overnight at 40 °C. The following day, larvae were washed with in RNAscope wash buffer and re-fixed with 4% paraformaldehyde in PBS for 10 min. After washing with PBS, fluorescent signal was developed using RNAscope Fluorescent Multiplex Reagent Kit according to the manufacturer's instructions. Wing discs were mounted in Vectashield (H-1000, Vector Laboratories) and dissected for imaging. Images of wing discs were obtained using a Leica TCS SP5 confocal microscope (section thickness 1 μm)."

I am confused by the description of Fig 3. In the text, it says 'we tested whether there was a substantial leakage of Dpp from the HA trap', but it is not explained how the subsequent experiments do this. Is it that tkv clones are normally eliminated unless Brk levels are high? Therefore, the authors infer from the survival of medial tkv clones that Brk is high and therefore Dpp has been successfully sequestered by the HA trap? If so, then I do not see what this tells us over and above the Brk staining shown in Fig. 2b. Maybe I have misunderstood this, but I don't see any clear evidence that the HA trap is not leaky. In contrast, based on the stainings shown in Fig. 2b, pMad appears slightly higher at the Oum position then declines over the distance to 50um (albeit at a greatly decreased level relative to wt), and Brk staining shows a peak in this region. As these 2 staining patterns are not flat, does this not suggest that there is slight leakage from the HA trap?

We are sorry for the confusion. In addition to our results showing that the HA trap is very efficient in trapping Ollas-HA-Dpp or HA-Dpp (Fig. 2, 3 in the revised manuscript), we further showed that trapping HA-Dpp using HA trap using both *nub-Gal4* and *ptc-Gal4* did not enhance the defects (Extended Data Fig.3 in the revised manuscript). Although these results show that HA trap is efficient in trapping Dpp, as the reviewer pointed out, there might be slight leakage of Dpp and the leaked Dpp may contribute to *5xQE.DsRed* expression and growth in the posterior compartment seen upon HA trap expression.

We are fully aware of the limitation of our approaches and do not want to conclude that there is no leakage at all even if the affinity of HA trap is high. Thus, the purpose of Fig.4 in the revised manuscript (=Fig.3 in the previous version) was actually not to demonstrate that HA trap is fully effective, but rather to ask whether leakage of Dpp from HA trap, if it would

occur, could explain the *5xQE.DsRed* expression and growth in the posterior compartment seen upon HA trap expression.

To test this, we removed *tkv* in the wing disc expressing HA trap, since *tkv* is essential for Dpp signal transduction and for the survival of wing pouch cells. The results suggest that posterior growth seen upon HA trap expression is *Tkv*-independent, indicating that, even if there were leakage of Dpp, this would not account for the posterior growth or *5xQE.DsRed* expression. We described these results as follows (line189-200):

“We therefore tested whether the posterior growth and *5xQE.DsRed* expression seen upon HA trap expression is caused by low levels of HA-Dpp leaking from the HA trap expressed in the source. In this case, the posterior growth and *5xQE.DsRed* expression seen upon HA trap expression should be dependent on *tkv*, an essential receptor for Dpp signaling. To test this, mutant clones of *tkv^{o12}* (characterized as a null allele^{49,50}) were induced in wing discs expressing HA trap with *ptc-Gal4* between mid-second and beginning of third instar stages and analyzed in the late third instar stage. We found that *tkv^{o12}* clones survived and expressed the *5xQE.DsRed* reporter in the anterior lateral regions as well as in the entire posterior region. We also noticed that *tkv^{o12}* clones survived and expressed the *5xQE.DsRed* reporter even next to the source cells in the P compartment (Fig. 4a). These results indicate that the lateral growth and *5xQE.DsRed* expression seen upon HA trap expression is independent of Dpp signaling, and not caused by a leakage of HA-Dpp from the HA trap, even if such leakage would occur.”

The authors should clarify what *5xQE.DsRed* is, as I don't think it is referenced, and the only explanation appears to be that it is the *vg* quadrant enhancer. I presume based on its behaviour this is a different enhancer from the *vg* quadrant enhancer that the Laughon lab suggested *Mad* directly binds to and activates?

We now clarified the nature of *5xQE.DsRed*, added references, and described the *vg* quadrant enhancer as follows(line160-164):

“Interestingly, although *5xQE.DsRed* contains five copies of the 806 bp Quadrant Enhancer (QE) of the wing master gene *vg* containing a *Mad* binding site and is therefore thought to be directly regulated by Dpp signaling^{46,47}, *5xQE.DsRed* remained expressed in the P compartment without detectable Dpp signaling (Fig. 3b arrow).”

As the reviewer pointed out, the *Vg* QE enhancer is the enhancer that the Laughon lab suggested *Mad* directly binds to and activates. Thus, it was surprising to us that *5xQE.DsRed* remained expressed in the P compartment independent of Dpp signaling. Therefore, we carefully addressed this issue in the following section (Fig.4, Extended Data Fig. 5, 7). Our results show that QE is largely induced by repressing *brk*, but physiological level of *Brk* is not sufficient to completely repress QE and suggest the presence of additional inductive inputs for QE activation.

Also, is the data in Extended Fig 3 to show that *5xQE.DsRed* does not require Dpp for activation, but instead needs Dpp to repress *Brk* to allow expression of the reporter? If so, why is it expressed in the posterior where there is high *Brk* in the HA trap discs?

Indeed, Extended Data Fig.7 in the revised manuscript (= Extended Data Fig.3 in the previous version) is to show that *5xQE.DsRed* is mainly induced by repressing Brk. As the reviewer asked, this raises the question why *5xQE.DsRed* is expressed in the P compartment where Brk is high upon HA trap expression. We noticed that Brk and *5xQE.DsRed* are co-expressed in the lateral region of the control wing disc (Fig. 4f, Extended Data Fig. 5b', 5f'), and the observed high Brk levels are comparable to the Brk levels in the lateral region of the control wing disc (Fig. 3f, 3p), we speculate that Brk is not sufficient to repress *5xQE.DsRed* expression at its physiological level and that there are additional factors required for *5xQE.DsRed* expression. We discussed this issue as follows (line244-256):

“How is *5xQE.DsRed* expression regulated if QE is not directly regulated by Dpp signaling? While *5xQE.DsRed* expression is completely lost in *dpp* mutants, we found that *5xQE.DsRed* reporter expression was rescued in *dpp, brk* double mutant wing discs (Extended Data Fig. 7), indicating that *5xQE.DsRed* expression is largely induced by repressing *brk*, similar to the regulation of other *dpp* target genes. Indeed, QE has also been shown to be activated in *brk* mutant clones in the wing disc⁵³. However, this notion appears inconsistent with the fact that *5xQE.DsRed* expression was not repressed in the region where Brk is high in various conditions, in which Dpp signaling is compromised (Fig. 3b, 4h, Extended Data Fig. 5d', 5h'). We noticed that the observed high Brk levels upon Dpp trapping were comparable to the Brk level in the lateral region of the control wing disc (Fig. 3f, 3p), and Brk and *5xQE.DsRed* were co-expressed in the lateral region of the control wing disc (Fig. 4f, Extended Data Fig. 5b', 5f'). Thus, we speculate that Brk is not sufficient to repress *5xQE.DsRed* expression at physiological levels in lateral regions and that there are additional inductive inputs. Consistent with this idea, QE has also been shown to be regulated by Wg^{45,54}.”

The authors suggest a memory of earlier signalling to allow anterior patterning and growth in the absence of Dpp dispersal. Could this relate to the ideas about signal duration being as important as signal strength in some contexts? Also, they suggest that is not mediated by epigenetic regulation or autoregulation of targetgene expression, as they report that Brk is quickly derepressed upon *tkv* removal. Am I correct in thinking that the time frame of analysis following genetic removal of *tkv* is at least 16 hrs? If so, I do not see how this rules out epigenetic regulation or autoregulation.

As mentioned by the reviewer, our results suggest that the memory of earlier Dpp signaling mediated by transient Dpp source allows anterior patterning and growth robust against the absence of Dpp dispersal, and are consistent with the idea that signal duration is an important factor.

Concerning the de-repression of Brk and epigenetic regulation or autoregulation of Brk, we consider two possibilities. One possibility is that persistent low pMad levels are continuously required to repress Brk. Alternatively, Brk repression by early Dpp signaling persists in the later stages without continuous Dpp signaling, for example, by epigenetic regulation or autoregulation.

We found de-repression of Brk as early as 16hr at 29°C upon *tkv* removal (Extended Data Fig. 13). By considering perdurance activity of Gal80ts for 6hr at 29°C after temperature shift

(Bergantinos, C et al., 2010), we can estimate that Brk is derepressed within 10hr at 29°C if Dpp signaling is lost.

In contrast, pMad signaling and Brk repression were still visible in the anterior lateral region 48hr at 29°C after *dpp* was removed from the anterior stripe of cells (Fig. 8). Even if we consider that it takes about 20hr at 29°C to eliminate the majority of the Dpp protein after genetic removal of *dpp* (Akiyama and Gibson., 2015)(Matsuda and Affolter, 2017), the pMad signaling and Brk repression mediated by the transient *dpp* transcription persist at least for 28hr at 29°C. Thus, we think that epigenetic regulation or autoregulation of Brk, even if it exists, unlikely explains the persistent Brk repression. These results were described in the results section as follows (line 397-405) and (line 424-437):

(line397-405)

“To directly test this, we applied Gal80ts to genetically remove *dpp* via *dpp^{FO}* allele upon FLP expression using *ptc-Gal4* from the mid-second instar stage. To do so, the larvae were raised at 18 °C for 5 days before a temperature shift to 29 °C and were then dissected 48 hr later. In this setup, *dpp* was removed approximately from the anterior Sal region, where cells in which the FRT cassette was removed were marked by lacZ staining (Fig. 8a-f). Given that it takes about 20 hr to eliminate the majority of Dpp protein under this condition^{24,41}, wing pouches are devoid of the majority of the Dpp protein derived from the anterior stripe of cells for 28 hr at 29°C until they reach the late third instar stage, which corresponds to a lack of Dpp protein secreted from the main source from early third instar stages onward.”

(line424-437)

“How can the transient *dpp* transcription sustain Dpp target gene expression? One possibility is that persistent low pMad levels are continuously required to repress Brk. Alternatively, Brk repression by early Dpp signaling persists in the later stages without continuous Dpp signaling, for example, by epigenetic regulation or via autoregulation. To distinguish between these possibilities, we applied *tubGal80ts* to genetically remove *tkv* via *tkvHA^{FO}* allele upon FLP expression from the entire A compartment using *ci-Gal4* at different time points. To do so, the larvae were raised at 18 °C until a temperature shift to 29 °C to induce Gal4 expression. Consistent with a role of *tkv* in wing pouch growth, the earlier *tkv* was removed, the smaller the A compartment was (Extended Data Fig. 13). We found that Brk is largely derepressed as early as 16 hr after *tkv* was removed (Extended Data Fig. 13). By considering perdurance activity of Gal80ts for 6 hr after temperature shift⁶², we can estimate that Brk is derepressed within 10 hr at 29 °C after Dpp signaling was lost. Given that the transient *dpp* transcription, which terminated in the early third instar stage, can sustain the anterior pMad signaling and Brk repression at least 28 hr at 29 °C after Dpp protein from the main source is eliminated, these results suggest that persistent weak pMad signaling is continuously required to repress Brk.”

So far it remains unknown how the transient Dpp source can maintain Dpp signaling after *dpp* expression is refined to the anterior stripe of cells. We discussed potential mechanisms to explain the “memory”, including (1) feedback mechanisms to sustain Dpp signaling, and (2) stability of *dpp* mRNA and/or Dpp protein as follows (line504-517);

“It remains unknown how transient *dpp* expression can maintain Dpp signaling in the anterior lateral region (Fig. 8). One possibility is that feedback factors control the duration of Dpp signaling. Various feedback factors have been shown to regulate Dpp signaling. For example, the Drosophila tumor necrosis factor α homolog Eiger and a secreted BMP-binding protein Crossveinless-2 (Cv-2) are positively regulated by Dpp signaling to either positively and negatively influence Dpp signaling in the early embryo, respectively⁶⁶. While JNK is not activated by Eiger in the wild type wing pouch⁶⁷, Cv-2 acts as a positive feedback factor to specify the posterior crossvein during pupal stages⁶⁸. Whether these two factors act during wing pouch patterning and growth remains to be addressed. Pentagon is a secreted feedback factor repressed by Dpp signaling to positively regulate Dpp signaling in the wing disc⁶⁹. Pent is produced in the lateral region of the wing pouch and regulates Dpp signaling and proliferation there. Thus, in addition to its role for scaling, Pent may control the duration of Dpp signaling in the lateral region. Another possibility is that, since it takes quite some time (20-24 hr at 29 °C) to eliminate Dpp protein upon excision of *dpp*^{24,41}, relatively stable *dpp* mRNA and/or protein after the termination of transient *dpp* transcription may contribute to Brk repression (~28 hr at 29 °C). ”

It would help the reader to briefly explain G-trace.

We added a schematic overview of the G-TRACE assay in Fig. 7b for readers to better understand the analysis, and we explained the G-TRACE analysis as follows (line363-365):

“We traced its lineage with G-TRACE analysis, in which RFP expression labels the real-time Gal4-expressing cells and GFP expression labels the entire lineage of the Gal4-expressing cells (Fig. 7b).”

What is the explanation for the sal expression pattern in Fig. 4d,g? I am not sure why it increases laterally.

We agree with the reviewer that the use of Dpp trap appears to lead to the induction of Sal expression in peripheral regions. It has previously been shown that Sal is expressed not only in the medial region but also in the lateral region (Journal of Cell Science 2012 125: 5811-5818; doi: 10.1242/jcs.110569). The same study also showed that the medial Sal expression is Dpp signaling-dependent but that lateral Sal expression is Brk-dependent. Thus, one possibility is that lateral Sal is induced by upregulation of Brk upon Dpp trap expression. However, when we re-analyzed the peripheral region of the control wing disc (in a basal confocal section), we found the lateral Sal expression of the control wing disc is actually comparable to that of the wing disc expressing Dpp trap (Extended Data Fig. 9a, b). Thus, it appears that the lateral Sal expression in the control wing disc (found in a basal confocal section) was simply missed when we focused on the medial Sal expression (found in a more apical confocal section) due to the curved tissue architecture of the relatively large wing disc.

To further follow up on this, as requested by the reviewer, we also removed *dpp* from the entire anterior wing pouch and analyzed Sal expression (Extended Data Fig. 9c-e). Consistent with the above results, we found that medial Sal expression was lost but lateral Sal expression was not significantly changed upon removal of *dpp*. These results were described as follows (line282-293);

“We note that upon Dpp trap expression, Sal expression was lost from the medial region but appeared to be upregulated in the lateral region (Fig. 5d, l, Extended Data Fig. 2o, arrow). It has previously been shown that Sal is expressed not only in the medial region but also in the lateral region⁵⁸. The same study also showed that the medial Sal expression is Dpp signaling-dependent but lateral Sal expression is Brk-dependent. Thus, upregulation of Brk upon Dpp trap expression could cause the lateral Sal upregulation. However, when we focused on the peripheral region of the control wing disc (basal confocal section), we noticed that the lateral Sal expression of the control wing disc was actually comparable to that of the wing disc expressing Dpp trap (Extended Data Fig. 9a, b). Thus, when we focused on the medial Sal expression (apical confocal section), the lateral Sal expression of the control wing disc was simply missed due to the tissue architecture. Consistently, when *dpp* was removed from the entire A compartment using *ci-Gal4* from mid-second instar stage, Sal expression was lost from the medial region but not significantly upregulated in the lateral region (Extended Data Fig. 9c-e).”

The authors suggest that the posterior growth control by anterior Dpp signaling is permissive – can they speculate about how this may occur mechanistically?.

We thank the reviewer for pointing this out. We now discussed the potential mechanisms as follows (line542-559).

“In addition to the Dpp signaling-independent lateral growth, requirement of anterior Dpp signaling for posterior growth (Fig. 5j, Extended Data Fig. S2u) and rescue of posterior growth in *dpp* mutants by anterior Dpp signaling (Fig. 6) indicates that anterior Dpp signaling is non-autonomously involved in posterior lateral growth. We note that similar rescue of posterior growth in *dpp* mutant by anterior Dpp signaling has previously been recognized, but the rescued posterior growth was interpreted as growth of the hinge region, without immunostainings for relevant markers⁵⁹. It remains unknown how anterior Dpp signaling contributes to posterior growth. One possibility is that factors from the A compartment may act non-autonomously to promote posterior growth. Such factors include, but are not limited to, direct downstream factors of Dpp signaling. Given that *5xQE.DsRed* is dependent on Wg^{45,54}, Wg derived from the rescued A compartment may regulate *5xQE.DsRed* expression and growth in the P compartment. Alternatively, mechanical forces may be involved in the non-autonomous growth. It has been proposed that growth factors such as Dpp induce medial growth and subsequently stretch the peripheral regions to induce lateral growth. As the wing disc grows, the peripheral regions in turn compresses the medial region of the wing disc to inhibit the growth of the medial region⁷²⁻⁷⁶. Thus, the growth of the A compartment may stretch the P compartment cells to stimulate their proliferation. It has also been shown that juxtaposition of cells with different Dpp signaling level can induce proliferation non-autonomously but the growth is transient^{26,27}. Therefore, we think it unlikely that the difference of Dpp signaling levels between two compartments can induce sustained growth.”

Reviewer #3 (Remarks to the Author):

This paper from the Affolter lab addresses the long standing problem of understanding the importance of morphogen dispersal in the formation and interpretation of morphogen gradients. This is an issue that the Affolter lab have addressed for many years.

Here the authors develop two different traps. One is the HA trap that presents dispersal of HA-tagged Dpp, but has little effect on signaling by source cells. The other is a Dpp trap made from a DARPIn that both inhibits Dpp dispersal and signaling.

The authors find that posterior patterning and growth require Dpp dispersal, while anterior patterning and growth is largely independent of Dpp dispersal. This conclusion contradicts the conclusions that the same lab reached using morphotrap traps that trap GFP-tagged Dpp.

They go on to show that in the anterior compartment, DPP expression is refined from an initial uniform expression domain, and they also present evidence for another transient source of Dpp expression that is important for signaling in the anterior compartment. Finally they show that lateral wing growth depends neither on Dpp signaling nor dispersal.

In general I think this is a solid piece of work that goes some way to answering the debate in the field about the role of Dpp dispersal. However, there are some key controls missing. I also feel that the paper is written in a cryptic style that makes it difficult to understand for a non-expert as explained below.

We would like to thank the reviewer for the positive comments on our work. We fully agree with the reviewer that the manuscript was written in a way that only specialists working on the topic could understand. In the revised manuscript, we tried our best to explain the logical connections from question/hypothesis to conclusion, experimental setup, results, interpretations, etc. as detailed as possible, so that non-specialists can also follow the text and understand.

1. In Figure 2, the authors haven't formally proved that Dpp dispersal is inhibited in the anterior. This needs to be demonstrated in these particular experiments.

We now provided the results that Dpp dispersal from the anterior stripe of cells is blocked in the anterior compartment (Fig. 2 f, g, j, k).

2. The authors conclude that HA-Dpp can still signal in the source cells, and explain this as a result of the HA antibody binding the HA tag and thus not preventing Dpp binding to its receptors. Why then doesn't HA-antibody bound HA-Dpp signal in the posterior compartment?

It is interesting to speculate how these artificial protein binders affect ligand-receptor binding. However, since we do not know how HA trap-bound HA-Dpp can bind to its receptors on the cell surface, our best guess is that, upon HA-trap binding, the receptor binding domains of HA-Dpp are placed in a conformation such that only the receptors from the producing cells (anterior compartment) can bind.

3. In Figure 5, later wings should also be shown.

Unfortunately, the adult flies from Fig.6b, f in the revised manuscript (Fig. 5 in the previous version) were not recovered at 25 °C. However, when temperature was shifted from 25 °C to 18 °C during mid- to late-third instar stages in order to reduce Gal4 activity during pupal stages, rare survivors from Fig.6b (*dppd8/dppd12, dpp>tkvQD*) were recovered from the pupal cases or managed to hatch although they died shortly after hatching. We provided the adult wing images from these rare survivors (Fig. 6c, d). and described the results as follows(line335-341):

“Unfortunately, the resulting adult flies were not recovered at 25 °C. However, when the temperature was shifted from 25 °C to 18 °C during mid- to late-third instar stages in order to reduce Gal4 activity during pupal stages, rare survivors were recovered from the pupal cases or managed to hatch although they died shortly after hatching. In such survivors, although the anterior wing veins tends to be affected, probably due to continuous TkvQD expression during pupal stages, the anterior growth was largely rescued and the posterior growth was also partially rescued, similar to phenotypes caused by HA trap (Fig. 6c, d).”

4. In Fig 6, the authors show that Dpp expression is initially uniform. Is this also true of PMad signaling at these early stages?

We performed pMad staining in the early wing disc (at 60hr AEL) (Extended Data Fig. 11). Interestingly, we found that pMad was lower in the middle of the wing disc and graded toward the periphery, similar to pMad gradient in the later stages. We described the result as follows (line376-378):

“Despite the initial broad anterior *dpp* expression, we found that pMad signal is low in the middle of the wing disc and graded toward the lateral regions, similar to the pMad gradient in the later stages (Extended Data Fig. 11).”

5. Concerning the clarity of the paper. The authors assume that the readers know where Ptc and Nub are expressed in the wing disc to be able to interpret the experiments. This needs to be made clear. The authors should also explain the G-TRACE analysis.

We now added information about where these Gal4 lines are expressed wherever possible to help the reader’s understanding. We also added a schematic overview of the G-TRACE assay in Fig. 7b for readers to better understand the analysis and explained the G-TRACE analysis as follows (line363-365):

“We traced its lineage with G-TRACE analysis, in which RFP expression labels the real-time Gal4-expressing cells and GFP expression labels the entire lineage of the Gal4-expressing cells (Fig. 7b).”

6. The authors never mention the existence of Gbb, which is another BMP family member expressed much more broadly than Dpp and shown to be important for cell fate specification in the wing and low threshold responses at the edges of the PMad gradient.

The authors need to address the role of Gbb in their experiments. If it is heterodimerized with Dpp, it too could be sequestered by these traps. In addition, Gbb homodimers, which would be unaffected, may contribute to some of the signaling that the authors ascribe to Dpp.

As the reviewer pointed out, it has been proposed that, while Dpp is required for more medial cell fates and growth, another BMP-type ligand, *gbb*, is ubiquitously expressed in the wing pouch and contribute to lateral cell fates (Bangji and Kristi 2006 Dev. Biol.). However, since Gbb signaling is also mediated by Tkv (Shimmi et al. 2005 Dev. Biol.) and the lateral growth takes place independent of Tkv (our results), we think it unlikely that Gbb directly regulates the lateral growth. Nevertheless, it remains an open question how Dpp and Gbb collaborate, either as homodimer or heterodimer, to control wing patterning and growth (Bangji and Kristi, 2006 Dev. Biol.; Matsuda and Shimmi, 2012 Dev. Biol.). For this purpose, we have already started to collaborate with Prof. Kristi Wharton to investigate the role of *gbb*, aiming at trapping Gbb and at dissecting the role of homodimers and heterodimers. To do so, we are currently generating various *gbb* alleles with different tags. We hope to report this study in the near future. We discussed these points as follows (line529-540).

“Consistent with the presence of Dpp signaling-independent lateral growth, it has been shown that lateral wing fates are less sensitive than medial wing fates in various *dpp* mutant alleles⁷⁰. However, this appears counter-intuitive since the lateral region, where morphogen level are low, is expected to be more sensitive than the medial region to a reduction of morphogen levels. It has been proposed that another BMP-type ligand, Glass bottom boat (Gbb), which is expressed ubiquitously in the wing pouch, contributes to lateral cell fates⁷⁰. However, since Gbb signaling is also mediated by Tkv⁷¹ but the lateral growth is independent of Tkv (Fig. 3, 4, 6, Extended Data Fig. 5, 6), we think that the lateral region develops independent of direct Dpp and Gbb signaling. Thus, the lateral region is less sensitive than the medial region in various *dpp* mutant alleles, probably because the lateral region can develop independent of Dpp signaling. What regulates the Dpp signaling-independent lateral wing pouch growth? Given that *5xQE.DsRed* is also dependent on *Wg*^{45,54}, *Wg* may regulate *5xQE.DsRed* expression and growth in the absence of Dpp dispersal.”

7. In Figure 6, what are the panels under d, e, f, and g?

These panels (Fig. 7 e', f', g', h' in the revised manuscript) are magnified images from Fig. 7e, f, g and h. We added dotted lines to help the readers to follow and labeled d', e', f', and g'.

Reviewer #4 (Remarks to the Author):

In this manuscript by Matsuda et al., the authors design synthetic tools to bind and “trap” Dpp, which would prima facie affect its signaling range. They also used a similar tool to trap HA, which affects signaling of HA-tagged Dpp in a different way than the Dpp trap. Using these two newly-developed tools, the authors found that, in the *Drosophila* wing disc, cells in the posterior compartment require the dispersal of Dpp (ie, the Dpp- or HA-trap inhibited patterning and growth), while cells in the anterior compartment do not. The fact that anterior compartment cells can still proliferate semi-normally when Dpp dispersal is blocked was attributed to the lingering effects of an earlier stage in which dpp was expressed throughout the entire A compartment, and thus it was sufficient for Dpp to signal only to the source cells. It remained unclear how the transient exposure to Dpp could be “remembered” by these cells, but it is a very interesting hypothesis that will hopefully get tested in the future. Similarly, it was unclear how the P compartment could still grow a little bit without Dpp ever reaching those cells, but this phenotype required the A compartment to receive Dpp signaling at some level. How such a non-autonomous effect occurred remained a mystery, but again, it is an interesting hypothesis.

The broader impact of this work is the development of the tool, which can then be applied to other morphogen systems, or any other system in which hindering protein movement could be beneficial to the researcher.

That being said, there are several outstanding questions that the authors must address before the manuscript can be considered further. In general the major problem is the paper is written so that only a specialist can read it. The authors often skip explanatory details of what they did, perhaps to economize space. I do not fault the authors for this; it is easy to forget how much others don't know. However, this economy of space results in many experimental details being left out (they are not in the Figure Legends, the Methods, or Supplementary Materials either). Oftentimes, the experiments have complex genotypes (found in the Methods, which was well done) or manipulations, and as such, they are difficult to interpret without more detailed explanations. To go along with that point, the flow of logic is also left out in many cases. In at least one instance, the lack of detail makes it difficult to inferring the logical connection between the experimental set up, the observed data, and the authors' conclusion. The authors should keep in mind that Dpp signaling in the wing disc is a system with a host of complicated (and frequently conflicting) literature. I urge the authors to consider this and flesh out the details of the experimental set ups and the logical connections from question/hypothesis to conclusion for each experiment. It is difficult to convey a positive tone in a critical review, but I honestly hope the authors see these criticisms as encouragements to make their manuscript more clear so that readers other than wing disc specialists can understand it.

We fully agree with the reviewer that the manuscript was written in a way that only specialists working on the topic could understand, and we would like to thank the reviewer for the constructive criticisms to improve our manuscript. In the revised manuscript, we tried our best to explain the logical connections from question/hypothesis to conclusion, experimental setup, results, interpretations, etc. as detailed as possible, so that non-specialists can also follow the text and understand.

First, the growth phenotype of the posterior compartment is difficult to convey. Without knowledge of the literature, which the authors do not spell out concretely, someone reading this paper would be very confused as to whether the posterior growth was “inhibited” or “substantial”. Thus, the authors need to pay very careful attention to how they describe the phenotype. For example, on lines 132 and 138, the authors use the word “substantial,” but that is quite vague. It makes it sound like there is a “lot” of growth, when in fact there is less growth than normal. A more precise word needs to be used. As another example, between Lines 140 and 149, the authors first say the growth defect is severe, (and that Dpp dispersal is critical for growth) to saying the posterior compartment does grow, to saying that the growth contradicts previous reports (what reports? What did these reports conclude?).

We agree that we should pay careful attention to how we describe the phenotype in Fig.3 in the revised manuscript (=Fig.2 in the previous version). We now described the following two points separately; (1) Upon HA trap expression, the posterior growth was “inhibited” compared with the control, and (2) Despite the growth defects, the posterior compartment did still grow “substantially” compared with the more severe *dpp* mutant phenotypes.

We first described the “inhibition” of growth as follows (line158-164):

“The posterior wing pouch growth was also affected as revealed by the expression of an intervein marker DSRF and a wing pouch marker *5xQE.DsRed*⁴⁵ (Fig. 3b arrow, 3i). Interestingly, although *5xQE.DsRed* contains five copies of the 806 bp Quadrant Enhancer (QE) of the wing master gene *vg* containing a Mad binding site and is therefore thought to be directly regulated by Dpp signaling^{46,47}, *5xQE.DsRed* remained expressed in the P compartment without detectable Dpp signaling (Fig. 3b arrow).”

We then described the “substantial” growth in the following paragraph. We avoided to use “substantial” in the text but precisely described the proportion of the posterior compartment (40%) that still grew upon HA trap expression. Furthermore, we briefly explained why the posterior growth without detectable Dpp signaling upon HA trap expression contradicts results presented in a previous report (Burke 1996) (line182-187);

“A critical role of Dpp dispersal for posterior patterning and growth is consistent with a role of Dpp as a morphogen. However, the overall phenotypes caused by HA trap was surprisingly mild when compared to the phenotypes seen in *dpp* mutants (Fig. 6a, c). Given the requirement of Dpp signaling for cell proliferation and survival in the entire wing pouch⁴⁸, it was surprising that about 40% of the posterior wing pouch was able to grow and differentiate into adult wing tissue without detectable Dpp signaling (Fig. 3d, i, j).”

Sentences such as (Line 117), “...clonal accumulation of Ollas-HA-Dpp in HA trap clones in the receiving cells was undetectable upon HA trap expression with *ptc*-Gal4 (Fig. 1k-n, arrow), indicating that the HA trap can block HA-Dpp dispersal efficiently,” do not sufficiently explain what is happening in the experiment, nor what to expect from the experiment. What do the authors mean by “receiving cells”? What is the staining in the middle panel of Fig. 1k? The label just says “HA trap.” Is it an anti-cherry staining?

More on the above sentence: the authors simply call them HA trap clones. I suspect the average reader would assume most of the disc is hetero (or hemi)-zygous for HA trap and the clones would be homozygous for the HA trap (with twin spots of cells lacking HA trap). There is very little in the manuscript at that point to help the reader ascertain what is meant by “HA trap clones.” There is a lot happening in these experiments, and a single sentence is not enough to explain the experimental set up, the flow of thought, nor the logic of the conclusion.

We are sorry for this confusion. We rewrote the entire sentence to explain the experiment and results as follows (line 133-141);

“To test if HA trap can trap Ollas-HA-Dpp outside the anterior stripe of cells, clones of cells expressing *Gal4* were randomly induced by heat-shock inducible FLP to express HA trap under UAS control. We found that Ollas-HA-Dpp accumulated in clones of cells expressing HA trap induced outside the main *dpp* source cells in both compartments (Fig. 2f-i, arrow). If HA trap can efficiently trap Ollas-HA-Dpp in the source cells, the clonal Ollas-HA-Dpp accumulation should be blocked upon HA trap expression in the source cells. Indeed, we found that clonal Ollas-HA-Dpp accumulation in both compartments was drastically reduced upon HA trap expression using *ptc-Gal4* (Fig. 2j-k, arrow), indicating that the HA trap can block HA-Dpp dispersal efficiently.”

-What do the authors mean by “receiving cells”?

We removed “receiving cells”.

-What is the staining in the middle panel of Fig. 1k? The label just says “HA trap.” Is it an anti-cherry staining?

The middle panel of Fig. 2f, 2h in the revised version represents the mCherry fluorescent signal derived from HA trap.

-There is very little in the manuscript at that point to help the reader ascertain what is meant by “HA trap clones.”

-We replaced “HA trap clone” with “clones of cells expressing HA trap”.

Another example (Line 149), “When *tkva12* clones characterized as a null allele 42, 43 were induced in wing discs expressing HA trap with *ptc-Gal4*, *tkva12* clones survived and expressed the 5xQE.DsRed reporter in the anterior lateral regions as well as in the entire posterior region, even next to the source cells (Fig. 3a), indicating that leakage is negligible.” Again, there is a lot going on in this experiment, and a single sentence is not sufficient to explain it. Why should a clone homozygous for a null allele for *tkv* show that there is no leakage of HA-Dpp out of the HA trap? The best I could come up with that matches the authors’ conclusion is that, when Dpp dispersal is blocked by the HA trap, then clones of *tkv* survive because they are not outcompeted by other cells in the P compartment which would have been growing had those other cells received the leaked Dpp. That’s why in the wildtype disc, only lateral clones are found (no medial). This seems like a convoluted logic, but if that’s correct, then it should be spelled out explicitly. If it’s not correct, then the actual flow of logic should be spelled out.

We are sorry for the confusion. In addition to our results showing that the HA trap is very

efficient in trapping Ollas-HA-Dpp or HA-Dpp (Fig. 2, 3 in the revised manuscript), we further showed that trapping HA-Dpp using HA trap using both *nub-Gal4* and *ptc-Gal4* did not enhance the defects (Extended Data Fig.3 in the revised manuscript). Although these results show that HA trap is efficient in trapping Dpp, as the reviewer pointed out, there might be slight leakage of Dpp and the leaked Dpp may contribute to *5xQE.DsRed* expression and growth in the posterior compartment seen upon HA trap expression.

We are fully aware of the limitation of our approaches and do not want to conclude that there is no leakage even if the affinity of HA trap is high. Thus, the purpose of Fig.4 in the revised manuscript (=Fig.3 in the previous version) was actually not to demonstrate that HA trap is fully effective, but rather to ask whether leakage of Dpp from HA trap, if it would occur, can explain the *5xQE.DsRed* expression and growth in the posterior compartment seen upon HA trap expression.

To test this, we removed *tkv* in the wing disc expressing HA trap, since *tkv* is essential for Dpp signal transduction and for the survival of wing pouch cells. The results suggest that posterior growth seen upon HA trap expression is *Tkv*-independent, indicating that, even if there were leakage of Dpp, this would not account for the posterior growth or *5xQE.DsRed* expression. We described these results as follows (line189-200):

“We therefore tested whether the posterior growth and *5xQE.DsRed* expression seen upon HA trap expression is caused by low levels of HA-Dpp leaking from the HA trap expressed in the source. In this case, the posterior growth and *5xQE.DsRed* expression seen upon HA trap expression should be dependent on *tkv*, an essential receptor for Dpp signaling. To test this, mutant clones of *tkv^{Δ12}* (characterized as a null allele^{49,50}) were induced in wing discs expressing HA trap with *ptc-Gal4* between mid-second and beginning of third instar stages and analyzed in the late third instar stage. We found that *tkv^{Δ12}* clones survived and expressed the *5xQE.DsRed* reporter in the anterior lateral regions as well as in the entire posterior region. We also noticed that *tkv^{Δ12}* clones survived and expressed the *5xQE.DsRed* reporter even next to the source cells in the P compartment (Fig. 4a). These results indicate that the lateral growth and *5xQE.DsRed* expression seen upon HA trap expression is independent of Dpp signaling, and not caused by a leakage of HA-Dpp from the HA trap, even if such leakage would occur.”

We discuss the possibility that *tkv* mutant clones are outcompeted by surrounding wildtype cells as a possible explanation of why *tkv*-independent growth had been previously missed as follows (line214-220);

“How can Dpp signaling-independent wing pouch growth and *5xQE.DsRed* expression be reconciled with a critical role of Dpp signaling for the entire wing pouch growth⁴⁸? First, *tkv* clones generated in the developing wing pouch have been shown to be eliminated by apoptosis or extrusion and do not survive in the adult wing^{48,51}. However, *tkv* clones survive better in the P compartment where Dpp signaling is blocked by HA trap (Fig. 4a) and in the lateral region of wild type wing disc where Dpp signaling is generally low (Fig. 4b-d). This raises a possibility that *tkv* clones are eliminated when surrounded by wild type cells, even if *tkv* clones could grow and survive to a certain extent.”

Another example (Line 155), “In rare cases, even medial *tkv* null clones survived and expressed *5xQE.DsRed* (Fig. 3d), indicating that the elimination of *tkv* mutant clones masked Dpp signaling-independent *5xQE.DsRed* expression in previous studies.” Again, it is unclear how the conclusion follows from the observation. There is also no explanation of what the previous studies found, how those conflict with the authors’ data, or a logical flow of thought by which the authors resolve the conflict. Further, there is no citation of said previous studies for the readers to try to figure this out themselves.

We are sorry for the confusion. We re-wrote the whole sentence in the revised manuscript as follows (line209-212);

“We also found that, while most often eliminated, medial *tkv* null clones survived and expressed *5xQE.DsRed* in rare cases (Fig. 4d), indicating that Dpp signaling is dispensable for *5xQE.DsRed* expression also in the medial region, but medial cells lacking Dpp signaling are normally eliminated⁴⁸.”

Another example (the very following sentence), “Consistently, upon genetic removal of *dpp* from the entire A compartment as early as the beginning of second instar stage, when the wing pouch is defined, *5xQE.DsRed* remained expressed despite severe growth defects (Fig. 3e-h).” An entire series of panels, including the experimental set up, what to expect, and why, is compressed into a sentence.

We rewrote the whole sentence in the revised manuscript to better explain the motivation of the experiment, experimental setup, results, and interpretation as follows (line214-242). Furthermore, in addition to genetic removal of *dpp* from the entire A compartment, we also provide the results removing *tkv* from each compartment (Extended Data Fig. 5, 6).

(line214-242)

“How can Dpp signaling-independent wing pouch growth and *5xQE.DsRed* expression be reconciled with a critical role of Dpp signaling for the entire wing pouch growth⁴⁸? First, *tkv* clones generated in the developing wing pouch have been shown to be eliminated by apoptosis or extrusion and do not survive in the adult wing^{48,51}. However, *tkv* clones survive better in the P compartment where Dpp signaling is blocked by HA trap (Fig. 4a) and in the lateral region of wild type wing disc where Dpp signaling is generally low (Fig. 4b-d). This raises a possibility that *tkv* clones are eliminated when surrounded by wild type cells, even if *tkv* clones could grow and survive to a certain extent. Second, wing pouch and *5xQE.DsRed* expression were completely lost in *dpp* mutants (Fig. 6a, Extended Data Fig. 7a). It has been shown that initial wing pouch specification is mediated by Dpp derived from the peripodial membrane, which covers the developing wing pouch, and this early *dpp* expression in the peripodial membrane is lost in *dpp* disc alleles⁵². Thus, wing pouch and *5xQE.DsRed* expression could be lost in *dpp* disc alleles due to failure of initial specification of the wing disc and subsequent elimination of cells.

To minimize these potential problems, we applied Gal80ts to conditionally remove *dpp* from the entire A compartment using *ci-Gal4*. At the permissive temperature of 18°C, Gal80ts actively represses Gal4 activity. At restrictive temperature of 29°C, Gal80ts can no longer

block Gal4 activity, thus Gal4 can be temporally activated using temperature shifts. Upon FLP expression, *dpp* was removed by FLP/FRT mediated excision via *dpp^{FO}* allele²⁴, in which a FRT cassette was inserted into the *dpp* locus. To remove *dpp* from the beginning of second instar stage when the wing pouch is specified, the larvae were raised at 18°C for 4 days and then shifted to 29°C. By removing *dpp* from the entire A compartment using *ci-Gal4* under this condition, we found that *5xQE.DsRed* remained expressed despite severe growth defects in the late third instar stage (Fig. 4e-h). Similarly, genetic removal of *tkv* via *tkvHA^{FO}* from the A compartment using *ci-Gal4* or from the P compartment using *hh-Gal4* from the second instar stage revealed that, despite severe growth defects, *5xQE.DsRed* remained expressed in each compartment lacking *tkv* (Extended Data Fig. 5). Surprisingly, similar results were obtained even when *tkv* was removed from the entire P compartment using *hh-Gal4* from the embryonic stages without Gal80ts (Extended Data Fig. 6). These results further support the presence of Dpp signaling-independent *5xQE.DsRed* expression and wing pouch growth.”

Another example (Line 221): “we expressed HA trap with *ptc-Gal4* at defined time points using *tubGal80ts*. Upon HA trap expression from the mid-second instar stage...” Temperature shift experiments could be quite complicated, depending on the set up. The authors did not give any indication of what temperatures the larvae were raised at, or when temperature shifts happened. Were the larvae raised at the permissive temperature until a permanent shift to the restrictive, or was there only a short period of the restrictive before shifting back? When did the temperature shift happen? How did they synchronize the ages of the larvae?

We now added schematic views of the temperature shift experiment in Extended Data Fig.12 to help readers understand the experimental setup, and explained the temperature shift experiment and detail results in the revised manuscript as follows (line384-395);

“To avoid HA trap expression in the entire A compartment, we applied *tubGal80ts* to express HA trap using *ptc-Gal4* at defined time points in the A compartment. To do so, the larvae were raised at 18 °C until a temperature shift to 29 °C to induce *Gal4* expression (Extended Data Fig. 12a). Upon HA trap expression from the mid-second instar stage, the lineage of *ptc-Gal4* covered at most the anterior Sal domain (Fig. 8b), which corresponds to the region between L2 and L4 in the adult wing. We found that the later the temperature shift, the milder the posterior growth defects (Extended Data Fig. 12a, b). In contrast, the A compartment size (between L1 and L4) remained rather normal, independent of the timing of the temperature shift (Extended Data Fig. 12a, c). Interestingly, the size of peripheral regions (between L1 and L2) and the specification of L2 were not affected, independent of the timing of the temperature shift (Extended Data Fig. 12a, d). These results are consistent with a role of an anterior lateral *dpp* source for patterning and/or growth in the anterior lateral regions.”

-We explained how to synchronize the ages of the larvae in the material and method section as follows (line705-706);

“To minimize variations, embryos were staged by collecting eggs for 2-4hrs.”

Line 227: Same as above – the authors did not explain the temperature shift experiment. When did the flip-out occur?

We explained the detail temperature shift experiment in the revised manuscript as follows (line397-405);

“To directly test this, we applied Gal80ts to genetically remove *dpp* via *dpp*^{FO} allele upon FLP expression using *ptc-Gal4* from the mid-second instar stage. To do so, the larvae were raised at 18 °C for 5 days before a temperature shift to 29 °C and were then dissected 48 hr later. In this setup, *dpp* was removed approximately from the anterior Sal region, where cells in which the FRT cassette was removed were marked by lacZ staining (Fig. 8a-f). Given that it takes about 20 hr to eliminate the majority of Dpp protein under this condition^{24,41}, wing pouches are devoid of the majority of the Dpp protein derived from the anterior stripe of cells for 28 hr at 29°C until they reach the late third instar stage, which corresponds to a lack of Dpp protein secreted from the main source from early third instar stages onward.”

Line 233: The authors abruptly state “By removing *dpp* from the entire A compartment” without any explanation that, to do this, they used a different Gal4 line. This should be stated explicitly (“*ci*” doesn’t appear anywhere in the paragraph), because this is embedded in a paragraph that begins with a sentence saying the authors used *ptc-Gal4*. Nor is there any explicit connection made between *ci* and “entire A compartment.” It is not enough to assume the readers will figure this out on their own from reading the Figure legend.

We thank the reviewer for pointing this out. We now explained explicitly that we used *ci-Gal4* to remove *dpp* from the entire A compartment as follows (line414-417);

“To test if the remaining anterior Dpp signaling activity is due to Dpp produced outside the anterior Sal domain, we then compared removal of *dpp* from the anterior Sal domain using *ptc-Gal4* (the same setup above) with removal of *dpp* from the entire A compartment using *ci-Gal4* from the mid-second instar stage.”

Line 238: “Upon genetic removal of *tkv*, *Brk* was quickly de-repressed (Extended Data Fig. 6), indicating that the “memory” of an earlier signal is not...” There is very little explanation of what the authors did in this experiment. The genotype is given in the Methods section, but that is not enough. The Extended Data Fig. 6 legend is sparse on detail, and is also confusing as stated. Are all discs the same age, and the only difference is how long *tkv* had been removed? Why does the 5th disc look smaller than the rest is it younger? Is it a growth phenotype? Is the scale bar the same in all images? Furthermore, what do the authors mean by “quickly?” This is a vague word. How long before *Brk* was de-repressed? How does that compare to the expected time and/or time scales of other events in this system?

We are sorry for the poor explanation. We now describe the details of temperature shift experiments and discuss why we speculate that the epigenetic regulation or autoregulation of *Brk* expression, even if it exists, unlikely account for all the persistent *brk* repression as follows (line424-437);

“How can the transient *dpp* transcription sustain Dpp target gene expression? One possibility is that persistent low pMad levels are continuously required to repress Brk. Alternatively, Brk repression by early Dpp signaling persists in the later stages without continuous Dpp signaling, for example, by epigenetic regulation or via autoregulation. To distinguish between these possibilities, we applied *tubGal80ts* to genetically remove *tkv* via *tkvHA^{FO}* allele upon FLP expression from the entire A compartment using *ci-Gal4* at different time points. To do so, the larvae were raised at 18 °C until a temperature shift to 29 °C to induce Gal4 expression. Consistent with a role of *tkv* in wing pouch growth, the earlier *tkv* was removed, the smaller the A compartment was (Extended Data Fig. 13). We found that Brk is largely derepressed as early as 16 hr after *tkv* was removed (Extended Data Fig. 13). By considering perdurance activity of Gal80ts for 6 hr after temperature shift⁵⁸, we can estimate that Brk is derepressed within 10 hr at 29 °C after Dpp signaling was lost. Given that the transient *dpp* transcription, which terminated in the early third instar stage, can sustain the anterior pMad signaling and Brk repression at least 28 hr at 29 °C after Dpp protein from the main source is eliminated, these results suggest that persistent weak pMad signaling is continuously required to repress Brk.”

-The Extended Data Fig. 6 legend is sparse on detail, and is also confusing as stated. We described more details in Extended Data Fig.13 in the revised manuscript (=Extended Data Fig.6 in the previous version). We also provided schematic overviews of each temperature shift experiment to help readers understand.

-Are all discs the same age, and the only difference is how long *tkv* had been removed?

All the discs are the same age and the only difference is how long *tkv* had been removed. We described this in the figure legend of Extended Data Fig.13.

-Why does the 5th disc look smaller than the rest is it younger? Is it a growth phenotype?

-Consistent with a role of *tkv* for wing pouch growth, the 5th wing disc looks smaller since *tkv* is removed earlier than the other discs in Extended Data Fig.13. This was explained in the results.

-Is the scale bar the same in all images?

-The scale bar is the same in all images (50um).

-Furthermore, what do the authors mean by “quickly?” This is a vague word.

We avoided to use “quickly” and use the precise time.

-How long before Brk was de-repressed? How does that compare to the expected time and/or time scales of other events in this system?

Concerning the de-repression of Brk and epigenetic regulation or autoregulation of Brk, we consider two possibilities. One possibility is that persistent low pMad levels are continuously required to repress Brk. Alternatively, Brk repression by early Dpp signaling persists in the later stages without continuous Dpp signaling, for example, by epigenetic regulation or autoregulation.

We found de-repression of Brk as early as 16hr at 29°C upon *tkv* removal (Extended Data Fig. 13). By considering perdurance activity of Gal80ts for 6hr at 29°C after temperature shift (Bergantinos, C et al., 2010), we can estimate that Brk is derepressed within 10hr at 29°C if Dpp signaling is lost.

In contrast, pMad signaling and Brk repression were still visible in the anterior lateral region 48hr at 29°C after *dpp* was removed from the anterior stripe of cells (Fig. 8). Even if we consider that it takes about 20hr at 29°C to eliminate the majority of the Dpp protein after genetic removal of *dpp* (Akiyama and Gibson., 2015)(Matsuda and Affolter, 2017), the pMad signaling and Brk repression mediated by the transient *dpp* transcription persist at least for 28hr at 29°C. Thus, we think that epigenetic regulation or autoregulation of Brk, even if it exists, unlikely explains the persistent Brk repression.

Line 306: “ubiquitous blocking of Dpp dispersal by HA trap during development caused lethality (data not shown).” This is an interesting claim, but it needs further explanation. Are the authors referring to an HA trap that was expressed starting in the early embryonic stages? Is that what they mean by “ubiquitous” and “during development?” This statement is too vague, and genotypes and other experimental details are not given. Perhaps that was the point? Perhaps, since the authors wanted put this claim into the Discussion section rather than the Results section, the thinking is it would not need to be fully explained? But that is not the case. In general, new claims about data should not be introduced in the Discussion section. If this is a phenotype they noticed upon early iterations of creation of the HA trap, then it could go in one of the first subsections of the Results section. Also, I am not sure if “data not shown” is acceptable for a Nature Communications paper. If the authors think it is not a solid enough result to go into the Results, then neither should it appear in the Discussion.

We thank the reviewer for pointing this out. The experimental setup was HA trap expression using *actin-Gal4* or *actin-LexA* in HA-Dpp homozygotes. Although it is interesting to follow this up in the future, we decided not to include it in the revised manuscript since the result is not critical for the current story.

Minor:

- It is unclear from the abstract or intro why HA trap should affect Dpp.

We agree with the reviewer that it is not clear from the abstract or introduction why HA trap should affect Dpp. We rewrote the abstract and introduction to explain why HA trap should affect Dpp as follows (abstract, line23-36; intro, line84-85):

(abstract line23-26)

“How morphogen gradients control patterning and growth in developing tissues remains largely unknown due to lack of tools manipulating morphogen gradients. Here, we generate two membrane-tethered protein binders that manipulate different aspects of Decapentaplegic (Dpp), a morphogen required for overall patterning and growth of the *Drosophila* wing. “HA trap” is based on a single-chain variable fragment (scFv) against the HA tag that traps HA-Dpp to mainly block its dispersal, while “Dpp trap” is based on a Designed Ankyrin Repeat Protein (DARPin) against Dpp that traps Dpp to block both its dispersal and signaling. Using these tools, we found that, while posterior patterning and growth require Dpp dispersal, anterior patterning and growth largely proceed without Dpp dispersal. We show that *dpp* transcriptional refinement from an initially uniform to a

localized expression and persistent signaling in transient *dpp* source cells render the anterior compartment robust against the absence of Dpp dispersal. Furthermore, despite a critical requirement of *dpp* for the overall wing growth, neither Dpp dispersal nor direct signaling is critical for lateral wing growth after wing pouch specification. These results challenge the long-standing dogma that Dpp dispersal is strictly required to control and coordinate overall wing patterning and growth.”

(introduction, line84-85)

“One is “HA trap” based on anti-HA scFv that traps HA-Dpp through the HA tag to mainly block Dpp dispersal”

- The expected function of OLLAS is not explained.

We explained the reason of having Ollas as follows (line123-126):

“While we attempted to visualize extracellular HA-Dpp distribution upon HA trap expression, we noticed that the HA tag can no longer be used for immunostaining when bound to HA trap. We therefore additionally inserted an Ollas tag to generate a functional *Ollas-HA-dpp* allele in order to visualize the extracellular Dpp distribution using the antibody against the Ollas tag.”

- Line 117, “Furthermore, clonal accumulation of Ollas-HA-Dpp in HA trap clones in the receiving cells was undetectable...” This does not seem to be the case. There is clear, detectable green fluorescence in the clone that the arrow is pointing to. Unless the authors mean something different from what I am inferring, in which case the flow of logic should be explained more clearly.

We thank the reviewer for pointing this out. The confusion is mainly because the arrows were not explained well. We now carefully described arrows in Fig.2 in the revised version as follows (Line133-141):

“To test if HA trap can trap Ollas-HA-Dpp outside the anterior stripe of cells, clones of cells expressing *Gal4* were randomly induced by heat-shock inducible FLP to express HA trap under UAS control. We found that Ollas-HA-Dpp accumulated in clones of cells expressing HA trap induced outside the main *dpp* source cells in both compartments (Fig. 2f-i, arrow). If HA trap can efficiently trap Ollas-HA-Dpp in the source cells, the clonal Ollas-HA-Dpp accumulation should be blocked upon HA trap expression in the source cells. Indeed, we found that clonal Ollas-HA-Dpp accumulation in both compartments was drastically reduced upon HA trap expression using *ptc-Gal4* (Fig. 2j-k, arrow), indicating that the HA trap can block HA-Dpp dispersal efficiently.”

- Fig. 1 legend: arrows should be explained in the legend.

We now explained the arrows in the Fig.2 legend in the revised version (=Fig.1 in the previous version).

- The abbreviation “ExHA” and similar are not explained.

The abbreviation of “ExHA” is “Extracellular α -HA staining”. This is now added in Fig.1e in the revised version and the corresponding figure legend. Similarly, the abbreviation of “Ex

Ollas” is “Extracellular α -Ollas staining”. This is now added in Fig.2 legend in the revised version.

- The specific method that generates an anti-ExHA (and similar), and what the results would mean, are not explained.

There is no specific antibody that visualizes only the extracellular proteins. Instead, there is a protocol to visualize only the extracellular pool of proteins. In the extracellular immunostaining protocol, tissues are not permeabilized before primary antibody incubation so that antibodies can access only the extracellular antigens. We described this in the result and in the method section as follows (line109-114, line 701-703):

(line109-114)

“Immunostainings for the HA-tag including permeabilization steps showed HA-Dpp expression in an anterior stripe of cells along the A-P compartment boundary in the late third instar wing disc (Fig. 1d). In contrast, immunostainings for the HA-tag without permeabilization, which allows antibodies to access only the extracellular antigens, revealed that a shallow extracellular HA-Dpp gradient overlapped with the gradient of phosphorylated Mad (pMad), a downstream transcription factor of Dpp signaling (Fig. 1e).”

(line701-703)

“For extracellular staining, dissected larvae were incubated with primary antibodies in M3 medium for 1 hr on ice before fixation to allow antibodies to access only the extracellular antigens.”

- Fig 1o: it is not clear what is being merged in the fourth panel. Panel 1 is green, Panel 2 is magenta, and if those two were merged, it would produce white. However, Panel 3 itself is white, so it is unclear what white means in Panel 4. On the clones, white appears to be the merge of green and magenta, but elsewhere, white appears to be pMad. How would one distinguish?

We thank the reviewer for pointing out this. We now changed the color of the Panel 4 to blue in Fig.2I in the revised manuscript.

- Sentence that begins on line 128 with “Upon HA trap expression...” is ambiguous. Does the whole sentence refer to the P compartment, or just the Brk upregulation? The fact that “in the P compartment” comes after the “and” makes it difficult to know. That prepositional phrase should come earlier in the sentence to relieve the ambiguity. Or, if it’s the latter, then “in the whole pouch” should be applied to the statement about pMad, Sal, and Omb. Sorry for the confusion. We now added “in the P compartment” earlier in the sentence (line155-158) as follows:

“Upon HA trap expression in the anterior stripe of cells using *ptc-Gal4*, pMad, Sal, and Omb expression were undetectable in the P compartment and Brk was also upregulated in the P compartment (Fig. 3b, e, f, g, h), indicating that HA trap efficiently blocked HA-Dpp dispersal from source cells and interfered with patterning.”

- Sentence that starts on line 149 with “When tkva12 clones characterized” is very long and awkward. Other sentences in this section also fall prey to the same difficulty, likely because the authors are trying to squeeze so much information into a single sentence (see my “major criticisms” above).

We agree with the reviewer that the sentences are very long and awkward. We rewrote the whole sentence as follows (line189-200):

“We therefore tested whether the posterior growth and *5xQE.DsRed* expression seen upon HA trap expression is caused by low levels of HA-Dpp leaking from the HA trap expressed in the source. In this case, the posterior growth and *5xQE.DsRed* expression seen upon HA trap expression should be dependent on *tkv*, an essential receptor for Dpp signaling. To test this, mutant clones of *tkv*^{Δ12} (characterized as a null allele^{49,50}) were induced in wing discs expressing HA trap with *ptc-Gal4* between mid-second and beginning of third instar stages and analyzed in the late third instar stage. We found that *tkv*^{Δ12} clones survived and expressed the *5xQE.DsRed* reporter in the anterior lateral regions as well as in the entire posterior region. We also noticed that *tkv*^{Δ12} clones survived and expressed the *5xQE.DsRed* reporter even next to the source cells in the P compartment (Fig. 4a). These results indicate that the lateral growth and *5xQE.DsRed* expression seen upon HA trap expression is independent of Dpp signaling, and not caused by a leakage of HA-Dpp from the HA trap, even if such leakage would occur.”

- Line 176: “Interestingly, anterior Dpp trap expression caused ... anterior growth defects than HA trap.” The authors should edit for grammar. As written, both parts of the sentence should work within the sentence as a whole. That is, when you remove “more severe posterior growth defects as well as,” the sentence should still make sense. Perhaps this could be solved just by putting “more” in front of anterior.

We reformulated the sentence “Interestingly, anterior Dpp trap expression caused more severe posterior growth defects as well as anterior growth defects than HA trap” to focus on the effect of anterior Dpp trap on the posterior compartment as follows (Line304-307):

“Interestingly, despite the slight leakage of Dpp from the Dpp trap (Extended Data Fig. 10), anterior Dpp trap expression caused more severe posterior growth defects than HA trap (Fig. 5j, Extended Data Fig. 2u), indicating that anterior Dpp signaling is non-autonomously required for the posterior growth.”

- Line 178: “This non-autonomous effect was hardly seen by simply removing tkv from the entire A compartment, probably because Dpp can still disperse and control patterning and growth under this condition (data not shown).” There are several issues with this sentence. First, it is unclear what the authors mean by the vague statement “...effect was hardly seen...” Second, it would be nice if the authors could comment as to how Dpp signaling could take place without tkv. Third, and I am not sure about this, but my guess is that Nature frowns upon “data not shown.”

We are sorry for the confusion. Based on the comparison between HA trap and Dpp trap phenotypes, our results suggest that anterior Dpp signaling is non-autonomously required

for the posterior growth. However, as we now provide results (Extended Data Fig. 5c, d), removal of *tkv* from the A compartment using *ci-Gal4* did not interfere with posterior growth as severe as Dpp trap, even when the anterior signaling was eliminated (Extended Data Fig. 5c, d). We speculate that this is because anterior Dpp can still disperse into the P compartment to control posterior growth in this situation. We now restructured the sentence in line307-311 as follows:

“We note that, even though the anterior Dpp signaling was eliminated, genetic removal of *tkv* from the A compartment using *ci-Gal4* did not interfere with posterior growth as severe as Dpp trap (Extended Data Fig. 5c, d), probably because Dpp secreted from the A compartment can disperse to control posterior growth.”

-Dpp signaling cannot be activated without Tkv. When *tkv* was removed from the anterior compartment, Dpp signaling was lost in the anterior compartment. However, under this condition, the size of the P compartment was not strongly affected most likely because Dpp is still produced and secreted in source cells and can still disperse from the A compartment to the P compartment.

- Line 180 (next sentence): the authors should probably temper their claim a bit. They should include a qualifier, such as “likely”.

We added “likely” in the line303-304 in the revised manuscript as follows:

“Thus, the severe phenotypes caused by Dpp trap are likely because Dpp trap blocks Dpp signaling more efficiently than HA trap.”

- Line 184: “Taken together...” It would be nice if the authors could comment on how signaling in just the source cells, without Dpp dispersal, could affect growth in the rest of the pouch.

We commented on how signaling in just the source cells, without Dpp dispersal, could affect growth in the rest of the pouch in the following paragraph (line315-325) as follows:

“Our results so far suggest that, while the requirement for Dpp dispersal is relatively minor and asymmetric, Dpp signaling in the source cells is critically required for the majority of patterning and growth seen upon blocking Dpp dispersal. This raises the question of how cell-autonomous Dpp signaling in the source cells can control patterning and growth outside the anterior stripe of cells, the main *dpp* source cells. There are a couple of possible scenarios; if *dpp* expression is successively restricted to the anterior stripe of cells during development, the anterior stripe of cells may retain and deliver the earlier Dpp signaling to the peripheral region after they leave from the anterior stripe of cells via proliferation⁵⁹, or downstream factor(s) of Dpp signaling in the anterior stripe of cells may act non-autonomously to control patterning and growth outside the stripe of cells. Alternatively, *dpp* expression may not be restricted to the anterior stripe of cells in the early stages, similar to what has been shown in the case of *wg*¹².”

- Line 193: Should be “depend”.

We corrected the mistake in line342 in the revised manuscript.

- Line 194: It is nice to see this question asked, but it should probably come earlier in the manuscript (see “Line 184” comment).

We asked the question also in the beginning of the section as follows (line 317-319).

“This raises the question of how cell-autonomous Dpp signaling in the source cells can control patterning and growth outside the anterior stripe of cells, the main *dpp* source cells.”

- Lines 195 – 201: It would be nice if the authors could comment on the anterior compartment’s phenotype in which the *dpp-Gal4* cells grow to engulf the entire A compartment. For example, Fig 5d shows that pMad, Sal, and Omb expand to the entire A compartment, and Brk is completely lacking there.

We now described the phenotypes of the anterior compartment in Fig.6f in the revised manuscript (=Fig.5d in the previous version) as follows (line349-351):

“Under this condition, pMad, Sal, and Omb were uniformly upregulated in the A compartment, and Brk was completely lost in the entire A compartment (Fig. 6f), indicating that *dpp-Gal4* has been expressed in the entire A compartment⁵⁶.”

- Line 205: the above point is slightly resolved here (incl. ref 47). However, this point should come in the previous section of the paper, when the reader would encounter the phenotype (referenced in Fig. 5d). The reader should not have to wait and wonder. Even so, this section with the destabilized GFP (Fig 6) is quite elegant and clearly proves the authors’ point.

We now moved “the lineage of *dpp-Gal4* was uniform in the A compartment.” to line349-351, where the phenotype was described.

“Under this condition, pMad, Sal, and Omb were uniformly upregulated in the A compartment, and Brk was completely lost in the entire A compartment (Fig. 6f), indicating that *dpp-Gal4* has been expressed in the entire A compartment⁶⁰.”

- Same line: “lineages...were uniform,” perhaps? “Lineages” is a plural noun, correct? We replaced “lineages” with “lineage” wherever possible.

- Fig. 6 Legend: I am pretty sure the legend does not correspond to the figure. Sorry for this mistake. We now corrected the legend in Fig.7 (Fig.6 in the previous version) in the revised manuscript.

- Fig. 7 Legend: There needs to be a more clear explanation of the difference between Fig. 7d and 7g, as well as Fig. 7f vs 7h.

The genotypes of the wing discs in Fig. 8d, g (Fig.7d, g in the previous version) and in Fig. 8f, h (Fig.7f, h in the previous version) are identical condition. We described this in Fig.8 legend in the revised manuscript.

- Line 259: should be “activates”

We corrected the mistake in line457 in the revised manuscript.

- Line 282: “generates a” or “generates the”

We corrected the mistake in line444 in the revised manuscript.

- Line 288: “generates”

We corrected the mistake in line449 in the revised manuscript.

- Around or after Line 320: It would be nice if the authors explicitly stated their preferred explanation as to *how there could be* “non-autonomous posterior growth induction by anterior Dpp signal” rather than only stating what it is not due to.

We thank the reviewer for the comment. We discussed a couple of possibilities that could explain the non-autonomous posterior growth induction by anterior Dpp signaling as follows (Line542-559);

“In addition to the Dpp signaling-independent lateral growth, requirement of anterior Dpp signaling for posterior growth (Fig. 5j, Extended Data Fig. S2u) and rescue of posterior growth in *dpp* mutants by anterior Dpp signaling (Fig. 6) indicates that anterior Dpp signaling is non-autonomously involved in posterior lateral growth. We note that similar rescue of posterior growth in *dpp* mutant by anterior Dpp signaling has previously been recognized, but the rescued posterior growth was interpreted as growth of the hinge region, without immunostainings for relevant markers⁵⁹. It remains unknown how anterior Dpp signaling contributes to posterior growth. One possibility is that factors from the A compartment may act non-autonomously to promote posterior growth. Such factors include, but are not limited to, direct downstream factors of Dpp signaling. Given that *5xQE.DsRed* is dependent on *Wg*^{45,54}, *Wg* derived from the rescued A compartment may regulate *5xQE.DsRed* expression and growth in the P compartment. Alternatively, mechanical forces may be involved in the non-autonomous growth. It has been proposed that growth factors such as Dpp induce medial growth and subsequently stretch the peripheral regions to induce lateral growth. As the wing disc grows, the peripheral regions in turn compresses the medial region of the wing disc to inhibit the growth of the medial region⁷²⁻⁷⁶. Thus, the growth of the A compartment may stretch the P compartment cells to stimulate their proliferation. It has also been shown that juxtaposition of cells with different Dpp signaling level can induce proliferation non-autonomously but the growth is transient^{26,27}. Therefore, we think it unlikely that the difference of Dpp signaling levels between two compartments can induce sustained growth.”

Reviewers' Comments:

Reviewer #1:

Remarks to the Author:

The authors have addressed the comments and concerns raised previously and the greater detail and explanation in the revised manuscript makes their chain of reasoning easier to follow. The only issue to address is that the image for Fig. 7B appears to be missing. Otherwise, I have no further comments or questions.

Reviewer #2:

Remarks to the Author:

The manuscript by Matsuda et al is significantly improved by the addition of new data and greater clarity in the description of the experiments and results. I only have one concern, which relates to their comparison of HA-trap and Dpp-trap in the paragraph starting on line 295. In line 299, the authors say "To test if the difference could be due to more efficient blocking of Dpp dispersal by Dpp trap than by HA trap, each trap was expressed in the anterior stripe of cells using *ptc-Gal4* and posterior pMad signal was analyzed". This does not make sense to me as the posterior pMad signal depends on dispersal and signaling. To address Dpp dispersal with each trap, I think the authors should directly compare the results from quantitating the Ollas-HA-Dpp stainings (Fig 2d,e and 5b). From what is shown in the Fig 2e and 5b panels, Dpp dispersal looks similar although it is hard to visualize properly without the Ollas-HA-Dpp quantitation on the same graph (as shown for pMad in Extended data Fig 10d).

As minor comments, Fig 7b is missing and I think Extended data Fig 12b and c have the A and P compartment labels the wrong way round.

Reviewer #3:

Remarks to the Author:

The authors have done a very good job of revising their paper, and have adequately addressed my comments.

My only remaining comment concerns the data in Extended Data Figure 11. The authors show PMad staining at the mid-second instar stage. The quantitation appears to be from just one sample. If so, they should quantitate more samples and show a mean and SD.

Reviewer #4:

Remarks to the Author:

In this resubmitted manuscript, the authors were very responsive to the reviewers' concerns. My main criticism was that the explanations of their work were very difficult to follow, not only the flow of logic of the experiments, but the authors previously left out many experimental details that would have made reproducing their work difficult. The authors addressed all of my concerns.

RESPONSE TO REVIEWERS' COMMENTS

Reviewer #1 (Remarks to the Author):

The authors have addressed the comments and concerns raised previously and the greater detail and explanation in the revised manuscript makes their chain of reasoning easier to follow. The only issue to address is that the image for Fig. 7B appears to be missing. Otherwise, I have no further comments or questions.

Thank you very much for the positive comments.
We now added the missing Fig. 7B.

Reviewer #2 (Remarks to the Author):

The manuscript by Matsuda et al is significantly improved by the addition of new data and greater clarity in the description of the experiments and results. I only have one concern, which relates to their comparison of HA-trap and Dpp-trap in the paragraph starting on line 295. In line 299, the authors say "To test if the difference could be due to more efficient blocking of Dpp dispersal by Dpp trap than by HA trap, each trap was expressed in the anterior stripe of cells using *ptc*-Gal4 and posterior pMad signal was analyzed". This does not make sense to me as the posterior pMad signal depends on dispersal and signaling. To address Dpp dispersal with each trap, I think the authors should directly compare the results from quantitating the Ollas-HA-Dpp stainings (Fig 2d,e and 5b). From what is shown in the Fig 2e and 5b panels, Dpp dispersal looks similar although it is hard to visualize properly without the Ollas-HA-Dpp quantitation on the same graph (as shown for pMad in Extended data Fig 10d).

As minor comments, Fig 7b is missing and I think Extended data Fig 12b and c have the A and P compartment labels the wrong way round.

Thank you very much for the positive comments.

Concerning the comparison of HA-trap and Dpp-trap, we agree with the reviewer that we should ideally compare the extracellular distribution of Ollas-HA-Dpp in the P compartment rather than pMad signal in the P compartment in order to compare the efficiency of the two traps. However, both traps are very efficient in trapping Ollas-HA-Dpp and extracellular staining was not sensitive enough to prove the difference in leakage. We thus could not confidently assess the difference from the extracellular staining as the reviewer pointed out (Fig 2e and 5b).

Therefore, we looked at pMad signaling in the P compartment under this condition, since pMad staining (conventional staining) is more robust than the extracellular staining without involving the sensitive extracellular staining protocol and phosphorylation of Mad by potential leaked Dpp concentrates in the nucleus. Furthermore, the posterior pMad activation should reflect the amount of leaked Dpp since the two traps are expressed only in the A compartment. With these points in mind, when we looked at pMad signaling in the P compartment under this condition,

we robustly observed difference in pMad signal activation in the P compartment when the two traps were expressed in the A compartment, and interpreted this result as evidence that Dpp trap is leakier than HA tarp. We explained this point in the manuscript as follows (line 299);

“To test if the difference could be due to more efficient blocking of Dpp dispersal by Dpp trap than by HA trap, each trap was expressed in the anterior stripe of cells using *ptc-Gal4* and posterior pMad signal was analyzed, since the extracellular staining was not sensitive enough to detect significant differences in leakage (Fig 2e and 5b), and the posterior pMad activation would reflect the amount of leaked Dpp since the two traps were specifically expressed in the anterior stripe of cells.”

We also added the missing Fig. 7B, and corrected Extended Data Fig. 12b/c for the labeling of A/P compartment.

Reviewer #3 (Remarks to the Author):

The authors have done a very good job of revising their paper, and have adequately addressed my comments.

My only remaining comment concerns the data in Extended Data Figure 11. The authors show pMad staining at the mid-second instar stage. The quantitation appears to be from just one sample. If so, they should quantitate more samples and show a mean and SD.

Thank you very much for the positive comments.

Although pMad staining is technically difficult in the early stage due to the small disc size, we now included a pMad gradient at the mid-second instar stage with a mean and SD from more samples (n=3) and updated Extended Data Figure 11.

Reviewer #4 (Remarks to the Author):

In this resubmitted manuscript, the authors were very responsive to the reviewers' concerns. My main criticism was that the explanations of their work were very difficult to follow, not only the flow of logic of the experiments, but the authors previously left out many experimental details that would have made reproducing their work difficult. The authors addressed all of my concerns.

Thank you very much for the positive comments.